# RoFt-Mol: Benchmarking Robust Fine-Tuning with Molecular Graph Foundation Models

## Abstract

In the era of foundation models, fine-tuning pre-trained models for specific downstream tasks has become crucial. This drives the need for robust fine-tuning methods to address challenges such as model overfitting and sparse labeling. Molecular graph foundation models (MGFMs) face unique difficulties that complicate fine-tuning. These models are limited by smaller pre-training datasets and more severe data scarcity for downstream tasks, both of which require enhanced model generalization. Moreover, MGFMs must accommodate diverse pre-training objectives, including both regression and classification tasks. To better understand and improve fine-tuning techniques under these conditions, we classify eight fine-tuning methods into three mechanisms: weight-based fine-tuning, representation-based fine-tuning, and partial fine-tuning. We benchmark these methods on downstream regression and classification tasks across both supervised and self-supervised pre-trained models in diverse labeling settings. This extensive evaluation provides valuable insights and informs the design of a refined robust fine-tuning method, DWiSE-FT. This approach combines the strengths of simple post-hoc weight interpolation with more complex weight ensemble fine-tuning methods, delivering improved performance across both task types while maintaining the ease of use inherent in post-hoc weight interpolation.

## 1 Introduction

In recent years, foundation models (Bommasani et al., 2021; Zhou et al., 2023) have achieved remarkable success in learning high-quality, general-purpose representations of images and text through pre-training on diverse datasets (Radford et al., 2021; Kirillov et al., 2023; Ramesh et al., 2022; Touvron et al., 2023; Bubeck et al., 2023; Zhao et al., 2023). To adapt these pre-trained models for downstream applications, additional training on task-specific data, known as fine-tuning, is often required. However, vanilla fine-tuning frequently encounters performance challenges, including model overfitting (Howard & Ruder, 2018; Li et al., 2020a; Kornblith et al., 2019), catastrophic forgetting of pre-trained knowledge (Lee et al., 2022; Li et al., 2019b; Xuhong et al., 2018; Lubana et al., 2022), and distribution shifts between fine-tuned and test samples, which can lead to negative transfer (Wang et al., 2019; Chen et al., 2019). These challenges highlight the need for robust fine-tuning strategies (Shen et al., 2021; Wortsman et al., 2022; Kumar et al., 2022; Shu et al., 2023; Andreassen et al., 2021; Kirichenko et al., 2022).

Recently, the advantages of foundation models have been extended to various scientific applications (Golling et al., 2024; Leung & Bovy, 2024; Nguyen et al., 2023). Among these, molecular graph foundation models (MGFMs) have gained significant attention for their promising potential in biochemistry (Hu et al., 2020a; Hou et al., 2022b; Xia et al., 2023b; Suresh et al., 2021; Shoghi et al., 2023; Beaini et al., 2023; Zheng et al., 2023; Ross et al., 2022; Rong et al., 2020; Mao et al., 2024). While MGFMs exhibit scaling behaviors similar to foundation models in other domains (Chen et al., 2024), they face unique challenges related to data and tasks.

A primary challenge stems from the significantly smaller pre-training datasets in this domain, typically consisting of at most $O(100M)$ molecular samples, compared to the billions of samples used in other domains (Sun et al., 2022). This limitation restricts the parameter scale of MGFMs ($O(100M)$ parameters) and their generalization capacity (Wang et al., 2024; Akhondzadeh et al., 2023). Furthermore, downstream tasks in this domain often involve limited data for fine-tuning, with datasets

containing only tens or a few hundred labeled samples (Wijaya et al., 2024), exacerbating the difficulty of achieving robust model generalization. In addition to data constraints, many downstream tasks, such as molecular property prediction, are regression-based (Wu et al., 2018; Hou et al., 2022a). These tasks require models to capture fine-grained numerical patterns, which presents a distinct requirement compared to the coarse-grained feature reliance typical in classification tasks in CV and NLP. These factors collectively highlight the need for a careful examination of fine-tuning strategies for MGFMs and their appropriate improvement.

To address this gap, we introduce ROFT-MOL, a benchmark designed to evaluate existing fine-tuning methods across diverse molecular property prediction tasks, including 8 classification and 4 regression tasks. To investigate the factors influencing the fine-tuning performance of MGFMs, we categorize 8 finetuning (FT) methods into 3 distinct mechanisms: 1) *weight-based FT*, which ensembles the weights from both pre-trained and fine-tuned models, 2) *representation-based FT*, which regularizes the proximity between pre-trained and fine-tuned latent data representations, and 3) *partial FT*, which optimizes only a subset of the pre-trained model's weights while keeping the rest frozen. To simulate the challenges encountered during the pre-training and fine-tuning stages of MGFMs, we evaluate models from both self-supervised and supervised pre-training, and assess their fine-tuning performance in few-shot and out-of-distribution settings. We summarize high-level insights as follows, with further detailed results presented in Sec. 4:

- **Different fine-tuning methods:** For self-supervised pre-trained models, *weight-based fine-tuning* often results in better performance by effectively integrating general knowledge from pre-training with task-specific knowledge from fine-tuning [**Finding 1**]. On the other hand, *partial fine-tuning* typically leads to underfitted molecular representations in few-shot fine-tuning, particularly for regression tasks [**Finding 2**]. For supervised pre-trained models, *representation-based fine-tuning* performs well due to the preservation of domain-relevant pre-trained representations [**Finding 4**].
- **Classification vs. Regression downstream tasks:** Due to the need for more precise numerical labels and finer molecule modeling, MGFMs generally face less risk of overfitting in regression tasks compared to classification tasks, particularly in the few-shot setting [**Q1**].
- **Supervised pre-trained vs. Self-supervised pre-trained models:** In few shot fine-tuning, supervised pre-training, which often involves more domain-relevant tasks, generally yields better finetuning performance than self-supervised pre-training based on more generic synthetic tasks. This holds true even when the pre-training tasks do not align well with the finetuning tasks. In contrast, for non-few-shot settings, supervised pre-training performs better only when the supervised pre-training tasks closely align with the downstream tasks [**Q2**].

Inspired by Finding 1 and Q1, we propose a **new method, DWiSE-FT**. We observe that simple post-hoc weight interpolation between pre-trained and fine-tuned model weights (WiSE-FT) performs well for classification tasks but struggles with regression tasks. In contrast, a more complex weight ensemble approach ($L^2$-SP) achieves better performance in regression tasks, though it comes with the cost of increased tuning complexity. DWiSE-FT combines the strengths of WiSE-FT and $L^2$-SP, providing strong performance for both task types while maintaining the plug-and-play ease of post-hoc interpolation. The success of DWiSE-FT illustrates how this benchmark can provide valuable insight for fine-tuning strategies for MGFMs.

## 2 PRELIMINARIES

As preliminaries, we briefly introduce representative methodologies used in pre-training and fine-tuning for molecular graph foundation models.

**Self-supervised Pre-training** strategies have been proven to be effective in generating transferable molecular representations for downstream tasks (Zhao et al., 2024). In a high level, they can be divided into *reconstruction* methods and *contrastive* methods. The generative-based strategies adopt mask-based graph reconstruction by utilizing graph autoencoders (Hou et al., 2022b; Tan et al., 2023; Wang et al., 2017; Pan et al., 2018), context predictions (Hu et al., 2020a; Rong et al., 2020) and generative language model pre-training (Hu et al., 2020b; Zhang et al., 2021b). On the other hand, contrastive-based methods aim for maximizing the similarity between perturbed instance pairs (Veličković et al., 2018; Suresh et al., 2021; You et al., 2020; Xia et al., 2023a; Wang et al., 2022; Zhu et al., 2022; You et al., 2021; Qiu et al., 2020; Li et al., 2022; Xu et al., 2021). Moreover, the advancement of language models has prompted numerous studies to employ multi-modal frame-

works. These approaches harness language models to enhance molecular understanding through techniques such as cross-modal contrastive learning and cross-modal alignment (Su et al., 2022; Liu et al., 2023a; Seidl et al., 2023; Liu et al., 2023b). In this work, we select *GraphMAE* (Hou et al., 2022b) as the representative of the recontruction-based pre-trained model, which focuses on masked feature reconstruction with scaled cosine error that enabled robust training. Regarding the contrastive pre-trained model, we choose *Mole-BERT* (Xia et al., 2023a) that combines the node-level masked atom modeling to predict the masked atom tokens and the graph-level contrastive learning through triplet loss and contrastive loss. Lastly, we choose *MoleculeSTM* (Liu et al., 2023a) as the representative of multi-modal molecule structure-text model that jointly learning molecules' chemical structures and textual descriptions via a contrastive learning strategy.

**Supervised Pre-training**. Recently, in order to leverage more diversified datasets and prediction tasks, researchers have started exploring the capability of supervised pre-training with multi-task learning for molecular representations (Gasteiger et al., 2022; Shoghi et al., 2023; Beaini et al., 2023). We adopt the pre-trained model by being trained on multi-task labeled samples in the supervised manner from the *Graphium* library (Beaini et al., 2023).

The overall goal for fine-tuning is to adapt the pre-trained model to downstream applications. Specifically, given a pre-trained GNN encoder $f_{\boldsymbol{\theta}}$ with parameters $\boldsymbol{\theta}$ initialized from the pretrained parameters $\boldsymbol{\theta}_{\text{pre}}$, fine-tuning optimizes the encoder $f_{\boldsymbol{\theta}}$ and an additional prediction head $g_{\boldsymbol{\phi}}$ with parameters $\boldsymbol{\phi}$ over downstream molecules $\{(\mathcal{G}_i, y_i)\}_{i=1}^N$. The vanilla fine-tuning version, **full-FT**, optimizes the entire model weights following:

$$\min_{\{\boldsymbol{\theta}, \boldsymbol{\phi}\}} \sum_{i=1}^N \mathcal{L}(g_{\boldsymbol{\phi}} \circ f_{\boldsymbol{\theta}}(\mathcal{G}_i), y_i), \quad \text{where } \boldsymbol{\theta} \text{ is initialized as } \boldsymbol{\theta}_{\text{pre}}. \tag{1}$$

Here, $\mathcal{L}$ denotes the loss function for prediction tasks. As discussed, there are advanced fine-tuning strategies proposed on top of the full-FT framework. As shown in Fig. 1, we group them into three categories based on their mechanisms and benchmark representative methods for each category. More fine-tuning methods that fall into each category or others will be discussed in Appendix C.

• **Partial model FT** strategies only optimizes partial weights of the pre-trained model. Namely, a subset of weights within $\{\boldsymbol{\theta}, \boldsymbol{\phi}\}$ will be updated following the same objective as Eq. 1.

† *Linear Probing (LP)* only trains the additional prediction head $g$ during the FT.

† *Surgical FT* (Lee et al., 2022) updates only partial layers within the encoder. For instance, we can update the weights for $k$-th layer of the GNN encoder as $\min_{\{[\boldsymbol{\theta}]_k, \boldsymbol{\phi}\}} \sum_{i=1}^N \mathcal{L}(g_{\boldsymbol{\phi}} \circ f_{\boldsymbol{\theta}}(\mathcal{G}_i), y_i)$, where $k$ is the hyperparameter that can be tuned.

† *LP-FT* (Kumar et al., 2022) aims to address the issue of pre-trained feature distortion during the full-FT process. It first performs the LP step to the prediction head $g_{\boldsymbol{\phi}}$ while keeping the encoder $f_{\boldsymbol{\theta}}$ with fixed pre-trained parameters $\boldsymbol{\theta}_{\text{pre}}$, followed by applying full-FT with the tuned prediction head.

• **Weight-based FT** strategies mainly update the entire model weights through combining pre-trained model weights and fine-tuned model weights.

† *WiSE-FT* (Wortsman et al., 2022) linearly interpolates between pre-training parameters $\boldsymbol{\theta}_{\text{pre}}$ and fine-tuning parameters $\boldsymbol{\theta}_{\text{ft}}$ using a mixing coefficient $\alpha$, to get the interpolated GNN $f_{\boldsymbol{\theta}_{\text{int}}}$ with weights $\boldsymbol{\theta}_{\text{int}} = (1 - \alpha) \cdot \boldsymbol{\theta}_{\text{pre}} + \alpha \cdot \boldsymbol{\theta}_{\text{ft}}$. We first perform full-FT to obtain the adapted encoder $f_{\boldsymbol{\theta}_{\text{ft}}}$ and classifier $g_{\boldsymbol{\phi}}$, then apply post-hoc weight ensembling to get $f_{\boldsymbol{\theta}_{\text{int}}}$, with final predictions given by $g_{\boldsymbol{\phi}} \circ f_{\boldsymbol{\theta}_{\text{int}}}(\mathcal{G}_i)$. $\alpha$ is tuned as a hyperparameter to control the weight ensemble.

† $L^2$-*SP* (Xuhong et al., 2018) regularizes the fine-tuning model weights $\boldsymbol{\theta}$ closer to the pre-trained weights $\boldsymbol{\theta}_{\text{pre}}$ by $\Omega(\boldsymbol{\theta}, \boldsymbol{\phi}) = \frac{\delta}{2} \|\boldsymbol{\theta} - \boldsymbol{\theta}_{\text{pre}}\|_2^2$. We optimize for $\boldsymbol{\theta}$ and $\boldsymbol{\phi}$ by combining the prediction loss from Eq. 1 and $\Omega(\boldsymbol{\theta}, \boldsymbol{\phi})$ with tunable trade-off coefficient $\delta$.

• **Representation-based FT** methods mainly regulate the latent representation space during FT.

† *Feature-map* (Li et al., 2019b) adds distance regularization between the latent representations of pre-trained and fine-tuned models to the Full-FT loss. The regularization is defined as $\Omega(\boldsymbol{\theta}) = \delta \sum_{i=1}^N \frac{1}{2} \|f_{\boldsymbol{\theta}}(\mathcal{G}_i) - f_{\boldsymbol{\theta}_{\text{pre}}}(\mathcal{G}_i)\|_2^2$, where $\delta$ controls the regularization strength.

† *BSS* (Chen et al., 2019) aims at resolving the negative transfer issue through eliminating the spectral components corresponding to small singular values that are less transferable. The regularization is done as $\Omega(\boldsymbol{F}) = \delta \sum_{i=1}^k \sigma_{-i}^2$, where $\boldsymbol{F} = [f_{\boldsymbol{\theta}}(\mathcal{G}_0), \cdots, f_{\boldsymbol{\theta}}(\mathcal{G}_b)]$ is the feature matrix of a

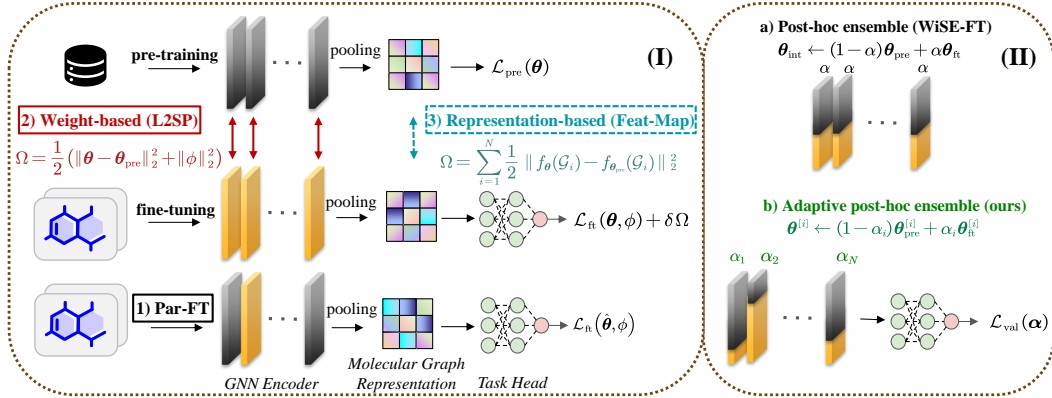

Figure 1: The overall framework of fine-tuning strategies evaluated in our benchmark, ROFT-MOL, and the proposed novel method, DWiSE-FT. **(I)** The GNN encoder is pre-trained on a large database by the pre-training objective $\mathcal{L}_{\text{pre}}$, and fine-tuned on the downstream dataset by $\mathcal{L}_{\text{ft}}$ as stated in Eq. 1. 1) Partial-FT, 2) Weight-based FT, and 3) Representation-based FT achieve robust fine-tuning by freezing partial pre-trained model weights, regularizing model weights and latent representations, respectively. **(II)** The refined method DWiSE-FT that combines the strength of simple post-hoc weight interpolation with more complex weight ensemble, demonstrating the improved performance while maintaining easy usage.

batch of graphs and $\sigma_{-i}$ are the $i$-th smallest singular values obtained from the SVD of $\boldsymbol{F}$. We can tune $k$ and $\delta$ to determine the number of singular values to penalize and the degree of penalty.

## 3 EXPERIMENTAL SETTINGS

In this section, We briefly introduce the experimental settings in this work, including the selection of foundation models and datasets, the strategies of dataset splitting and fine-tuning training size configurations, as well as evaluation metrics. The selection of fine-tuning algorithms can be seen in Sec. 2. More detailed experimental settings like hyperparameters tuning and training implementations can be found in Appendix E.

**Foundation Models**. For self-supervised pre-training, we adopt the open-source pre-trained checkpoints from *Mole-BERT* and *GraphMAE* both of which are pre-trained over 2M molecules sampled from the ZINC15 database (Sterling & Irwin, 2015), following previous works (Hu et al., 2019). For *MoleculeSTM*, we utilize the publicly available pre-trained checkpoint. This model is initially trained on PubChemSTM, a large multimodal dataset comprising over 280,000 chemical structure–text pairs contructed from the PubChem database (Kim et al., 2021). For supervised pre-training, we use the model from the *Graphium* (Beaini et al., 2023) library, which gets pre-trained on the Toymix dataset provided in this library. Here, we consider adopting the Toymix dataset mainly due to the data-processing computation constraints and to keep a more fair comparison to the other self-supervised pre-trained models in terms of pre-training model and data scale. The ToyMix dataset (Beaini et al., 2023), totally 154K molecules, contains QM9 (Ramakrishnan et al., 2014), Tox21 (Wu et al., 2018) and Zinc12K (Dwivedi et al., 2023). Specifically, QM9 consists of 19 graph-level quantum properties associated to an energy-minimized 3D conformation of the molecules. Zinc12K is to predict the constrained solubility which is the term $\log \text{P} - \text{SA} - \text{cycle}$ (octanol-water partition coefficients, logP, penalized by the synthetic accessibility score, SA, and number of long cycles, cycle). The pre-trained model size is around 2M parameters and the GIN backbone is known as having same expressive power as 1-WL test, which cannot distinguish non-isomorphic graphs that 1-WL fails to differentiate (Xu et al., 2018).

**Downstream Datasets**. We use 8 classification and 4 regression datasets for downstream task evaluation as follows. Detailed statistics for the 12 downstream tasks are in Appendix D.

† *Classification.* The BBBP (Martins et al., 2012) dataset measures if a molecule will penetrate blood-brain barrier. All three datasets, Tox21, ToxCast (Richard et al., 2016), and ClinTox (Gayvert et al., 2016) are related to toxicity qualitative measurements. The Sider (Kuhn et al., 2016) dataset stores qualitative results of different types of adverse drug reactions. The MUV dataset (Rohrer & Baumann, 2009) contains 17 challenging tasks and is specifically designed for validation of virtual

screening techniques. The HIV, collected from Zaharevitz (2015), provides qualitative activity results of the molecular ability to inhibit HIV replication. BACE (Subramanian et al., 2016) contains qualitative binding results for a set of inhibitors of human $\beta$-secretase 1 (BACE-1).

† *Regression.* Esol (Delaney, 2004) is a standard regression dataset which measures aqueous solubility of molecules. The Lipo dataset is a subset of ChEMBL (Gaulton et al., 2012) measuring the octanol-water partition coefficient. Cep is a subset of the Havard Clean Energy Project (CEP) (Hachmann et al., 2011), which estimates the organic photovoltaic efficiency. Malaria (Gamo et al., 2010) measures the drug efficacy against the parasite that causes malaria.

**Dataset Splits**. For each downstream dataset, we experiment with *random, scaffold*, and *size* splits to create the Train/Val/Test subsets. Specifically, the random splitting shuffles the data, maintaining the Train/Val/Test sets as in-distribution (ID). The other two splitting methods simulate out-of-distribution (OOD) challenges in real-world applications. For scaffold splitting, we follow prior works (Ramsundar et al., 2019), ensuring structural differences in molecular scaffolds across splits. Size splitting, following (Zou et al., 2023), arranges molecules in ascending order by size, evaluating model generalization across different molecule sizes.

**Number of fine-tuning samples**. In practice, molecular property prediction tasks can have very limited experimentally-validated data, e.g. with less than 100 samples (Wijaya et al., 2024). Thus, we consider both *non-few-shot* and *few-shot* settings to better simulate the label scarcity issue. In the non-few-shot setting, we use all available samples from the splitted train set. In the few-shot settings, we sample subsets of 50, 100, and 500 molecules from the Train set for fine-tuning, while keeping the Val/Test sets unchanged to ensure a fair comparison. Note that we exclude MUV, Tox21, and ToxCast datasets for the fewshot settings, as we cannot *randomly* select training samples while ensuring that all tasks have a specified number of labels simultaneously, due to the severe label scarcity issues in these datasets.

**Evaluation**. We use AUC to evaluate the performance for classification datasets and RMSE for regression datasets. We report the model performance over 5 random seeds and the test performance are reported based on the best validation performance. The AVG, AVG-F, AVG-R denote the average metrics, average metrics without max and min values, and average rank over all the datasets for each evaluated method, respectively.

## 4 RESULTS AND ANALYSIS

We put experimental results of Mole-BERT (self-supervised) and Graphium (supervised) models under the non-few-shot setting to Table 1 and 2, and visualize results of these two models under the few-shot-50 and 100 settings to Fig. 2. The results of few-shot-500 settings are put in Appendix F due to the limited space. Also, the results of the Graph-MAE and MoleculeSTM model, which we find follow similar trends with Mole-BERT, are put in Appendix F. In each section, we begin by analyzing how different pre-training objectives influence the downstream finetuning and then present the findings after accessing different fine-tuning strategies across each experimental setting.

### 4.1 SELF-SUPERVISED PRE-TRAINED MODELS

*Q1: Can self-supervised pre-training help downstream molecular property prediction tasks?*

**(1a) Molecular representations learned from self-supervised pre-training are not informative enough for downstream tasks. In particular, regression tasks require more task-specific knowledge from downstream fine-tuning compared to classification tasks.**

As shown in Tables 1 and 2, as well as Fig. 2a and 2c, LP is consistently the worst performing method for self-supervised pre-trained models across all data splits, even under the few-shot fine-tuning. This contrasts the observations in CV where LP demonstrates robust OOD performance by preserving high quality and generalizable features from pre-trained embeddings (Wortsman et al., 2022; Kumar et al., 2022). We attribute this to the misalignment between general-purpose representations produced by self-supervised pre-training and the features required by the specific molecular tasks. Consequently, relying solely on tuning the classifier $g_\phi$ is insufficient to extract meaningful predictions from these non-informative representations.

Table 1: Robust fine-tuning performance on 8 classification datasets (AUC metrics) in the Non-Fewshot setting, evaluated across 3 dataset splits (RANDOM, SCAFFOLD, SIZE) and 2 pre-training strategies (SELF-SUPERVISED, SUPERVISED). AVG, AVG-F, AVG-R denote the average AUC metrics, average AUC without max and min values, and average rank over all the datasets for each evaluated method, respectively. Standard deviations across five replicates are shown in parentheses. We **bold** and underline the best and second-best performances in each scenario.

| SPLIT | METHODS | CLINTOX | BBBP | BACE | HIV | MUV | SIDER | TOX21 | TOXCAST | AVG | AVG-F | AVG-R |
|---|---|---|---|---|---|---|---|---|---|---|---|---|
| | | | | | SELF-SUPERVISED PRE-TRAINING (MOLE-BERT) | | | | | | | |
| RANDOM | FULL-FT | 87.17 ± 0.28 | 93.52 ± 0.37 | 89.27 ± 0.21 | 85.98 ± 0.44 | 85.34 ± 0.82 | 61.94 ± 0.99 | 83.45 ± 0.34 | 74.74 ± 0.35 | 82.68 | 84.33 | 4.62 |
| | LP | 84.80 ± 0.41 | 90.26 ± 0.17 | 77.31 ± 0.18 | 79.09 ± 0.38 | **88.38 ± 0.71** | 61.17 ± 0.20 | 83.90 ± 0.09 | 72.86 ± 0.17 | 79.72 | 81.06 | 6.12 |
| | SURGICAL-FT | **90.14 ± 1.61** | **94.19 ± 0.35** | 89.20 ± 0.16 | **86.81 ± 0.24** | 87.88 ± 1.20 | 61.60 ± 0.46 | **84.03 ± 0.30** | 74.66 ± 0.22 | 83.56 | 85.45 | 2.62 |
| | LP-FT | 87.65 ± 2.00 | 93.43 ± 0.40 | **90.17 ± 0.16** | 85.70 ± 0.29 | 86.99 ± 0.02 | **63.14 ± 0.30** | 83.84 ± 0.60 | 73.65 ± 0.22 | 83.07 | 84.67 | 3.38 |
| | WISE-FT | 87.73 ± 0.83 | 94.15 ± 0.46 | 89.26 ± 0.58 | 85.89 ± 0.57 | 86.38 ± 1.56 | 62.13 ± 0.62 | 83.54 ± 0.29 | 74.86 ± 0.22 | 82.99 | 84.61 | 3.50 |
| | $L^2$-SP | 87.62 ± 1.26 | 93.81 ± 0.49 | 89.11 ± 0.65 | 82.39 ± 0.50 | 83.72 ± 0.19 | 60.92 ± 0.59 | 83.73 ± 0.19 | 72.59 ± 0.11 | 81.74 | 83.19 | 5.75 |
| | FEATURE-MAP | 86.36 ± 2.49 | 92.01 ± 0.19 | 81.15 ± 0.26 | 80.66 ± 0.64 | 86.49 ± 0.69 | 61.62 ± 0.41 | 82.25 ± 0.08 | 73.20 ± 0.23 | 80.47 | 81.69 | 6.38 |
| | BSS | 87.61 ± 0.66 | 93.74 ± 0.51 | 89.38 ± 0.54 | 86.42 ± 0.36 | 80.20 ± 0.44 | 62.36 ± 0.65 | 83.61 ± 0.12 | **75.67 ± 0.32** | 82.37 | 83.81 | 3.62 |
| SCAFFOLD | FULL-FT | **77.70 ± 1.50** | 67.93 ± 3.85 | 80.12 ± 1.07 | 77.00 ± 0.80 | 80.50 ± 0.81 | 63.47 ± 0.77 | 78.31 ± 0.28 | 65.18 ± 0.35 | 73.78 | 74.37 | 3.75 |
| | LP | 66.49 ± 0.46 | 65.42 ± 0.26 | 78.70 ± 0.27 | 77.15 ± 0.12 | 79.27 ± 0.48 | 62.01 ± 0.60 | 78.12 ± 0.15 | 64.75 ± 0.17 | 71.49 | 71.77 | 6.12 |
| | SURGICAL-FT | 68.19 ± 1.58 | 67.70 ± 0.54 | **84.24 ± 0.37** | 76.65 ± 0.46 | 81.60 ± 1.02 | 64.61 ± 0.31 | 78.34 ± 0.10 | 65.21 ± 0.28 | 73.32 | 72.95 | 3.62 |
| | LP-FT | 70.35 ± 0.99 | **68.30 ± 0.65** | 81.90 ± 0.70 | 76.69 ± 0.40 | 77.65 ± 1.15 | 63.38 ± 0.67 | 77.60 ± 0.19 | 65.32 ± 0.24 | 72.65 | 72.65 | 4.88 |
| | WISE-FT | 73.59 ± 3.74 | 66.52 ± 3.29 | 82.73 ± 0.87 | **77.21 ± 0.69** | **81.92 ± 0.94** | 63.62 ± 0.62 | 78.05 ± 0.28 | 65.41 ± 0.25 | 73.63 | 73.78 | 3.38 |
| | $L^2$-SP | 73.95 ± 1.86 | 67.86 ± 1.68 | 81.47 ± 0.80 | 76.63 ± 0.56 | 77.21 ± 0.72 | **65.27 ± 0.45** | **78.66 ± 0.17** | 63.55 ± 0.16 | 73.07 | 73.26 | 4.50 |
| | FEATURE-MAP | 70.65 ± 0.76 | 65.41 ± 2.37 | 73.44 ± 0.23 | 76.71 ± 0.26 | 80.03 ± 0.47 | 64.35 ± 0.17 | 76.61 ± 0.39 | **65.77 ± 0.15** | 71.62 | 71.43 | 5.25 |
| | BSS | 76.07 ± 3.23 | 67.47 ± 3.80 | 80.98 ± 1.27 | 77.12 ± 0.86 | 77.35 ± 1.76 | 63.88 ± 0.80 | 78.19 ± 0.40 | 65.00 ± 0.27 | 73.26 | 73.53 | 4.50 |
| SIZE | FULL-FT | 72.78 ± 1.74 | 87.37 ± 0.82 | 66.00 ± 1.99 | 79.85 ± 0.64 | 77.02 ± 2.15 | 52.46 ± 0.29 | 75.74 ± 0.48 | 63.13 ± 0.32 | 71.79 | 72.42 | 4.88 |
| | LP | **76.07 ± 0.32** | 82.73 ± 0.76 | 47.18 ± 0.45 | 78.16 ± 0.24 | 78.52 ± 1.60 | 51.25 ± 0.22 | 74.92 ± 0.22 | 63.33 ± 0.20 | 69.02 | 70.37 | 6.00 |
| | SURGICAL-FT | 73.55 ± 0.81 | **88.82 ± 0.53** | 66.43 ± 0.88 | 79.30 ± 0.87 | **80.52 ± 1.47** | 51.87 ± 0.23 | 76.32 ± 0.16 | **64.51 ± 0.20** | 72.66 | 73.44 | 3.50 |
| | LP-FT | 75.32 ± 0.93 | 83.42 ± 1.67 | 64.84 ± 1.38 | 79.10 ± 1.14 | 79.38 ± 1.86 | 52.82 ± 0.32 | 76.31 ± 0.29 | 63.37 ± 0.29 | 71.83 | 73.07 | 3.88 |
| | WISE-FT | 73.45 ± 1.08 | 87.79 ± 1.53 | 66.58 ± 1.11 | 79.89 ± 1.75 | 78.41 ± 1.88 | 52.46 ± 0.49 | 76.46 ± 0.46 | 63.53 ± 0.65 | 72.32 | 73.05 | 3.00 |
| | $L^2$-SP | 73.97 ± 0.88 | 87.15 ± 0.68 | 64.58 ± 1.93 | **80.05 ± 0.53** | 74.83 ± 1.06 | 52.37 ± 0.22 | 75.84 ± 0.28 | 60.63 ± 0.36 | 71.18 | 71.65 | 5.12 |
| | FEATURE-MAP | 74.61 ± 0.53 | 85.42 ± 0.31 | 51.23 ± 0.46 | 76.39 ± 0.91 | 75.20 ± 2.27 | 51.96 ± 0.26 | **76.81 ± 0.25** | 63.42 ± 0.76 | 69.38 | 69.73 | 5.00 |
| | BSS | 73.99 ± 0.77 | 86.84 ± 1.00 | 66.97 ± 1.58 | 79.64 ± 1.44 | 73.42 ± 2.60 | **53.50 ± 0.66** | 75.69 ± 0.26 | 62.41 ± 0.69 | 71.56 | 72.02 | 4.62 |
| | | | | | SUPERVISED PRE-TRAINING (GRAPHIUM) | | | | | | | |
| RANDOM | FULL-FT | 94.42 ± 2.36 | 92.25 ± 0.88 | 88.54 ± 0.72 | 83.87 ± 1.03 | 77.08 ± 1.58 | 58.19 ± 0.21 | 82.91 ± 0.33 | 73.61 ± 0.23 | 81.36 | 83.04 | 4.12 |
| | LP | 93.66 ± 0.00 | 87.00 ± 0.00 | 83.77 ± 0.00 | 77.67 ± 0.00 | **79.65 ± 0.00** | 59.29 ± 0.00 | 81.14 ± 0.00 | 71.14 ± 0.00 | 79.41 | 80.39 | 5.62 |
| | SURGICAL-FT | **96.27 ± 0.00** | **93.12 ± 0.00** | 90.11 ± 0.00 | **84.20 ± 0.00** | 76.43 ± 0.00 | 59.80 ± 0.00 | 83.19 ± 0.00 | 73.80 ± 0.00 | 82.12 | 83.48 | 2.50 |
| | LP-FT | 93.56 ± 1.21 | 91.70 ± 0.79 | 89.33 ± 0.79 | 83.54 ± 0.90 | 75.60 ± 1.48 | 59.64 ± 0.29 | 83.03 ± 0.52 | 72.82 ± 0.00 | 81.22 | 82.71 | 4.25 |
| | WISE-FT | 93.37 ± 2.74 | 91.80 ± 0.39 | 88.31 ± 0.79 | 82.99 ± 0.94 | 76.15 ± 3.11 | 59.53 ± 0.30 | 83.03 ± 0.52 | 73.28 ± 0.21 | 81.06 | 82.59 | 5.00 |
| | $L^2$-SP | 90.82 ± 2.30 | 88.80 ± 1.01 | 85.41 ± 0.52 | 64.96 ± 0.05 | 67.30 ± 0.00 | **60.56 ± 1.73** | 83.71 ± 0.24 | 70.35 ± 0.32 | 76.49 | 76.76 | 5.75 |
| | FEATURE-MAP | 95.40 ± 0.39 | 92.08 ± 0.47 | **90.79 ± 0.03** | 69.54 ± 0.09 | **78.25 ± 0.07** | 60.38 ± 0.03 | **84.73 ± 0.04** | 69.73 ± 0.02 | 80.11 | 80.85 | 3.12 |
| | BSS | 90.07 ± 3.70 | 90.46 ± 0.83 | 85.22 ± 0.67 | 67.00 ± 0.01 | 66.63 ± 1.68 | 59.43 ± 1.34 | 83.81 ± 0.63 | **74.05 ± 0.44** | 77.08 | 77.80 | 5.62 |
| SCAFFOLD | FULL-FT | 81.27 ± 3.88 | 69.17 ± 1.32 | 79.75 ± 1.07 | 76.42 ± 0.72 | 76.84 ± 1.80 | 63.63 ± 0.06 | 78.12 ± 0.46 | 66.37 ± 0.26 | 73.95 | 74.45 | 3.75 |
| | LP | 80.48 ± 0.00 | 66.90 ± 0.00 | 80.44 ± 0.00 | 75.83 ± 0.00 | 73.35 ± 0.00 | 62.03 ± 0.00 | 79.02 ± 0.00 | 66.09 ± 0.00 | 73.02 | 73.61 | 5.12 |
| | SURGICAL-FT | 86.17 ± 0.00 | **73.71 ± 0.00** | 84.16 ± 0.00 | **77.47 ± 0.00** | **78.87 ± 0.00** | 64.02 ± 0.00 | 78.23 ± 0.00 | 67.34 ± 0.00 | 76.25 | 76.63 | 1.38 |
| | LP-FT | 83.67 ± 3.53 | 69.98 ± 0.83 | 79.28 ± 0.32 | 76.17 ± 2.01 | 77.82 ± 1.15 | 61.20 ± 0.00 | 76.94 ± 0.00 | 66.28 ± 0.00 | 73.92 | 74.41 | 4.62 |
| | WISE-FT | 85.40 ± 1.61 | 71.89 ± 1.79 | 78.13 ± 2.92 | 76.69 ± 1.76 | 74.37 ± 1.79 | 63.58 ± 0.00 | 77.98 ± 0.33 | 66.48 ± 0.43 | 74.31 | 74.26 | 3.62 |
| | $L^2$-SP | 76.83 ± 8.87 | 67.35 ± 0.82 | 78.17 ± 0.02 | 73.69 ± 0.03 | 62.35 ± 0.15 | 62.21 ± 0.43 | 76.27 ± 0.32 | 62.75 ± 0.88 | 69.95 | 69.87 | 6.62 |
| | FEATURE-MAP | **90.13 ± 2.12** | 70.99 ± 0.27 | 83.17 ± 0.49 | 73.61 ± 0.03 | 78.74 ± 0.76 | 62.12 ± 0.02 | **79.99 ± 0.12** | 65.03 ± 0.08 | 75.47 | 75.25 | 3.50 |
| | BSS | 79.99 ± 5.89 | 67.10 ± 0.93 | 78.12 ± 2.32 | 72.50 ± 0.51 | 61.20 ± 0.08 | 61.13 ± 0.95 | 76.69 ± 0.64 | 65.45 ± 0.89 | 70.27 | 70.18 | 7.38 |
| SIZE | FULL-FT | 85.96 ± 4.28 | 87.62 ± 0.90 | 67.41 ± 2.44 | 81.47 ± 1.94 | 72.03 ± 2.55 | 54.72 ± 0.01 | 81.31 ± 0.37 | 61.31 ± 0.37 | 72.53 | 72.98 | 3.88 |
| | LP | 81.84 ± 0.02 | 78.09 ± 0.00 | 58.08 ± 0.01 | 77.48 ± 0.00 | 69.46 ± 0.00 | 53.59 ± 0.00 | 73.65 ± 0.00 | 61.25 ± 0.00 | 69.18 | 69.67 | 5.38 |
| | SURGICAL-FT | 86.59 ± 0.01 | **89.07 ± 0.00** | 70.94 ± 0.01 | 82.50 ± 0.00 | **74.47 ± 0.00** | **56.24 ± 0.00** | 72.30 ± 0.00 | **62.74 ± 0.00** | 74.36 | 74.92 | 1.62 |
| | LP-FT | **86.78 ± 2.69** | 88.02 ± 1.50 | 63.72 ± 1.85 | **82.57 ± 0.46** | 73.51 ± 1.77 | 52.40 ± 0.00 | 68.23 ± 0.87 | 60.85 ± 0.00 | 72.01 | 72.61 | 4.00 |
| | WISE-FT | 82.44 ± 3.02 | 87.76 ± 0.15 | 68.82 ± 0.06 | **72.89 ± 0.66** | 81.37 ± 1.07 | 73.67 ± 3.44 | 55.87 ± 0.01 | 60.61 ± 0.53 | 72.93 | 73.31 | 3.62 |
| | $L^2$-SP | 71.03 ± 3.67 | 81.32 ± 1.51 | 68.82 ± 0.06 | 70.66 ± 0.00 | 64.69 ± 0.32 | 52.08 ± 0.84 | 70.91 ± 0.34 | 56.50 ± 0.01 | 67.00 | 67.10 | 6.88 |
| | FEATURE-MAP | 82.48 ± 3.25 | 87.70 ± 0.64 | 69.56 ± 0.20 | 67.23 ± 1.93 | 71.49 ± 0.13 | 54.43 ± 0.03 | **74.12 ± 0.09** | 58.73 ± 0.04 | 70.72 | 70.60 | 4.38 |
| | BSS | 72.42 ± 0.03 | 82.92 ± 1.60 | 62.76 ± 4.23 | 72.81 ± 0.66 | 65.79 ± 5.31 | 52.89 ± 1.12 | 71.91 ± 0.44 | 57.79 ± 1.80 | 67.41 | 67.25 | 6.25 |

Table 2: Robust fine-tuning performance on 4 regression datasets (RMSE metrics) in the Non-Fewshot setting, evaluated across 3 dataset splits (RANDOM, SCAFFOLD, SIZE) and 2 pre-training strategies (SELF-SUPERVISED, SUPERVISED). AVG-R, AVG-R* denote the average rank and the rank based on the average normalized performance over all the datasets for each evaluated method, respectively. Standard deviations across five replicates are shown in parentheses. We **bold** and underline the best and second-best performances in each scenario.

| SPLIT | METHODS | SELF-SUPERVISED PRE-TRAINING (MOLE-BERT) | | | | | | SUPERVISED PRE-TRAINING (GRAPHIUM) | | | | | |
|---|---|---|---|---|---|---|---|---|---|---|---|---|---|
| | | ESOL | LIPO | MALARIA | CEP | AVG-R | AVG-R* | ESOL | LIPO | MALARIA | CEP | AVG-R | AVG-R* |
| RANDOM | FULL-FT | 0.852 ± 0.014 | 0.652 ± 0.006 | 1.076 ± 0.007 | **1.394 ± 0.030** | 2.25 | 2 | 0.752 ± 0.022 | 0.634 ± 0.018 | 1.098 ± 0.010 | 1.449 ± 0.017 | 4.50 | 5 |
| | LP | 1.147 ± 0.015 | 0.889 ± 0.002 | 1.154 ± 0.001 | 2.008 ± 0.001 | 8.00 | 8 | 0.972 ± 0.000 | 0.882 ± 0.000 | 1.166 ± 0.000 | 1.834 ± 0.000 | 7.50 | 8 |
| | SURGICAL-FT | 0.929 ± 0.014 | 0.707 ± 0.010 | 1.088 ± 0.003 | 1.614 ± 0.006 | 5.25 | 6 | 0.668 ± 0.000 | 0.635 ± 0.000 | **1.044 ± 0.000** | 1.607 ± 0.000 | 4.00 | 3 |
| | LP-FT | 0.839 ± 0.017 | 0.658 ± 0.009 | 1.080 ± 0.008 | 1.413 ± 0.017 | 3.00 | 3 | 0.715 ± 0.011 | 0.637 ± 0.016 | 1.082 ± 0.014 | **1.389 ± 0.018** | 3.75 | 2 |
| | WISE-FT | 0.973 ± 0.012 | 0.691 ± 0.016 | **1.051 ± 0.005** | 1.507 ± 0.022 | 4.00 | 4 | 0.707 ± 0.025 | 0.620 ± 0.017 | 1.095 ± 0.010 | 1.512 ± 0.041 | 4.00 | 4 |
| | $L^2$-SP | **0.835 ± 0.023** | 0.672 ± 0.004 | 1.091 ± 0.013 | 1.634 ± 0.009 | 4.25 | 5 | 0.653 ± 0.022 | 0.670 ± 0.017 | 1.261 ± 0.004 | 1.605 ± 0.029 | 5.75 | 7 |
| | FEATURE-MAP | 1.039 ± 0.014 | 0.832 ± 0.005 | 1.130 ± 0.001 | 1.820 ± 0.004 | 7.00 | 7 | **0.647 ± 0.018** | **0.605 ± 0.016** | 1.064 ± 0.011 | 1.451 ± 0.012 | 2.00 | 1 |
| | BSS | 0.854 ± 0.013 | **0.640 ± 0.006** | 1.057 ± 0.009 | 1.406 ± 0.012 | 2.25 | 1 | 0.652 ± 0.023 | 0.662 ± 0.016 | 1.271 ± 0.003 | 1.437 ± 0.035 | 4.50 | 6 |
| SCAFFOLD | FULL-FT | 1.126 ± 0.014 | 0.728 ± 0.011 | 1.152 ± 0.015 | **1.377 ± 0.015** | 3.75 | 5 | 0.911 ± 0.041 | 0.709 ± 0.009 | 1.110 ± 0.009 | 1.419 ± 0.014 | 4.00 | 4 |
| | LP | 1.614 ± 0.010 | 0.870 ± 0.003 | 1.110 ± 0.002 | 2.006 ± 0.002 | 7.00 | 8 | 0.973 ± 0.000 | 0.881 ± 0.000 | 1.105 ± 0.000 | 1.826 ± 0.000 | 6.75 | 8 |
| | SURGICAL-FT | 1.166 ± 0.017 | 0.783 ± 0.003 | 1.120 ± 0.014 | 1.601 ± 0.006 | 5.25 | 6 | 0.892 ± 0.000 | 0.709 ± 0.000 | 1.105 ± 0.000 | 1.419 ± 0.000 | 3.50 | 2 |
| | LP-FT | **1.070 ± 0.021** | 0.730 ± 0.002 | 1.144 ± 0.022 | 1.397 ± 0.013 | 3.50 | 4 | 0.922 ± 0.004 | 0.735 ± 0.019 | 1.080 ± 0.005 | **1.368 ± 0.037** | 4.00 | 3 |
| | WISE-FT | 1.264 ± 0.055 | 0.768 ± 0.010 | **1.072 ± 0.001** | 1.470 ± 0.029 | 4.00 | 2 | **0.888 ± 0.014** | 0.708 ± 0.008 | 1.128 ± 0.021 | 1.490 ± 0.024 | 3.75 | 6 |
| | $L^2$-SP | 1.099 ± 0.030 | 0.742 ± 0.008 | 1.101 ± 0.001 | 1.631 ± 0.006 | 3.75 | 3 | 0.948 ± 0.022 | 0.729 ± 0.015 | 1.141 ± 0.015 | 1.606 ± 0.013 | 7.00 | 7 |
| | FEATURE-MAP | 1.403 ± 0.012 | 0.842 ± 0.004 | 1.083 ± 0.002 | 1.787 ± 0.003 | 5.75 | 7 | 0.895 ± 0.016 | **0.688 ± 0.018** | **1.074 ± 0.004** | 1.472 ± 0.010 | 2.50 | 1 |
| | BSS | 1.110 ± 0.022 | **0.726 ± 0.004** | 1.125 ± 0.018 | 1.385 ± 0.018 | 3.00 | 1 | 0.896 ± 0.018 | 0.718 ± 0.018 | 1.130 ± 0.005 | 1.408 ± 0.039 | 4.50 | 5 |
| SIZE | FULL-FT | 1.419 ± 0.044 | 0.745 ± 0.008 | 0.896 ± 0.007 | **1.893 ± 0.035** | 3.25 | 4 | 1.070 ± 0.082 | 0.719 ± 0.010 | 0.886 ± 0.007 | 1.906 ± 0.006 | 4.00 | 4 |
| | LP | 2.073 ± 0.012 | 0.912 ± 0.004 | 0.921 ± 0.008 | 2.381 ± 0.006 | 8.00 | 8 | 1.115 ± 0.000 | 0.829 ± 0.000 | 0.907 ± 0.000 | 2.246 ± 0.000 | 8.00 | 8 |
| | SURGICAL-FT | 1.685 ± 0.060 | 0.775 ± 0.007 | 0.890 ± 0.005 | 2.145 ± 0.022 | 5.00 | 6 | **0.993 ± 0.000** | 0.717 ± 0.000 | **0.860 ± 0.000** | 1.906 ± 0.000 | 2.50 | 1 |
| | LP-FT | 1.440 ± 0.081 | 0.735 ± 0.013 | 0.893 ± 0.007 | 1.905 ± 0.016 | 3.50 | 2 | 1.038 ± 0.038 | 0.694 ± 0.012 | 0.883 ± 0.005 | 1.913 ± 0.031 | 2.75 | 2 |
| | WISE-FT | 1.814 ± 0.092 | 0.831 ± 0.007 | **0.873 ± 0.005** | 1.951 ± 0.024 | 4.50 | 6 | 1.100 ± 0.005 | **0.691 ± 0.015** | 0.894 ± 0.007 | 1.943 ± 0.039 | 4.50 | 6 |
| | $L^2$-SP | 1.438 ± 0.046 | 0.799 ± 0.002 | 0.888 ± 0.005 | 2.101 ± 0.016 | 4.00 | 4 | 1.053 ± 0.026 | 0.720 ± 0.015 | 0.904 ± 0.002 | 2.122 ± 0.018 | 6.00 | 7 |
| | FEATURE-MAP | 1.656 ± 0.025 | 0.880 ± 0.011 | 0.893 ± 0.002 | 2.252 ± 0.008 | 6.25 | 7 | **0.993 ± 0.034** | 0.724 ± 0.009 | 0.884 ± 0.001 | 1.970 ± 0.013 | 4.50 | 3 |
| | BSS | **1.375 ± 0.019** | **0.731 ± 0.007** | 0.887 ± 0.010 | 1.900 ± 0.016 | 1.50 | 1 | 1.043 ± 0.022 | 0.703 ± 0.016 | 0.905 ± 0.005 | **1.890 ± 0.071** | 3.75 | 5 |

Furthermore, we observe that this behavior is more pronounced in regression tasks than in classification tasks. Specifically, full fine-tuning ranks the highest for regression tasks but only achieves mid-tier performance for classification tasks. This disparity likely arises from the distinct nature of these tasks. Classification tasks typically require coarser-grained features, as exemplified by the Tox21 dataset. In this case, determining toxicity may largely rely on recognizing certain functional groups,

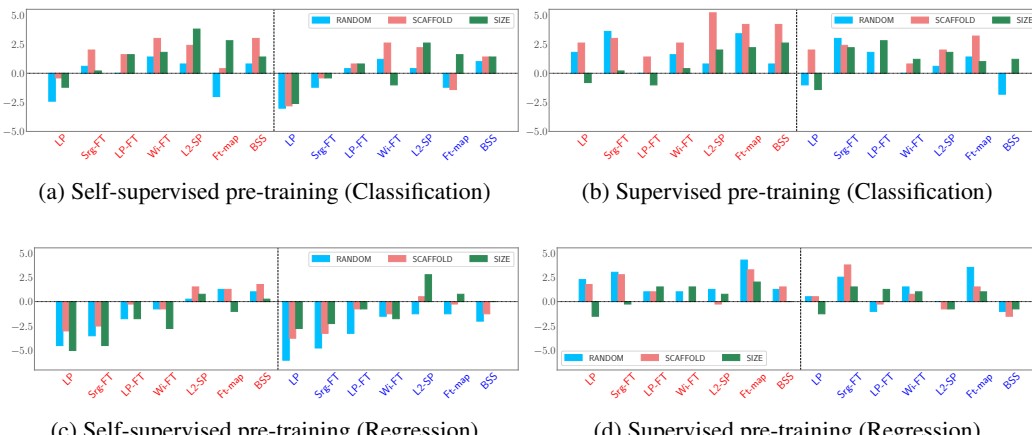

(a) Self-supervised pre-training (Classification)    (b) Supervised pre-training (Classification)

(c) Self-supervised pre-training (Regression)    (d) Supervised pre-training (Regression)

Figure 2: Average Rank improvements over Full-fine-tuning for 7 robust fine-tuning methods in self-supervised and supervised pre-training scenarios across 8 *classification* (*a, b*) datasets and across 4 *regression* (*c, d*) datasets. Each subfigure presents both few-shot-50 (left of the dashed line, colored in red) and few-shot-100 (right of the dashed line, colored in blue) settings, with **random**, **scaffold**, and **size** splits.

such as toxicophores or structural alerts (Singh et al., 2016). In contrast, regression tasks demand finer-grained features. For example, predicting precise solubility involves factors such as partial charge distribution, conformational flexibility, and hydrogen bond patterns, among others (Faller & Ertl, 2007). Consequently, models fine-tuned for regression tasks must acquire more downstream knowledge during the fine-tuning process and are generally less prone to overfitting compared to those used for classification tasks.

Below, we summarize some insightful findings by examining the performance of different fine-tuning strategies and explain the observations in the context of molecular representation learning.

• **Finding 1. Weight-based fine-tuning strategies stand out under few-shot fine-tuning, with WiSE-FT for classification tasks and $L^2$-SP for regression tasks.**

Among various fine-tuning methods, weight-based approaches consistently outperform others across a wide range of experiments, regardless of the few-shot sample sizes (*cf.*, Fig.2a and 2c). Self-supervised models are known to capture general-purpose knowledge for substructure discovery(Wang et al., 2024). During fine-tuning, combining pre-trained and fine-tuned weights proves effective in extracting molecular patterns relevant to downstream tasks. Notably, WiSE-FT demonstrates superior performance on classification datasets, whereas $L^2$-SP excels in regression tasks. WiSE-FT applies a straightforward post-hoc linear interpolation between pre-trained and fine-tuned models, governed by a single coefficient. In contrast, $L^2$-SP implicitly determines the weight combination through the training loss (Lubana et al., 2022; Xuhong et al., 2018), aligning with the idea that regression tasks typically demand more nuanced modeling.

• **Finding 2. Partial fine-tuning results in underfitted molecular representations under few-shot fine-tuning, which is more severe for regression tasks compared to classification.**

For the non-few-shot fine-tuning (*c.f.*, Tables 1 and 2), surgical FT and LP-FT improve over full FT in both classification and regression tasks. However, in few-shot fine-tuning, both methods rank as the worst methods. This is likely because partial fine-tuning underfits and bias towards the the limited samples. This issue is more pronounced in regression tasks.

• **Finding 3. Regulating feature representations brings significant benefits under few-shot fine-tuning but has only a marginal impact in non-few-shot fine-tuning.**

Representation-based methods incorporates additional representation regularization in addition to full FT. BSS aims to eliminate noisy or non-transferable dimensions by regularizing small singular values of representations and Feature-map enforces a close distance of the fine-tuned representations to the pre-trained representations. Since the baseline full FT performs well under non-few-shot settings (*c.f.*, Tables 1 and 2), and pre-trained molecular representations are unsatisfying as discussed in **Q1**, having fine-tuned representations to unsatisfying pre-trained representations does not lead to

any benefits. While under few-shot fine-tuning, representation regularization prevents overfitting with limited samples on top of full FT to some extend.

## 4.2 SUPERVISED PRE-TRAINED MODELS

***Q2: Can supervised pre-training help downstream molecular property prediction tasks?***

We first discuss the **task similarity** between the datasets used in the pre-training and downstream fine-tuning process. As introduced in Sec. 3, the ToyMix dataset used for supervised pre-training contains QM9, Tox21 and Zinc12K. The predictions from QM9 are not directly related to our downstream tasks, but we do not rule out potential indirect correlations, as the quantum chemical properties provided by QM9 are highly valuable for characterizing molecular features. **Tox21** is an overlapping dataset that also exists as one of the downstream datasets. Its tasks in predicting qualitative toxicity measurements are *highly related* to the downstream **ClinTox** and **ToxCast** datasets, and also *correlate* to the **Sider** dataset which contains evaluation in drug side effects. Lastly, Zinc12K, which is to predict the constrained solubility, is relevant to the **Esol** and **Lipo** datasets that involve solubility predictions. Other downstream tasks *do not share* the same tasks with pre-training *directly*.

**(2a) Under few-shot fine-tuning, supervised pre-training models generally yield higher fine-tuning performance compared to self-supervised pre-training, regardless of the task correlations between pre-training and fine-tuning.**

Supervised pre-training brings more benefits to downstream tasks than self-supervised pre-training in few-shot situations when checking Tables 5 and 6. Besides, the benefits are less relevant to the task similarity in contrast to the non-few-shot cases. For example, the improvements are also observed in HIV and Cep datasets even their tasks do not share with pre-training tasks directly.

**(2b) Under non-few-shot fine-tuning, supervised pre-training has better fine-tuning performances than self-supervised pre-training when its objectives align closely with downstream tasks. However, it may hurt downstream performance if the tasks do not align.**

From Tables 1 and 2, we observe consistent fine-tuning performance improvements over self-supervised pre-training on highly task-correlated downstream datasets including ClinTox, Esol, Lipo and Tox21. We can see that even pre-training uses regression tasks and some of the downstream tasks are classification, there is still performance gain if the physical meaning of the tasks are aligned. For datasets that do not directly share tasks with pre-training, we observe mixed performance on Sider, Malaria, and Cep datasets, and even performance declines on HIV and MUV datasets. This finding resonates with the previous work (Sun et al., 2022) to some extend. They concluded that if the supervised pre-training with target labels that are aligned with the downstream tasks, pre-training with pure supervised objective leads to marginal improvement over self-supervised pre-training and adding supervised objective on top of self-supervised pre-training leads to further benefits. The difference is that they pre-trained on single ChEMBL dataset (Gaulton et al., 2012) and did not evaluate for few-shot fine-tuning or on regression datasets.

Below are some detailed findings with different fine-tuning methods given supervised pre-training.

• **Finding 4. Fine-tuning strategies that regularizes towards pre-trained molecular representations rank top, while weight-based methods are suboptimal.**

From both non-few-shot (*c.f.*, Tables 1 and 2) and few-shot fine-tuning (*c.f.*, Fig. 2b and 2d), surgical FT and Feature-map tend to be the top-ranking methods. However, best performing weight-based methods for self-supervised pre-training, only show mediocre performance here. In addition, the other representation-based method BSS show limited performance compared to Feature-map that directly regularize the distance to pre-trained representations. These observations suggest that given the task alignment between supervised pre-training and downstream fine-tuning, pre-trained representations tend to contain transferable features for downsteam tasks. Consequently, controlling the degree to preserve pre-trained representations is the key to downstream fine-tuning performance.

• **Finding 5. LP with pre-trained molecular representations from supervised pre-training surpasses full FT under few-shot fine-tuning, except for size splits.**

For few-shot fine-tuning with 50 and 100 samples (*c.f.*, Fig. 2b and 2d), LP surpasses full FT in random and scaffold splits, differing from self-supervised pre-training discussed in **(1a)**. This again

supports the claim that directly adopting molecular representations from supervised pre-training retain useful knowledge for downstream tasks. But interestingly, this does not hold for size splits. We believe it is due to the susceptibility of graph level tasks under size shift, as noted in prior OOD studies (Zou et al., 2023). Namely, the prediction head tends to overfit to the mapping from representations to output labels with molecules in a specific range of sizes, and thus cannot generalize to OOD molecules of different sizes.

## 5 METHODOLOGY EXPLORATION

Upon investigating the findings in Section 4, we observe that weight-based fine-tuning generally performs well under self-supervised pre-training. However, the top strategy varies: WiSE-FT excels in classification tasks, while $L^2$-SP is more effective for regression tasks. This motivates us to further explore the connections and trade-offs between these methods to identify potential improvements. In this section, we introduce DWiSE-FT, an extension of the weight ensemble method unifying the strengths from WiSE-FT and $L^2$-SP. DWiSE-FT demonstrates top-ranking results through efficient post-processing that better suits the practical fine-tuning needs.

### 5.1 MOTIVATION

As introduced in Sec. 2, WiSE-FT adopts the post-hoc linear interpolation between the pre-trained and fine-tuned model weights as $(1 - \alpha) \cdot \boldsymbol{\theta}_{\text{pre}} + \alpha \cdot \boldsymbol{\theta}_{\text{ft}}$. Although $L^2$-SP does not explicitly have weight interpolation in the form, the optimal weight $\tilde{\boldsymbol{\theta}}_{\text{ft}}$ from the weight-regularized loss $\tilde{\mathcal{L}}(\boldsymbol{\theta})$ is indeed the linear interpolation of the optimal model from full FT $\boldsymbol{\theta}_{\text{ft}}^*$ and the pre-trained model $\boldsymbol{\theta}_{\text{pre}}$.

**Proposition 1.** *Given $\tilde{\mathcal{L}}(\boldsymbol{\theta}) = \mathcal{L}(\boldsymbol{\theta}) + \frac{\delta}{2}\|\boldsymbol{\theta} - \boldsymbol{\theta}_{pre}\|_2^2$, we define the optimal weights as $\tilde{\boldsymbol{\theta}}_{ft} = \arg\min_{\boldsymbol{\theta}} \tilde{\mathcal{L}}(\boldsymbol{\theta})$ and $\boldsymbol{\theta}_{ft}^* = \arg\min_{\boldsymbol{\theta}} \mathcal{L}(\boldsymbol{\theta})$.*

$$\mathbf{Q}^T \tilde{\boldsymbol{\theta}}_{ft} = (\boldsymbol{\Lambda} + \delta\mathbf{I})^{-1} \boldsymbol{\Lambda} \mathbf{Q}^T \boldsymbol{\theta}_{ft}^* + \delta(\boldsymbol{\Lambda} + \delta\mathbf{I})^{-1} \mathbf{Q}^T \boldsymbol{\theta}_{pre} \ . \tag{2}$$

*where $\boldsymbol{H}$ is the hessian matrix of $\mathcal{L}$ evaluated at $\boldsymbol{\theta}_{ft}^*$ and $\boldsymbol{H} = \boldsymbol{Q}\boldsymbol{\Lambda}\boldsymbol{Q}^T$.*

Namely, $L^2$-SP can be seen as a more tailored weight ensemble method, employing variable mixing coefficients for different weights. This approach balances the influence of the prediction loss and the degree of weight regularization, unlike the fixed interpolation controlled by $\alpha$ across all weights in WiSE-FT. By accounting for subtle differences in loss values, $L^2$-SP is better suited for regression tasks, which are more sensitive to numerical variations.

While $L^2$-SP excels on regression datasets, its regularization coefficient is less interpretable and necessitates retraining when experimenting with different values. In contrast, WiSE-FT offers a simpler and more flexible approach, performing post-hoc interpolation without additional training once the model is fine-tuned once. Furthermore, the mixing coefficient $\alpha$ is both easy to adjust and straightforward to interpret. Therefore, our goal is to find a method that benefits from both WiSE-FT and $L^2$-SP to accommodate regression and classification tasks at the same time.

### 5.2 ALGORITHM

We propose DWiSE-FT that shares the framework of using the $\alpha$ to control the weight ensemble between the pre-trained model and fine-tuned model. The key idea, inspired by Eq. 4 is to enable different $\alpha$ values when ensembling the weights for different encoder layers as shown in Fig. 1. Given the pre-trained model with parameters $\boldsymbol{\theta}_{\text{pre}}$ and model after full fine-tuning with parameters $\boldsymbol{\theta}_{\text{ft}}$, The interpolated model has weights $\boldsymbol{\theta}^{[i]}$ with mixing coefficient $\alpha_i$ for the $i$-th layer as:

$$\boldsymbol{\theta}^{[i]} = (1 - \alpha_i) \cdot \boldsymbol{\theta}_{\text{pre}}^{[i]} + \alpha_i \cdot \boldsymbol{\theta}_{\text{ft}}^{[i]} \tag{3}$$

This approach naturally incorporates the characteristics of $L^2$-SP and even surgical FT: The weight ensemble in DWiSE-FT offers the flexibility through varying mixing layer-wise coefficients between the pre-trained and fine-tuned models, addressing the limitations of WiSE-FT. Additionally, we enable the selection of $\boldsymbol{\alpha}$ through optimization via validation loss gradient inspired by the Gradient-based Neural Architecture Search (NAS) (Dong & Yang, 2019).

Table 3: DWiSE-FT performance on 4 regression datasets (RMSE metrics) in the few-shot setting with 50, 100 samples, evaluated across 3 dataset splits (RANDOM, SCAFFOLD, SIZE) given Mole-BERT model. AVG-R denote the average rank. Standard deviations across five replicates are shown in parentheses. We **bold** and underline the best and second-best performances in each scenario.

| | | FEWSHOT 50 | | | | | FEWSHOT 100 | | | | |
| --- | --- | --- | --- | --- | --- | --- | --- | --- | --- | --- | --- |
| SPLIT | METHODS | ESOL | LIPO | MALARIA | CEP | AVG | ESOL | LIPO | MALARIA | CEP | AVG |
| RANDOM | WiSE-FT | 1.384 ± 0.047 | 1.212 ± 0.020 | 1.276 ± 0.007 | 2.410 ± 0.051 | 3.75 | 1.189 ± 0.030 | 1.142 ± 0.025 | **1.256 ± 0.006** | 2.211 ± 0.028 | 3.00 |
| | $L^2$-SP | 1.372 ± 0.029 | 1.196 ± 0.019 | 1.277 ± 0.006 | 2.280 ± 0.031 | 3.00 | 1.161 ± 0.016 | 1.149 ± 0.007 | 1.260 ± 0.004 | 2.131 ± 0.014 | 3.25 |
| | TOP | **1.329 ± 0.021** | **1.164 ± 0.010** | **1.271 ± 0.007** | 2.275 ± 0.022 | 1.25 | **1.120 ± 0.038** | 1.139 ± 0.017 | **1.256 ± 0.006** | 2.131 ± 0.014 | 1.50 |
| | DWiSE-FT | 1.378 ± 0.055 | 1.189 ± 0.020 | 1.273 ± 0.009 | **2.222 ± 0.059** | 2.00 | 1.132 ± 0.025 | **1.138 ± 0.028** | **1.256 ± 0.004** | **2.129 ± 0.020** | 1.25 |
| SCAFFOLD | WiSE-FT | 1.842 ± 0.056 | 1.177 ± 0.009 | 1.162 ± 0.004 | 2.454 ± 0.043 | 3.50 | 1.544 ± 0.063 | 1.041 ± 0.017 | 1.151 ± 0.007 | 2.301 ± 0.042 | 3.50 |
| | $L^2$-SP | 1.699 ± 0.049 | 1.086 ± 0.009 | 1.162 ± 0.002 | 2.331 ± 0.024 | 2.50 | 1.473 ± 0.009 | 0.961 ± 0.003 | 1.153 ± 0.002 | 2.201 ± 0.038 | 2.50 |
| | TOP | 1.680 ± 0.042 | **1.036 ± 0.007** | **1.159 ± 0.000** | 2.292 ± 0.026 | 1.25 | 1.436 ± 0.054 | **0.937 ± 0.008** | 1.149 ± 0.003 | 2.187 ± 0.034 | 1.25 |
| | DWiSE-FT | **1.616 ± 0.047** | 1.110 ± 0.013 | 1.173 ± 0.005 | 2.306 ± 0.030 | 2.50 | 1.485 ± 0.041 | 0.979 ± 0.014 | 1.158 ± 0.009 | **2.149 ± 0.040** | 2.75 |
| SIZE | WiSE-FT | 2.615 ± 0.072 | 1.391 ± 0.042 | 0.929 ± 0.004 | 2.762 ± 0.053 | 4.00 | 2.216 ± 0.056 | 1.124 ± 0.031 | 0.917 ± 0.004 | 2.543 ± 0.027 | 3.75 |
| | $L^2$-SP | 2.393 ± 0.068 | 1.306 ± 0.037 | 0.915 ± 0.002 | **2.497 ± 0.019** | 2.50 | 1.731 ± 0.071 | 1.025 ± 0.028 | 0.905 ± 0.002 | 2.424 ± 0.024 | 1.75 |
| | TOP | 2.369 ± 0.075 | 1.297 ± 0.040 | **0.911 ± 0.002** | **2.497 ± 0.019** | 1.50 | 1.731 ± 0.071 | 1.025 ± 0.028 | **0.898 ± 0.003** | 2.424 ± 0.024 | 1.50 |
| | DWiSE-FT | **1.488 ± 0.101** | **1.113 ± 0.021** | 0.913 ± 0.007 | 2.539 ± 0.023 | 1.75 | **1.469 ± 0.052** | 1.031 ± 0.022 | 0.920 ± 0.006 | **2.390 ± 0.025** | 2.25 |

## 5.3 EXPERIMENT RESULTS

Regarding the classification datasets, DWiSE-FT should have the performance at least as good as WiSE-FT since WiSE-FT is the special case of DWiSE-FT with one fixed mixing coefficient. We evaluate DWiSE-FT to see how it improves upon WiSE-FT and matches the superior performance of $L^2$-SP for regression tasks under few-shot fine-tuning. Please note that, due to space constraints, we only present the experiments for few-shot fine-tuning with 50 and 100 samples in the main text. The complete table is available in Appendix E, Table 10. In Table 3, we compare DWiSE-FT's performance against WiSE-FT, $L^2$-SP, and the best-performing method in each setting. Specifically, we find that DWiSE-FT consistently outperforms WiSE-FT. Furthermore, DWiSE-FT often surpasses $L^2$-SP or at least maintains comparable results in most scenarios. Additionally, in some cases, DWiSE-FT even exceeds the performance of the best-performing methods. Therefore, DWiSE-FT can be a great candidate for fine-tuning on regression datasets in practice since it guarantees top performance with easier usage.

## 6 CONCLUSION

This work benchmarks totally 8 fine-tuning methods, categorizing them into three groups, and evaluate them across 12 downstream datasets under 36 different experimental settings covering 3 dataset splits, 4 training sample sizes, and 3 molecular pre-trained models. The design of these settings reflects practical demands of molecular representation fine-tuning under 1) diversified foundation model with both supervised and self-supervised pre-training, 2) wide range of downstream tasks in both classification and regression that has not been widely studied by previous literature and 3) scarcely labeled molecules for fine-tuning. The study analyzes what is needed when facing classification vs. regression tasks and when given supervised vs. self-supervised pre-training. Then, we provide insights in best performing fine-tuning methods accordingly under aforementioned scenarios. Additionally, we propose an extended fine-tuning method DWiSE-FT, driven by our observations, that maintains top-ranking results through a more efficient and automated design for certain fine-tuning scenarios. This highlights the value of our benchmark in offering valuable insights for both fine-tuning methodology design and practical guidance in molecular representation learning.

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

## A    PROOF OF PROPOSITION 1

**Proposition 2.** *Given* $\tilde{\mathcal{L}}(\boldsymbol{\theta}) = \mathcal{L}(\boldsymbol{\theta}) + \frac{\delta}{2}\|\boldsymbol{\theta} - \boldsymbol{\theta}_{pre}\|_2^2$, *we define the optimal weights as* $\tilde{\boldsymbol{\theta}}_{ft} = \operatorname{argmin}_{\boldsymbol{\theta}}\tilde{\mathcal{L}}(\boldsymbol{\theta})$ *and* $\boldsymbol{\theta}_{ft}^* = \operatorname{argmin}_{\boldsymbol{\theta}}\mathcal{L}(\boldsymbol{\theta})$.

$$\mathbf{Q}^T\tilde{\boldsymbol{\theta}}_{ft} = (\boldsymbol{\Lambda} + \delta\mathbf{I})^{-1}\boldsymbol{\Lambda}\mathbf{Q}^T\boldsymbol{\theta}_{ft}^* + \delta(\boldsymbol{\Lambda} + \delta\mathbf{I})^{-1}\mathbf{Q}^T\boldsymbol{\theta}_{pre} \ . \tag{4}$$

*where $H$ is the hessian matrix of $\mathcal{L}$ evaluated at $\boldsymbol{\theta}_{ft}^*$ and $\boldsymbol{H} = \boldsymbol{Q}\boldsymbol{\Lambda}\boldsymbol{Q}^T$.*

*Proof.* Based on the quadratic approximation, we can approximate $\mathcal{L}(\boldsymbol{\theta})$ as follows:

$$\mathcal{L}(\boldsymbol{\theta}) = \mathcal{L}(\boldsymbol{\theta}_{\text{ft}}^*) + \mathcal{L}'(\boldsymbol{\theta}_{\text{ft}}^*)(\boldsymbol{\theta} - \boldsymbol{\theta}_{\text{ft}}^*) + \frac{1}{2}(\boldsymbol{\theta} - \boldsymbol{\theta}_{\text{ft}}^*)^T\boldsymbol{H}(\boldsymbol{\theta} - \boldsymbol{\theta}_{\text{ft}}^*)$$

$$= \mathcal{L}(\boldsymbol{\theta}_{\text{ft}}^*) + \frac{1}{2}(\boldsymbol{\theta} - \boldsymbol{\theta}_{\text{ft}}^*)^T\boldsymbol{H}(\boldsymbol{\theta} - \boldsymbol{\theta}_{\text{ft}}^*)$$

since $\mathcal{L}'(\boldsymbol{\theta}_{\text{ft}}^*) = 0$ as $\boldsymbol{\theta}_{\text{ft}}^*$ is the minimum. Then, we add the weight regularization term, such that

$$\tilde{\mathcal{L}}(\boldsymbol{\theta}) = \mathcal{L}(\boldsymbol{\theta}_{\text{ft}}^*) + \frac{1}{2}(\boldsymbol{\theta} - \boldsymbol{\theta}_{\text{ft}}^*)^T\boldsymbol{H}(\boldsymbol{\theta} - \boldsymbol{\theta}_{\text{ft}}^*) + \delta\|\boldsymbol{\theta}_{\text{ft}} - \boldsymbol{\theta}_{\text{pre}}\|_2^2$$

Then, we solve for $\tilde{\boldsymbol{\theta}}_{\text{ft}}$ by setting $\nabla\tilde{\mathcal{L}}(\boldsymbol{\theta}) = 0$

$$\boldsymbol{H}(\tilde{\boldsymbol{\theta}}_{\text{ft}} - \boldsymbol{\theta}_{\text{ft}}^*) + \delta(\tilde{\boldsymbol{\theta}}_{\text{ft}} - \boldsymbol{\theta}_{\text{pre}}) = 0$$

$$(\boldsymbol{H} + \delta\boldsymbol{I})\tilde{\boldsymbol{\theta}}_{\text{ft}} = \boldsymbol{H}\boldsymbol{\theta}_{\text{ft}}^* + \delta\boldsymbol{\theta}_{\text{pre}}$$

$$\tilde{\boldsymbol{\theta}}_{\text{ft}} = (\boldsymbol{H} + \delta\boldsymbol{I})^{-1}(\boldsymbol{H}\boldsymbol{\theta}_{\text{ft}}^* + \delta\boldsymbol{\theta}_{\text{pre}})$$

$$\tilde{\boldsymbol{\theta}}_{\text{ft}} = (\boldsymbol{Q}\boldsymbol{\Lambda}\boldsymbol{Q}^T + \delta\boldsymbol{I})^{-1}(\boldsymbol{Q}\boldsymbol{\Lambda}\boldsymbol{Q}^T\boldsymbol{\theta}_{\text{ft}}^* + \delta\boldsymbol{\theta}_{\text{pre}})$$

$$\tilde{\boldsymbol{\theta}}_{\text{ft}} = (\boldsymbol{Q}(\boldsymbol{\Lambda} + \delta\boldsymbol{I})\boldsymbol{Q}^T)^{-1}(\boldsymbol{Q}\boldsymbol{\Lambda}\boldsymbol{Q}^T\boldsymbol{\theta}_{\text{ft}}^* + \delta\boldsymbol{\theta}_{\text{pre}})$$

$$\mathbf{Q}^T\tilde{\boldsymbol{\theta}}_{\text{ft}} = (\boldsymbol{\Lambda} + \delta\mathbf{I})^{-1}\boldsymbol{\Lambda}\mathbf{Q}^T\boldsymbol{\theta}_{\text{ft}}^* + \delta(\boldsymbol{\Lambda} + \delta\mathbf{I})^{-1}\mathbf{Q}^T\boldsymbol{\theta}_{\text{pre}}$$

$\square$

## B    LIMITATIONS AND FUTURE WORKS

We acknowledge certain limitations in this current work and highlight potential improvements for future research. Firstly, this study primarily focuses on the *property prediction tasks* of *small molecules* using *2D-graph* based foundation models. Exploring a broader array of foundation models across a wider range of applications–such as covering more areas like DNA, proteins, and materials, addressing various scientific tasks like linker design and chemical reactions, and incorporating diverse data formats like 3D geometric data–is highly worthwhile. Secondly, although we attempt to include many representative fine-tuning methods from various categories in this study, additional fine-tuning methods from different categories, as discussed in Appendix C, deserve investigation. For instance, future research could explore whether graph-specific fine-tuning methods offer additional benefits over non-graph fine-tuning approaches across various settings we design. Thirdly, the method DWiSE-FT introduced here is an extension and combination of existing methods directly motivated by our benchmark findings for specific fine-tuning scenarios. Future work may involve more thorough exploration into fine-tuning methodology design inspired by our current findings, and aiming to develop approaches effective across a broader range of fine-tuning scenarios.

## C    ADDITIONAL DISCUSSIONS OF RELATED WORKS

In this section, we additionally discuss more related works about fine-tuning (FT) techniques. Designing advanced fine-tuning strategies first gained attention in the computer vision (CV) and natural language processing (NLP) domains, leading to the development of various research directions. We categorize the mainstream approaches into the following groups.

**Partial model FT.** Numerous studies demonstrate that freezing certain parameters while fine-tuning only specific components of the pre-trained model can help mitigate overfitting during the fine-tuning process (Kirkpatrick et al., 2017; Lee et al., 2019; Ramasesh et al., 2020; Eastwood et al., 2021; Evci et al., 2022; Cohen et al., 2022). Specifically, Linear Probing (LP) only trains the additional prediction head during FT. Surgical FT (Lee et al., 2022) selectively fine-tunes a subset of layers based on the specific mechanism of distribution shifts.

**Weight-based FT** strategies mainly control the model weights during the FT. Specifically, WiSE-FT (Wortsman et al., 2022), grounded on the linear mode connectivity (Frankle et al., 2020), linearly interpolates between pre-training parameters and fine-tuning parameters by a mixing coefficient. $L^2$-SP (Xuhong et al., 2018) regularizes the fine-tuning model weights using $L^2$ distance to constrain the parameters around pre-trained ones. REGSL (Li & Zhang, 2021) further introduces a layer-wise parameter regularization, where the constraint strength gradually reduces from the top to bottom layers. MARS-SP (Gouk et al., 2020) adopts the projected gradient method (PGM) to constrain the fine-tuning model weights within a small sphere centered on the pre-trained ones. More recently, TPGM (Tian et al., 2023) further incorporates trainable weight projection radii constraint for each layer, inspired by MARS-SP, to support layer-wise regularization optimization.

**Representation-based FT** methods mainly regulate the latent representation space during FT. Feature-map (Li et al., 2019b) adds distance regularization between the latent representations of pre-trained and fine-tuned models to the Full-FT loss. DELTA (Li et al., 2019a) specifically constrains feature maps with the pre-trained activations selected by channel-wise attention. BSS (Chen et al., 2019) penalizes the spectral components corresponding to small singular values that are less transferable to prevent negative transfer. Li et al. (2020b) proposes to transfer representations by encouraging small deviations from the reference one through an regularizer based on optimal transport. Inspired by this, GTOT-Tuning (Zhang et al., 2022) presents optimal transport-based fine-tuning framework. LP-FT (Kumar et al., 2022) first performs LP to prediction head while keeping the pre-trained encoder fixed, followed by applying full-FT with the tuned prediction head.

**Architecture Refinement.** Besides the weight and representation based FT, StochNorm (Kou et al., 2020) refactors the widely used Batch Normalization (BN) module and proposes Stochastic Normalization, to transfer more pre-trained knowledge during the fine-tuning process and mitigate overfitting.

**Contrastive-based FT.** As discussed in Sec. 2, contrastive-based strategies have been widely demonstrated to be effective in the pre-training stage. There are other works which explore its effectiveness in the fine-tuning process. Gunel et al. (2020), Bi-tuning (Zhong et al., 2020), Core-tuning (Zhang et al., 2021a) and COIN (Pan et al., 2023) introduce supervised contrastive learning (Khosla et al., 2020) to better leverage the label information in the target datasets with more discriminative representations as a result. More recently, FLYP (Goyal et al., 2023) shows that simply finetuning a classifier via the same contrastive loss as pre-training leads to superior performance in finetuning image-text models. Oh et al. (2024) fine-tunes the model with contrastive loss on additional hard negative samples, which are generated by geodesic multi-modal Mixup, for robust fine-tuning in multi-modal models.

**Graph-specific fine-tuning techniques.** Apart from the CV and NLP domains, several fine-tuning techniques specifically designed for the Graph-ML domain have recently been proposed. GTOT-Tuning (Zhang et al., 2022) achieves efficient knowledge transfer from the pre-trained models by an optimal transport-based FT framework. Bridge-Tune (Huang et al., 2024) introduces an intermediate step that bridges pre-training and downstream tasks by considering the task similarity between them. G-tuning (Sun et al., 2024) tunes the pre-trained GNN so that it can reconstruct the generative patterns (graphons) of the downstream graphs. Li et al. (2024) leverages expressive adapters for GNNs, to boost adaptation to the downstream tasks.

## D    DATASET STATISTICS

The statistics of the downstream datasets included in this work are shown in Table 4.

Table 4: Summary for the molecular datasets used for downstream FT, where "# TASKS" and "# MOLECULES" denote the number of tasks and molecules of each dataset, respectively.

| DATASET | EVALUATION METRICS | TASK | # TASKS | # MOLECULES |
|---|---|---|---|---|
| BBBP | AUC | CLASSIFICATION | 1 | 2,039 |
| TOX21 | AUC | CLASSIFICATION | 12 | 7,831 |
| TOXCAST | AUC | CLASSIFICATION | 617 | 8,576 |
| SIDER | AUC | CLASSIFICATION | 27 | 1,427 |
| CLINTOX | AUC | CLASSIFICATION | 2 | 1,478 |
| MUV | AUC | CLASSIFICATION | 17 | 93,087 |
| HIV | AUC | CLASSIFICATION | 1 | 41,127 |
| BACE | AUC | CLASSIFICATION | 1 | 1,513 |
| ESOL | RMSE | REGRESSION | 1 | 1,128 |
| LIPO | RMSE | REGRESSION | 1 | 4,200 |
| MALARIA | RMSE | REGRESSION | 1 | 9,999 |
| CEP | RMSE | REGRESSION | 1 | 29,978 |

## E   DETAILS OF EXPERIMENTAL IMPLEMENTATION

**Pre-training Implementations.** For self-supervised pre-training, we use the open-source pre-trained checkpoints of Mole-BERT[1] and GraphMAE[2]. For supervised pre-training, we follow the same training pipeline as proposed in the Graphium[3]. We drop out the task head MLPs used for supervised pre-training during the downstream fine-tuning process, keeping only the graph encoder component. Note that we keep the architecture of the GNN encoder and the graph pooling strategy the same across the three pre-training models. Specifically, we use a 5-layer Graph Isomorphism Networks (GINs) with 300 hidden dimension and mean pooling as the readout function.

**Fine-tuning Implementations.** We keep the same training configurations across all the downstream datasets, pre-training models, and fine-tuning strategies, following Hu et al. (2020a). Specifically, for each distinct setting, we fine-tune the pre-training models with 5 random seeds (0-4). We use a batch size of 32 and a dropout rate of 0.5. For each dataset, We train models for 100 epochs and report the test performance when the optimal validation performance is achieved.

**Hyperparameter Tuning.** We set learning rate to be 0.001 for all the methods and train for 100 epochs. Below is the detailed sets of hyperparameters we tuned for each fine-tuning strategy.

- *Surgical FT:* We tune $k$ as which layer in GNN encoder to be updated from $\{0, 1, 2, 3, 4\}$ since our backbone architecture is a 5-layer GIN.

- *WiSE-FT:* We tune the mixing coefficient $\alpha$ from $\{0.1, 0.2, 0.3, 0.4, 0.5, 0.6, 0.7, 0.8, 0.9\}$ to control the weight ensemble from pre-trained model and fine-tuned model. A larger $\alpha$ indicates the weights are adopted more from the fine-tuned model.

- *$L^2$-SP/ BSS/ Feature-map:* For these three methods that involve an additional regularization term in the loss, we tune the regularization coefficient $\delta$ from $\{1, 0.1, 0.01, 0.001, 0.0001\}$ to control the degree of regularization. For BSS, we follow the original paper and set $k$ to be 1 meaning that we are regularizing the smallest singular value.

- *LP-FT:* We train the LP step before full fine-tuning for 100 epochs and then use the updated prediction head as initilization for the full-FT afterwards for 100 epochs. The training all use the default learning rate 0.001.

- *Full FT/ LP:* There is no additional hyperparameter tuning, where we use the default fine-tuning setting.

- *DWiSE-FT:* We tune the initialization of $\alpha_i$ for each layer $i$, where we use the same value to initialize for all layers from $\{0.9, 0.7, 0.5\}$ and the learning rate for validation loss descent from $\{0.001, 0.005, 0.01\}$. We tune $\alpha$ over validation sets over 200 epochs.

---

[1] https://github.com/junxia97/Mole-BERT
[2] https://github.com/THUDM/GraphMAE
[3] https://github.com/datamol-io/graphium

Table 5: Robust fine-tuning performance on 5 classification datasets (AUC metrics) in the Fewshot setting (covering FEWSHOT-50, FEWSHOT-100, FEWSHOT-500), evaluated across 3 dataset splits (RANDOM, SCAFFOLD, SIZE) and 2 pre-training strategies (SELF-SUPERVISED, SUPERVISED). We **bold** and underline the best and second-best performances in each scenario.

| SPLIT | METHODS | SELF-SUPERVISED PRE-TRAINING (MOLE-BERT) | | | | | | | | SUPERVISED PRE-TRAINING (GRAPHIUM) | | | | | | | |
|---|---|---|---|---|---|---|---|---|---|---|---|---|---|---|---|---|---|
| | | CLINTOX | BBBP | BACE | HIV | SIDER | AVG | AVG-F | AVG-R | CLINTOX | BBBP | BACE | HIV | SIDER | AVG | AVG-F | AVG-R |
| **FEWSHOT-50** | | | | | | | | | | | | | | | | | |
| RANDOM | FULL-FT | 74.45±2.10 | 88.56±0.83 | 75.80±0.43 | 57.41±0.69 | 52.22±0.48 | 69.69 | 69.22 | 4.40 | 70.14±0.52 | 77.57±0.01 | 80.45±0.00 | 63.57±0.00 | 55.57±0.00 | 69.46 | 70.43 | 6.00 |
| | LP | 77.50±1.31 | 82.05±0.37 | 75.04±0.58 | 53.34±2.39 | 51.40±0.11 | 67.87 | 68.63 | 6.80 | 84.09±0.00 | 81.04±0.00 | 81.57±0.00 | 49.05±0.00 | 55.62±0.00 | 70.27 | 72.74 | 4.20 |
| | SURGICAL-FT | 77.91±1.25 | 85.41±0.66 | 75.94±0.40 | 57.90±0.40 | 51.99±0.18 | 69.83 | 70.58 | 3.80 | 77.64±0.00 | 84.99±0.00 | 81.93±0.00 | 64.72±0.00 | 56.40±0.00 | 73.14 | 74.76 | 2.40 |
| | LP-FT | 77.66±0.74 | 88.99±0.14 | 75.18±0.48 | 57.38±0.37 | 51.68±0.16 | 70.18 | 70.07 | 4.40 | 69.84±0.00 | 80.15±0.00 | 78.64±0.00 | 65.82±0.00 | 53.56±0.00 | 69.60 | 71.43 | 6.00 |
| | WISE-FT | 76.12±3.87 | 88.72±1.05 | 75.59±0.51 | 58.59±0.77 | 52.23±0.50 | 70.25 | 70.12 | 3.00 | 81.94±0.03 | 83.74±0.00 | 78.47±0.00 | 63.17±0.00 | 56.44±0.00 | 72.75 | 74.53 | 4.40 |
| | L2-SP | 76.27±1.05 | 88.50±1.25 | 75.17±0.90 | 59.09±1.33 | 52.27±0.32 | 70.26 | 70.18 | 3.60 | 72.26±1.46 | 81.07±0.13 | 79.75±0.50 | 63.68±0.92 | 55.48±0.00 | 70.45 | 71.90 | 5.20 |
| | FEATURE-MAP | 74.43±2.07 | 88.40±0.84 | 73.84±0.66 | 57.93±1.13 | 51.82±0.31 | 69.28 | 68.73 | 6.40 | 84.80±0.129 | 85.33±0.021 | 81.53±0.194 | 60.64±0.016 | 56.49±0.005 | 73.76 | 75.66 | 2.60 |
| | BSS | 75.31±3.21 | 88.69±0.54 | 75.50±0.38 | 59.19±1.58 | 52.13±0.37 | 70.16 | 70.00 | 3.60 | 74.14±2.15 | 77.94±0.35 | 78.82±1.14 | 64.45±1.10 | 55.57±0.05 | 70.18 | 72.18 | 5.20 |
| SCAFFOLD | FULL-FT | 60.18±1.70 | 59.68±1.79 | 68.88±2.31 | 55.47±6.57 | 53.12±0.45 | 59.47 | 58.44 | 6.00 | 62.14±0.00 | 76.51±0.54 | 63.74±0.00 | 54.02±0.00 | 55.88±0.00 | 63.67 | 62.61 | 7.40 |
| | LP | 60.36±0.84 | 57.58±0.82 | 70.25±1.28 | 57.45±5.76 | 51.76±0.37 | 59.48 | 58.46 | 6.40 | 79.10±0.00 | 57.74±0.00 | 76.54±0.00 | 65.43±0.00 | 55.88±0.00 | 66.50 | 66.57 | 4.80 |
| | SURGICAL-FT | 60.80±1.05 | 60.86±0.98 | 71.16±0.84 | 58.60±6.33 | 52.24±0.21 | 60.73 | 60.09 | 4.00 | 71.30±0.00 | 63.24±0.00 | 76.34±0.00 | 66.81±0.00 | 56.56±0.00 | 66.85 | 67.12 | 4.40 |
| | LP-FT | 59.59±1.11 | 60.36±1.20 | 71.57±0.37 | 56.18±2.07 | 53.31±0.29 | 60.20 | 58.71 | 4.40 | 65.30±0.00 | 63.16±0.00 | 77.15±0.00 | 66.60±0.00 | 53.65±0.00 | 65.17 | 65.02 | 6.00 |
| | WISE-FT | 67.60±3.67 | 60.51±1.64 | 72.25±1.25 | 63.65±2.09 | 50.66±0.93 | 62.93 | 63.92 | 3.00 | 67.34±0.00 | 65.55±0.00 | 78.66±0.00 | 65.28±0.00 | 55.17±0.00 | 66.40 | 66.06 | 4.80 |
| | L2-SP | 61.76±1.22 | 59.53±2.09 | 70.81±0.79 | 64.76±2.40 | 52.95±0.45 | 61.96 | 62.02 | 3.60 | 83.15±0.03 | 66.76±0.00 | 78.75±0.74 | 68.22±0.02 | 55.86±0.00 | 70.55 | 71.24 | 2.20 |
| | FEATURE-MAP | 61.30±1.94 | 55.91±2.04 | 65.37±0.99 | 61.18±2.35 | 52.64±1.03 | 59.28 | 59.46 | 5.60 | 77.49±0.04 | 67.13±0.01 | 78.57±0.03 | 64.39±0.01 | 56.74±0.00 | 68.86 | 69.67 | 3.20 |
| | BSS | 67.94±2.58 | 60.40±2.18 | 70.51±1.82 | 60.39±2.23 | 53.18±0.46 | 62.48 | 62.91 | 3.00 | 69.74±0.02 | 65.64±0.00 | 79.10±0.00 | 68.47±0.01 | 54.97±0.03 | 67.58 | 67.95 | 3.20 |
| SIZE | FULL-FT | 66.75±0.92 | 80.03±0.54 | 43.23±1.52 | 62.00±3.04 | 47.81±0.77 | 59.96 | 58.85 | 5.80 | 67.61±0.01 | 71.89±5.76 | 48.57±0.01 | 52.54±0.00 | 53.48±0.00 | 58.82 | 57.88 | 5.20 |
| | LP | 69.17±0.41 | 78.19±0.32 | 39.81±0.34 | 48.97±1.66 | 46.13±0.24 | 56.45 | 54.76 | 7.00 | 71.21±0.01 | 57.79±0.00 | 40.44±0.01 | 48.13±0.00 | 55.62±0.00 | 54.64 | 53.85 | 6.00 |
| | SURGICAL-FT | 68.76±0.63 | 82.19±0.86 | 42.26±2.37 | 56.73±1.32 | 46.77±0.14 | 59.34 | 57.42 | 5.60 | 71.70±0.01 | 68.21±0.00 | 46.06±0.01 | 53.09±0.00 | 54.86±0.00 | 58.78 | 58.72 | 5.00 |
| | LP-FT | 69.43±0.30 | 82.00±0.83 | 42.83±1.39 | 61.12±1.15 | 48.77±0.32 | 60.83 | 59.77 | 4.20 | 68.90±0.01 | 65.03±0.01 | 47.57±0.00 | 47.28±0.00 | 54.15±0.00 | 56.59 | 55.58 | 6.20 |
| | WISE-FT | 70.76±1.31 | 81.92±3.19 | 65.58±2.49 | 56.58±10.19 | 47.24±0.57 | 64.42 | 64.31 | 4.00 | 72.03±0.01 | 70.14±5.65 | 45.24±0.01 | 53.43±0.00 | 53.59±0.00 | 58.89 | 59.05 | 4.80 |
| | L2-SP | 69.09±1.06 | 83.98±1.98 | 52.70±4.51 | 63.68±3.16 | 50.80±2.97 | 64.05 | 61.82 | 2.00 | 72.95±0.73 | 63.38±5.27 | 63.46±3.90 | 66.83±0.03 | 54.89±0.01 | 64.30 | 64.56 | 3.20 |
| | FEATURE-MAP | 67.57±1.45 | 82.52±0.74 | 51.61±1.25 | 66.37±3.56 | 49.65±0.57 | 63.54 | 61.85 | 3.00 | 76.65±0.06 | 71.39±0.05 | 65.20±0.01 | 57.29±0.43 | 53.01±0.01 | 64.71 | 64.63 | 3.00 |
| | BSS | 67.65±1.32 | 80.29±3.12 | 50.73±6.35 | 62.56±2.53 | 49.05±0.64 | 62.06 | 60.31 | 4.40 | 72.26±0.16 | 68.79±6.08 | 55.61±0.00 | 55.40±0.01 | 63.43 | 63.79 | 2.60 | |
| **FEWSHOT-100** | | | | | | | | | | | | | | | | | |
| RANDOM | FULL-FT | 78.70±5.25 | 86.87±0.80 | 79.91±0.70 | 60.88±1.37 | 53.88±0.69 | 72.05 | 73.16 | 4.20 | 69.31±1.27 | 82.85±0.00 | 83.76±0.44 | 64.82±2.36 | 56.88±0.00 | 71.52 | 72.33 | 5.00 |
| | LP | 79.45±0.85 | 84.18±0.62 | 73.16±0.46 | 51.26±1.30 | 52.78±0.31 | 68.17 | 68.46 | 7.20 | 81.85±0.00 | 80.80±0.00 | 79.25±0.00 | 51.60±0.00 | 57.78±0.00 | 70.26 | 72.61 | 6.00 |
| | SURGICAL-FT | 81.54±1.62 | 85.66±0.52 | 77.00±0.74 | 59.34±0.42 | 53.63±0.44 | 71.43 | 72.63 | 5.40 | 75.51±0.00 | 86.37±0.00 | 84.51±0.00 | 66.28±0.00 | 58.87±0.00 | 74.31 | 75.43 | 2.00 |
| | LP-FT | 79.86±1.12 | 87.26±0.81 | 78.86±0.48 | 59.37±0.51 | 54.31±0.32 | 71.93 | 72.70 | 3.80 | 81.73±0.32 | 83.54±0.02 | 81.91±0.00 | 65.46±0.62 | 58.74±0.00 | 74.28 | 76.37 | 3.20 |
| | WISE-FT | 85.55±1.43 | 86.76±0.42 | 74.53±0.97 | 61.90±1.36 | 56.41±0.69 | 73.03 | 73.90 | 3.00 | 71.90±1.49 | 83.18±0.83 | 83.63±0.95 | 63.80±0.36 | 57.66±0.00 | 72.03 | 72.96 | 5.00 |
| | L2-SP | 79.13±3.68 | 86.89±0.40 | 79.66±0.35 | 59.92±1.04 | 54.64±0.35 | 72.05 | 72.90 | 3.80 | 76.28±0.02 | 81.15±1.52 | 80.71±1.44 | 64.00±0.98 | 59.02±0.54 | 72.23 | 73.66 | 4.40 |
| | FEATURE-MAP | 78.12±3.01 | 87.80±0.62 | 73.50±0.69 | 59.97±0.75 | 53.50±0.24 | 70.58 | 70.53 | 5.40 | 82.51±0.15 | 85.94±0.56 | 82.09±1.02 | 63.34±0.11 | 57.82±0.05 | 74.34 | 75.98 | 3.60 |
| | BSS | 79.00±4.62 | 87.38±0.52 | 80.12±0.33 | 60.22±1.07 | 53.88±0.72 | 72.12 | 73.11 | 3.20 | 72.38±1.42 | 80.11±0.78 | 81.64±0.64 | 63.65±0.65 | 56.85±0.81 | 70.93 | 72.05 | 6.80 |
| SCAFFOLD | FULL-FT | 70.51±70.51 | 62.11±1.32 | 68.39±3.19 | 61.60±1.74 | 52.20±0.26 | 62.96 | 64.03 | 4.40 | 70.75±0.00 | 65.39±0.25 | 77.66±0.30 | 59.73±0.00 | 54.53±0.00 | 65.61 | 65.29 | 5.80 |
| | LP | 60.68±60.68 | 58.10±0.99 | 69.41±1.69 | 57.12±4.63 | 52.11±0.51 | 59.48 | 58.63 | 7.60 | 80.09±0.00 | 53.89±0.00 | 78.39±0.00 | 64.11±0.00 | 56.03±0.00 | 66.50 | 66.18 | 3.80 |
| | SURGICAL-FT | 65.93±65.93 | 61.45±1.01 | 70.20±1.91 | 59.62±0.64 | 52.49±0.67 | 61.94 | 62.33 | 5.20 | 75.08±0.00 | 64.49±0.00 | 78.42±0.00 | 67.41±0.00 | 54.87±0.00 | 68.05 | 68.99 | 3.40 |
| | LP-FT | 66.18±2.14 | 61.52±0.91 | 71.48±0.58 | 60.76±1.04 | 53.68±0.46 | 62.72 | 62.82 | 4.00 | 67.42±0.00 | 66.33±0.00 | 74.91±0.44 | 64.40±0.00 | 53.25±0.00 | 65.26 | 66.05 | 5.80 |
| | WISE-FT | 64.71±2.82 | 62.88±2.30 | 75.95±1.63 | 62.67±2.42 | 54.27±0.82 | 64.10 | 63.42 | 2.20 | 74.35±0.00 | 64.90±0.06 | 78.06±0.96 | 62.56±0.00 | 54.55±0.00 | 66.88 | 67.27 | 5.00 |
| | L2-SP | 70.98±2.49 | 61.93±2.03 | 72.49±0.86 | 66.43±0.76 | 52.51±0.93 | 64.87 | 66.45 | 2.60 | 74.06±0.20 | 66.14±0.00 | 77.15±0.00 | 72.98±1.69 | 54.82±0.78 | 69.03 | 71.06 | 3.80 |
| | FEATURE-MAP | 63.83±1.60 | 58.74±1.66 | 67.61±0.30 | 58.27±3.68 | 53.97±1.51 | 60.49 | 60.29 | 6.20 | 79.79±0.36 | 63.60±0.03 | 78.91±0.38 | 56.33±0.43 | 56.33±0.63 | 65.41 | 65.01 | 5.80 |
| | BSS | 70.99±1.94 | 62.47±0.62 | 69.47±2.49 | 62.09±0.93 | 52.22±0.33 | 63.45 | 64.68 | 3.40 | 68.24±1.75 | 65.35±0.00 | 78.31±0.01 | 61.43±0.16 | 53.73±0.45 | 65.41 | 65.01 | 5.80 |
| SIZE | FULL-FT | 72.17±2.23 | 80.54±1.53 | 59.53±0.71 | 61.90±2.19 | 48.97±0.30 | 64.62 | 64.53 | 4.80 | 73.66±0.01 | 81.77±0.00 | 60.31±1.27 | 59.36±1.43 | 54.37±0.00 | 65.89 | 64.44 | 5.60 |
| | LP | 68.13±0.43 | 81.53±0.52 | 49.67±2.12 | 46.66±3.40 | 47.08±0.22 | 58.61 | 54.96 | 7.40 | 72.12±0.01 | 52.13±0.00 | 47.81±0.07 | 47.18±0.49 | 55.11±0.00 | 54.87 | 51.68 | 7.00 |
| | SURGICAL-FT | 70.80±0.56 | 83.61±0.40 | 58.55±3.14 | 55.86±1.29 | 47.75±0.49 | 63.31 | 61.74 | 5.20 | 78.60±0.01 | 80.76±0.00 | 56.62±0.01 | 66.14±0.00 | 55.12±0.00 | 67.45 | 67.12 | 3.80 |
| | LP-FT | 68.05±0.12 | 83.02±0.40 | 59.92±1.08 | 60.87±1.57 | 50.40±0.29 | 64.57 | 62.95 | 4.00 | 76.90±2.09 | 85.29±0.00 | 66.72±0.02 | 51.80±0.00 | 56.61±0.00 | 67.46 | 66.74 | 2.80 |
| | WISE-FT | 71.91±1.19 | 81.89±5.23 | 55.66±2.06 | 53.27±8.19 | 48.26±0.31 | 62.20 | 60.28 | 5.80 | 73.22±0.01 | 82.39±0.00 | 62.81±1.46 | 61.23±0.03 | 54.99±0.00 | 66.93 | 65.75 | 4.40 |
| | L2-SP | 73.25±1.91 | 83.39±0.71 | 60.46±1.08 | 63.14±2.17 | 50.74±2.54 | 66.20 | 65.62 | 2.20 | 72.06±0.04 | 82.89±0.00 | 66.17±0.04 | 54.76±0.88 | 54.99±0.78 | 71.85 | 3.80 | |
| | FEATURE-MAP | 69.78±2.65 | 83.55±1.25 | 62.51±1.38 | 57.64±3.25 | 51.26±0.38 | 64.95 | 63.31 | 3.20 | 76.90±0.04 | 76.51±0.06 | 61.49±3.16 | 62.51±1.43 | 54.57±0.09 | 66.40 | 66.84 | 4.60 |
| | BSS | 73.74±2.81 | 80.91±1.12 | 60.12±1.15 | 63.05±2.33 | 50.20±0.94 | 65.60 | 65.04 | 3.40 | 78.11±1.47 | 73.92±0.09 | 64.84±0.40 | 48.42±0.08 | 53.54±1.60 | 67.77 | 69.06 | 4.40 |
| **FEWSHOT-500** | | | | | | | | | | | | | | | | | |
| RANDOM | FULL-FT | 86.07±1.80 | 92.76±0.54 | 85.99±0.40 | 67.49±0.86 | 61.33±0.24 | 78.73 | 79.85 | 3.40 | 88.53±1.79 | 91.44±1.06 | 83.72±0.59 | 70.25±1.76 | 58.51±0.00 | 78.49 | 80.83 | 4.20 |
| | LP | 84.85±0.40 | 87.91±0.20 | 73.59±0.24 | 55.25±0.21 | 59.54±0.14 | 72.23 | 72.66 | 7.60 | 91.56±0.00 | 85.15±0.00 | 83.18±0.00 | 66.82±0.00 | 58.78±0.00 | 77.10 | 78.38 | 4.20 |
| | SURGICAL-FT | 87.77±0.56 | 92.14±0.57 | 84.09±0.45 | 67.76±0.31 | 59.66±0.22 | 78.28 | 79.87 | 4.40 | 91.31±0.00 | 92.11±0.00 | 84.49±0.00 | 69.71±0.00 | 59.93±0.00 | 79.51 | 81.84 | 2.40 |
| | LP-FT | 85.55±0.75 | 92.20±0.29 | 85.79±0.37 | 68.44±0.80 | 61.06±0.55 | 78.61 | 79.93 | 3.60 | 88.82±1.84 | 91.07±0.99 | 83.89±0.00 | 66.62±0.69 | 57.89±0.00 | 77.66 | 79.78 | 5.20 |
| | WISE-FT | 87.70±1.47 | 91.02±0.72 | 85.36±0.44 | 62.00±2.20 | 64.11±0.55 | 78.04 | 79.06 | 4.00 | 89.75±1.06 | 92.30±0.39 | 83.58±0.00 | 66.27±2.15 | 58.65±0.00 | 78.11 | 79.87 | 4.20 |
| | L2-SP | 85.46±1.06 | 92.44±0.82 | 85.11±0.32 | 68.42±0.77 | 59.37±0.56 | 78.16 | 79.66 | 5.00 | 85.29±1.89 | 82.38±1.17 | 80.83±0.91 | 66.64±1.36 | 57.95±0.76 | 74.62 | 76.62 | 6.60 |
| | FEATURE-MAP | 83.42±3.42 | 90.57±0.49 | 76.69±0.41 | 48.24±0.93 | 51.80±0.36 | 75.71 | 76.12 | 6.40 | 91.58±0.23 | 83.40±0.00 | 85.29±0.81 | 72.78±0.13 | 60.19±0.04 | 80.33 | 83.22 | 1.40 |
| | BSS | 86.17±1.34 | 92.76±0.38 | 86.04±0.32 | 69.34±0.40 | 61.45±0.51 | 79.15 | 80.52 | 1.60 | 82.20±1.72 | 81.21±1.30 | 83.13±1.36 | 64.65±1.05 | 57.16±0.83 | 73.67 | 76.02 | 7.80 |
| SCAFFOLD | FULL-FT | 69.18±2.51 | 69.56±0.99 | 79.14±0.95 | 69.86±1.35 | 56.92±0.20 | 68.93 | 69.53 | 4.20 | 77.16±1.95 | 67.79±0.50 | 74.30±3.48 | 64.63±2.67 | 57.97±0.00 | 68.37 | 68.91 | 6.00 |
| | LP | 61.91±0.52 | 64.03±0.55 | 77.67±0.10 | 66.13±1.48 | 59.60±0.30 | 65.87 | 64.02 | 6.60 | 81.39±0.00 | 65.24±0.00 | 80.66±0.00 | 67.92±0.00 | 58.93±0.00 | 70.83 | 71.27 | 4.20 |
| | SURGICAL-FT | 66.75±0.43 | 67.11±0.90 | 80.66±0.43 | 72.20±0.63 | 58.92±0.38 | 69.13 | 68.69 | 4.00 | 80.56±0.00 | 70.47±0.00 | 80.77±0.00 | 72.03±0.00 | 54.85±0.00 | 71.74 | 74.35 | 3.80 |
| | LP-FT | 69.91±1.83 | 68.58±0.18 | 78.46±0.74 | 69.38±0.59 | 58.07±0.20 | 68.88 | 69.29 | 4.20 | 85.20±1.39 | 68.48±0.55 | 77.44±0.32 | 66.97±0.52 | 54.41±0.00 | 70.50 | 71.35 | 4.40 |
| | WISE-FT | 68.66±1.86 | 64.82±1.71 | 82.01±0.60 | 72.95±0.97 | 60.35±1.11 | 69.76 | 68.81 | 3.20 | 80.66±1.12 | 68.94±0.8 | 80.28±0.18 | 64.84±3.83 | 57.45±0.02 | 70.39 | 71.35 | 4.40 |
| | L2-SP | 69.22±2.59 | 68.11±0.95 | 77.74±1.08 | 73.06±0.43 | 58.86±0.63 | 69.40 | 70.13 | 3.80 | 71.73±4.37 | 67.66±0.75 | 77.77±0.03 | 69.70±0.04 | 56.84±1.27 | 68.74 | 69.70 | 6.00 |
| | FEATURE-MAP | 66.14±1.79 | 64.83±2.23 | 72.50±0.52 | 71.49±1.13 | 59.56±0.29 | 67.49 | 66.90 | 5.60 | 83.65±0.24 | 70.95±0.40 | 82.56±0.05 | 59.58±0.07 | 73.97 | 75.53 | 1.40 | |
| | BSS | 69.65±1.86 | 69.04±0.33 | 78.20±1.39 | 70.85±0.75 | 56.75±0.46 | 68.90 | 69.85 | 4.40 | 74.20±5.33 | 66.12±1.31 | 78.40±1.52 | 73.95±0.94 | 57.05±0.91 | 69.94 | 71.42 | 5.00 |
| SIZE | FULL-FT | 74.96±1.19 | 87.81±1.32 | 54.53±1.81 | 65.86±0.67 | 51.45±0.67 | 66.85 | 65.12 | 3.60 | 82.67±0.65 | 59.41±0.01 | 71.78±4.10 | 53.99±0.00 | 67.63 | 67.17 | 5.00 | |
| | LP | 67.80±0.62 | 82.24±0.47 | 48.77±0.42 | 52.20±3.32 | 50.51±0.31 | 60.30 | 56.84 | 7.20 | 75.60±0.01 | 75.14±0.00 | 50.85±0.10 | 58.39±0.00 | 54.81±0.00 | 62.96 | 62.78 | 6.20 |
| | SURGICAL-FT | 70.35±0.30 | 88.56±0.70 | 60.12±1.38 | 61.09±0.81 | 51.85±0.40 | 66.39 | 63.85 | 3.60 | 77.94±0.01 | 88.47±0.00 | 52.64±0.01 | 54.82±0.00 | 58.78 | 58.74 | 5.20 | |
| | LP-FT | 71.38±0.64 | 86.43±0.68 | 53.50±1.98 | 65.30±0.73 | 49.99±0.30 | 65.32 | 63.99 | 4.20 | 75.59±1.96 | 85.72±1.98 | 49.10±3.22 | 71.61±4.67 | 55.43±0.00 | 67.05 | 67.54 | 4.20 |
| | WISE-FT | 73.53±1.46 | 86.56±1.25 | 65.74±1.37 | 51.55±9.46 | 48.62±0.38 | 65.20 | 63.61 | 5.20 | 68.48±2.42 | 85.26±1.99 | 48.52±0.83 | 75.23±1.71 | 55.22±0.00 | 66.54 | 66.31 | 4.20 |
| | L2-SP | 73.43±1.31 | 86.82±1.64 | 56.73±3.41 | 67.80±1.83 | 51.01±0.60 | 67.16 | 65.99 | 3.80 | 74.24±5.74 | 78.60±2.29 | 59.94±0.02 | 73.61±1.82 | 55.14±1.49 | 68.31 | 69.26 | 3.60 |
| | FEATURE-MAP | 76.06±0.62 | 81.83±0.64 | 58.42±0.90 | 67.94±1.41 | 50.84±0.30 | 67.02 | 67.47 | 3.60 | 80.69±0.11 | 88.49±0.80 | 58.95±0.13 | 67.62±2.74 | 54.76±0.09 | 70.10 | 69.89 | 4.00 |
| | BSS | 74.26±1.07 | 88.06±0.96 | 56.71±1.82 | 66.29±1.10 | 52.91±0.65 | 67.65 | 65.75 | 2.80 | 68.01±0.70 | 79.45±2.68 | 59.39±6.07 | 71.78±1.54 | 54.88±1.50 | 66.70 | 66.39 | 4.80 |

Indeed, from the DWiSE-FT experiments with different starting points of mixing coefficients, the variance of final results is small since it will converge towards the optimal value of mixing coefficients regardless of the initial starting point given a reasonable training time.

# F ADDITIONAL EXPERIMENTAL RESULTS

In this section, we present complementary baseline results that are not shown in the main text due to space limit. Specifically, the results on classification tasks in the Fewshot settings over the Mole-BERT (self-supervised pre-training) and Graphium (supervised pre-training) models are in Table 5. The results on regression tasks in the Fewshot settings over the Mole-BERT and Graphium models are in Table 6. The results on classification tasks in the Non-Fewshot setting over the Graph-MAE (self-supervised pre-training) model are in Table 7. The results on classification tasks in the Fewshot settings over the Graph-MAE model are in Table 8. The results on regression tasks over the Graph-MAE model, including both Non-Fewshot and Fewshot settings, are in Table 9.

The results of classification datasets over the MoleculeSTM model are in Tables 11-14. The results of regression datasets over the MoleculeSTM model are in Tables 15-18.

The complete table including all few-shot fine-tuning results for DWiSE-FT are in Table 10.

Table 6: Robust fine-tuning performance on 4 regression datasets (RMSE metrics) in the Fewshot setting (covering FEWSHOT-50, FEWSHOT-100, and FEWSHOT-500), evaluated across 3 dataset splits (RANDOM, SCAFFOLD, SIZE) and 2 pre-training strategies (SELF-SUPERVISED, SUPERVISED). AVG-R, AVG-R$^*$ denote the average rank and the rank based on the average normalized performance over all the datasets for each evavluated method, respectively. Standard deviations across five replicates are shown in parentheses. We **bold** and underline the best and second-best performances in each scenario.

| SPLIT | METHODS | SELF-SUPERVISED Pre-training (MOLE-BERT) | | | | | | SUPERVISED Pre-training (GRAPHIUM) | | | | | |
|---|---|---|---|---|---|---|---|---|---|---|---|---|---|
| | | ESOL | LIPO | MALARIA | CEP | AVG-R | AVG-R$^*$ | ESOL | LIPO | MALARIA | CEP | AVG-R | AVG-R$^*$ |
| | | | | | | FEWSHOT-50 | | | | | | | |
| RANDOM | FULL-FT | 1.390 ± 0.051 | 1.189 ± 0.016 | 1.276 ± 0.006 | 2.383 ± 0.046 | 3.50 | 4 | 1.223 ± 0.000 | 1.062 ± 0.000 | 1.284 ± 0.000 | 2.359 ± 0.000 | 6.25 | 7 |
| | LP | 2.654 ± 0.016 | 1.825 ± 0.011 | 1.296 ± 0.005 | 3.736 ± 0.020 | 8.00 | 8 | **1.085 ± 0.000** | 1.072 ± 0.000 | **1.272 ± 0.000** | 2.571 ± 0.000 | 4.00 | 3 |
| | SURGICAL-FT | 2.647 ± 0.022 | 1.618 ± 0.014 | 1.295 ± 0.004 | 3.596 ± 0.037 | 7.00 | 7 | 1.174 ± 0.000 | **1.009 ± 0.000** | 1.277 ± 0.000 | 2.355 ± 0.000 | 3.25 | 2 |
| | LP-FT | 1.422 ± 0.027 | 1.237 ± 0.027 | 1.291 ± 0.005 | 2.296 ± 0.012 | 5.25 | 6 | 1.386 ± 0.000 | 1.019 ± 0.000 | 1.286 ± 0.000 | 2.287 ± 0.000 | 5.25 | 8 |
| | WiSE-FT | 1.384 ± 0.047 | 1.212 ± 0.020 | 1.276 ± 0.007 | 2.410 ± 0.051 | 4.25 | 5 | 1.219 ± 0.000 | 1.060 ± 0.000 | 1.280 ± 0.000 | 2.366 ± 0.000 | 5.25 | 4 |
| | L2-SP | 1.372 ± 0.029 | 1.196 ± 0.019 | 1.277 ± 0.006 | 2.280 ± 0.031 | 3.25 | 3 | 1.147 ± 0.026 | 1.092 ± 0.001 | 1.283 ± 0.000 | 2.312 ± 0.020 | 5.00 | 5 |
| | FEATURE-MAP | 1.329 ± 0.021 | 1.164 ± 0.010 | 1.271 ± 0.007 | 2.448 ± 0.010 | 2.25 | 1 | 1.089 ± 0.001 | 1.046 ± 0.000 | 1.276 ± 0.000 | **2.191 ± 0.017** | 2.00 | 1 |
| | BSS | 1.365 ± 0.028 | 1.186 ± 0.017 | 1.277 ± 0.006 | **2.275 ± 0.022** | 2.50 | 2 | 1.175 ± 0.011 | 1.128 ± 0.035 | 1.281 ± 0.000 | 2.262 ± 0.064 | 5.00 | 6 |
| SCAFFOLD | FULL-FT | 1.696 ± 0.058 | 1.124 ± 0.006 | 1.178 ± 0.005 | 2.356 ± 0.033 | 4.25 | 5 | 1.353 ± 0.000 | 1.071 ± 0.000 | 1.168 ± 0.000 | 2.001 ± 0.000 | 5.75 | 8 |
| | LP | 3.754 ± 0.020 | 1.858 ± 0.005 | 1.167 ± 0.002 | 3.849 ± 0.009 | 7.25 | 8 | 1.226 ± 0.000 | 1.013 ± 0.000 | 1.166 ± 0.000 | 2.450 ± 0.000 | 4.00 | 6 |
| | SURGICAL-FT | 3.599 ± 0.039 | 1.843 ± 0.006 | 1.167 ± 0.003 | 3.819 ± 0.017 | 6.75 | 7 | 1.239 ± 0.000 | 1.019 ± 0.000 | **1.162 ± 0.000** | 2.083 ± 0.000 | 3.00 | 2 |
| | LP-FT | 1.822 ± 0.014 | 1.134 ± 0.012 | 1.184 ± 0.004 | **2.292 ± 0.026** | 4.50 | 6 | 1.283 ± 0.000 | 1.033 ± 0.000 | 1.169 ± 0.000 | **1.949 ± 0.000** | 4.75 | 5 |
| | WiSE-FT | 1.842 ± 0.056 | 1.177 ± 0.009 | 1.162 ± 0.004 | 2.454 ± 0.043 | 5.00 | 4 | 1.320 ± 0.000 | 1.071 ± 0.000 | 1.168 ± 0.000 | 1.992 ± 0.000 | 5.75 | 7 |
| | L2-SP | 1.699 ± 0.049 | 1.086 ± 0.009 | 1.162 ± 0.002 | 2.331 ± 0.024 | 2.75 | 2 | 1.273 ± 0.047 | 1.015 ± 0.007 | 1.166 ± 0.000 | 2.132 ± 0.048 | 6.00 | 4 |
| | FEATURE-MAP | 1.823 ± 0.028 | 1.036 ± 0.007 | 1.159 ± 0.000 | 2.425 ± 0.012 | 3.00 | 1 | 1.213 ± 0.001 | 0.991 ± 0.000 | 1.164 ± 0.000 | 2.128 ± 0.006 | 2.50 | 1 |
| | BSS | 1.680 ± 0.042 | 1.114 ± 0.008 | 1.165 ± 0.001 | 2.319 ± 0.025 | 2.50 | 3 | 1.222 ± 0.012 | 1.039 ± 0.000 | 1.166 ± 0.000 | 2.121 ± 0.012 | 4.25 | 3 |
| SIZE | FULL-FT | 2.382 ± 0.079 | **1.297 ± 0.040** | 0.929 ± 0.004 | 2.656 ± 0.039 | 2.75 | 4 | 1.441 ± 0.000 | 1.055 ± 0.000 | 0.914 ± 0.000 | 2.329 ± 0.000 | 5.00 | 7 |
| | LP | 4.534 ± 0.021 | 2.157 ± 0.012 | 0.941 ± 0.004 | 4.706 ± 0.022 | 7.75 | 8 | 1.443 ± 0.000 | 1.003 ± 0.000 | 0.936 ± 0.000 | 2.688 ± 0.000 | 6.50 | 8 |
| | SURGICAL-FT | 4.344 ± 0.026 | 2.111 ± 0.021 | 0.943 ± 0.004 | 4.265 ± 0.028 | 7.25 | 7 | 1.469 ± 0.000 | 1.015 ± 0.000 | 0.914 ± 0.000 | 2.313 ± 0.000 | 5.25 | 5 |
| | LP-FT | 2.421 ± 0.060 | 1.395 ± 0.018 | 0.939 ± 0.007 | 2.525 ± 0.013 | 4.50 | 6 | **1.395 ± 0.000** | 0.999 ± 0.000 | 0.907 ± 0.000 | 2.410 ± 0.000 | 3.50 | 1 |
| | WiSE-FT | 2.615 ± 0.072 | 1.391 ± 0.042 | 0.929 ± 0.004 | 2.762 ± 0.053 | 5.50 | 5 | 1.411 ± 0.000 | 1.071 ± 0.000 | **0.905 ± 0.000** | 2.324 ± 0.000 | 3.50 | 4 |
| | L2-SP | 2.393 ± 0.068 | 1.306 ± 0.032 | 0.915 ± 0.002 | **2.497 ± 0.019** | 2.00 | 2 | 1.446 ± 0.055 | 0.997 ± 0.000 | 0.908 ± 0.000 | 2.340 ± 0.020 | 4.25 | 3 |
| | FEATURE-MAP | 2.422 ± 0.021 | 1.327 ± 0.022 | **0.911 ± 0.002** | 2.659 ± 0.021 | 3.75 | 1 | 1.415 ± 0.005 | **0.989 ± 0.027** | 0.921 ± 0.002 | **2.254 ± 0.001** | 3.00 | 2 |
| | BSS | **2.369 ± 0.075** | 1.319 ± 0.050 | 0.925 ± 0.003 | 2.563 ± 0.022 | 5.00 | 3 | 1.499 ± 0.028 | 0.997 ± 0.000 | 0.907 ± 0.000 | 2.381 ± 0.006 | 5.00 | 6 |
| | | | | | | FEWSHOT-100 | | | | | | | |
| RANDOM | FULL-FT | 1.141 ± 0.030 | 1.141 ± 0.023 | **1.256 ± 0.006** | 2.150 ± 0.021 | 2.00 | 2 | 1.191 ± 0.000 | 1.103 ± 0.000 | 1.258 ± 0.000 | 2.076 ± 0.015 | 5.25 | 4 |
| | LP | 2.273 ± 0.029 | 1.569 ± 0.008 | 1.280 ± 0.003 | 3.235 ± 0.019 | 8.00 | 8 | 1.066 ± 0.000 | 1.045 ± 0.000 | 1.267 ± 0.000 | 2.383 ± 0.000 | 4.75 | 5 |
| | SURGICAL-FT | 1.953 ± 0.039 | 1.281 ± 0.020 | 1.270 ± 0.006 | 3.019 ± 0.047 | 6.75 | 7 | 1.075 ± 0.000 | 1.030 ± 0.000 | 1.266 ± 0.000 | **1.935 ± 0.000** | 2.75 | 2 |
| | LP-FT | 1.244 ± 0.057 | 1.147 ± 0.018 | 1.277 ± 0.003 | 2.156 ± 0.019 | 5.25 | 6 | 1.689 ± 0.000 | 1.097 ± 0.000 | 1.273 ± 0.000 | 2.044 ± 0.015 | 6.25 | 8 |
| | WiSE-FT | 1.189 ± 0.030 | 1.142 ± 0.025 | **1.256 ± 0.006** | 2.211 ± 0.028 | 3.50 | 2 | 1.131 ± 0.000 | 1.078 ± 0.000 | **1.256 ± 0.000** | 2.001 ± 0.071 | 3.75 | 3 |
| | L2-SP | 1.161 ± 0.016 | 1.149 ± 0.007 | 1.260 ± 0.004 | **2.131 ± 0.014** | 3.25 | 4 | 1.098 ± 0.012 | 1.077 ± 0.001 | 1.270 ± 0.001 | 2.261 ± 0.008 | 5.25 | 6 |
| | FEATURE-MAP | 1.120 ± 0.038 | 1.139 ± 0.017 | 1.266 ± 0.004 | 2.283 ± 0.011 | 3.25 | 5 | 0.995 ± 0.018 | 1.025 ± 0.000 | 1.258 ± 0.003 | 1.937 ± 0.023 | 1.75 | 1 |
| | BSS | 1.199 ± 0.033 | 1.149 ± 0.023 | 1.259 ± 0.006 | 2.132 ± 0.019 | 4.00 | 3 | 1.055 ± 0.009 | 1.136 ± 0.000 | 1.274 ± 0.000 | 2.269 ± 0.010 | 6.25 | 7 |
| SCAFFOLD | FULL-FT | **1.436 ± 0.054** | 1.026 ± 0.009 | 1.160 ± 0.011 | 2.198 ± 0.034 | 3.25 | 4 | 1.111 ± 0.000 | 1.037 ± 0.000 | 1.172 ± 0.000 | 1.965 ± 0.023 | 5.00 | 6 |
| | LP | 3.255 ± 0.025 | 1.503 ± 0.008 | 1.154 ± 0.003 | 3.350 ± 0.007 | 7.00 | 8 | 1.228 ± 0.000 | 0.960 ± 0.000 | 1.162 ± 0.000 | 2.423 ± 0.000 | 4.50 | 5 |
| | SURGICAL-FT | 2.587 ± 0.076 | 1.192 ± 0.015 | 1.156 ± 0.003 | 2.914 ± 0.066 | 6.50 | 7 | **1.087 ± 0.000** | 0.966 ± 0.000 | **1.156 ± 0.000** | **1.959 ± 0.000** | 1.25 | 1 |
| | LP-FT | 1.544 ± 0.042 | 1.010 ± 0.011 | 1.163 ± 0.004 | **2.187 ± 0.034** | 4.00 | 6 | 1.111 ± 0.000 | 0.984 ± 0.000 | 1.173 ± 0.000 | 2.149 ± 0.012 | 5.25 | 4 |
| | WiSE-FT | 1.544 ± 0.063 | 1.041 ± 0.017 | **1.151 ± 0.007** | 2.301 ± 0.042 | 4.50 | 3 | 1.110 ± 0.000 | 1.027 ± 0.000 | 1.169 ± 0.000 | 2.013 ± 0.049 | 4.25 | 3 |
| | L2-SP | 1.473 ± 0.009 | 0.961 ± 0.008 | 1.153 ± 0.002 | 2.201 ± 0.038 | 2.75 | 2 | 1.252 ± 0.000 | 0.994 ± 0.013 | 1.163 ± 0.000 | 2.367 ± 0.000 | 5.75 | 7 |
| | FEATURE-MAP | 1.677 ± 0.020 | **0.937 ± 0.008** | 1.149 ± 0.003 | 2.356 ± 0.018 | 3.50 | 1 | 1.158 ± 0.020 | 0.966 ± 0.010 | 1.161 ± 0.000 | 2.024 ± 0.019 | 3.50 | 2 |
| | BSS | 1.463 ± 0.008 | 1.040 ± 0.018 | 1.160 ± 0.006 | 2.210 ± 0.018 | 4.50 | 5 | 1.253 ± 0.027 | 1.033 ± 0.015 | 1.167 ± 0.000 | 2.333 ± 0.022 | 6.50 | 8 |
| SIZE | FULL-FT | 1.889 ± 0.065 | 1.077 ± 0.028 | 0.918 ± 0.005 | **2.425 ± 0.024** | 4.00 | 3 | 1.411 ± 0.000 | **0.962 ± 0.000** | 0.921 ± 0.006 | 2.328 ± 0.015 | 4.75 | 5 |
| | LP | 3.851 ± 0.033 | 1.676 ± 0.025 | 0.911 ± 0.003 | 4.115 ± 0.038 | 6.75 | 8 | 1.253 ± 0.000 | 0.981 ± 0.000 | 0.924 ± 0.000 | 2.635 ± 0.000 | 6.00 | 8 |
| | SURGICAL-FT | 3.237 ± 0.085 | 1.374 ± 0.031 | 0.912 ± 0.002 | 3.174 ± 0.048 | 6.25 | 7 | 1.329 ± 0.000 | 0.965 ± 0.000 | 0.910 ± 0.000 | 2.283 ± 0.000 | 3.25 | 2 |
| | LP-FT | 1.831 ± 0.066 | 1.085 ± 0.014 | 0.920 ± 0.008 | 2.468 ± 0.021 | 4.75 | 4 | **1.242 ± 0.000** | **0.962 ± 0.000** | 0.912 ± 0.000 | 2.375 ± 0.000 | 3.50 | 1 |
| | WiSE-FT | 2.216 ± 0.056 | 1.124 ± 0.031 | 0.917 ± 0.004 | 2.543 ± 0.027 | 5.75 | 5 | 1.398 ± 0.000 | 0.963 ± 0.000 | 0.907 ± 0.002 | 2.319 ± 0.014 | 3.75 | 4 |
| | L2-SP | **1.731 ± 0.071** | 1.025 ± 0.028 | 0.905 ± 0.002 | 2.424 ± 0.024 | 1.25 | 1 | 1.418 ± 0.035 | 0.998 ± 0.038 | **0.906 ± 0.000** | 2.436 ± 0.072 | 5.50 | 6 |
| | FEATURE-MAP | 2.135 ± 0.077 | 1.049 ± 0.013 | **0.898 ± 0.003** | 2.500 ± 0.017 | 3.25 | 2 | 1.335 ± 0.005 | 0.967 ± 0.008 | 0.911 ± 0.001 | **2.265 ± 0.020** | 3.75 | 3 |
| | BSS | 1.734 ± 0.060 | 1.073 ± 0.024 | 0.931 ± 0.008 | 2.439 ± 0.015 | 4.00 | 6 | 1.387 ± 0.039 | 0.998 ± 0.006 | **0.906 ± 0.000** | 2.518 ± 0.137 | 5.50 | 7 |
| | | | | | | FEWSHOT-500 | | | | | | | |
| RANDOM | FULL-FT | 0.883 ± 0.032 | 0.817 ± 0.012 | 1.194 ± 0.003 | 1.891 ± 0.026 | 2.50 | 3 | 0.753 ± 0.000 | 0.842 ± 0.000 | 1.221 ± 0.012 | 1.806 ± 0.005 | 4.75 | 4 |
| | LP | 1.274 ± 0.011 | 1.036 ± 0.004 | 1.216 ± 0.002 | 2.285 ± 0.004 | 8.00 | 8 | 1.007 ± 0.000 | 0.972 ± 0.000 | 1.223 ± 0.000 | 2.117 ± 0.000 | 7.25 | 8 |
| | SURGICAL-FT | 0.961 ± 0.013 | 0.888 ± 0.005 | 1.201 ± 0.005 | 1.962 ± 0.009 | 5.75 | 6 | 0.748 ± 0.000 | **0.825 ± 0.000** | 1.210 ± 0.000 | 1.795 ± 0.000 | 3.00 | 2 |
| | LP-FT | 0.884 ± 0.035 | 0.842 ± 0.013 | 1.215 ± 0.002 | 1.904 ± 0.011 | 4.75 | 5 | **0.697 ± 0.000** | 0.835 ± 0.016 | 1.220 ± 0.008 | 1.794 ± 0.004 | 2.00 | 3 |
| | WiSE-FT | 0.995 ± 0.010 | 0.855 ± 0.011 | 1.193 ± 0.003 | 1.893 ± 0.021 | 4.00 | 4 | 0.742 ± 0.000 | 0.852 ± 0.001 | 1.228 ± 0.004 | 1.809 ± 0.006 | 5.25 | 5 |
| | L2-SP | 0.878 ± 0.026 | **0.806 ± 0.007** | 1.192 ± 0.004 | 1.893 ± 0.018 | 1.75 | 1 | 0.741 ± 0.029 | 0.907 ± 0.020 | 1.243 ± 0.006 | 1.822 ± 0.003 | 6.00 | 7 |
| | FEATURE-MAP | 1.057 ± 0.008 | 0.894 ± 0.009 | 1.196 ± 0.002 | 2.019 ± 0.004 | 6.50 | 7 | 0.706 ± 0.005 | 0.840 ± 0.013 | **1.200 ± 0.014** | 1.773 ± 0.008 | 1.75 | 1 |
| | BSS | 0.886 ± 0.010 | 0.809 ± 0.005 | 1.194 ± 0.006 | **1.862 ± 0.010** | 2.75 | 2 | 0.715 ± 0.024 | 0.892 ± 0.014 | 1.248 ± 0.006 | 1.824 ± 0.006 | 6.00 | 6 |
| SCAFFOLD | FULL-FT | 1.196 ± 0.013 | 0.819 ± 0.009 | 1.137 ± 0.016 | 1.892 ± 0.017 | 4.25 | 4 | 0.956 ± 0.000 | 0.888 ± 0.011 | 1.149 ± 0.014 | **1.787 ± 0.020** | 4.50 | 5 |
| | LP | 1.867 ± 0.006 | 0.937 ± 0.004 | 1.140 ± 0.002 | 2.338 ± 0.005 | 7.75 | 8 | 1.006 ± 0.000 | 0.921 ± 0.000 | 1.162 ± 0.000 | 2.183 ± 0.000 | 8.00 | 8 |
| | SURGICAL-FT | 1.221 ± 0.011 | 0.883 ± 0.010 | 1.130 ± 0.005 | 1.953 ± 0.007 | 5.75 | 6 | 0.955 ± 0.000 | 0.887 ± 0.000 | 1.138 ± 0.000 | **1.787 ± 0.000** | 3.75 | 3 |
| | LP-FT | 1.112 ± 0.015 | **0.802 ± 0.003** | 1.153 ± 0.005 | 1.895 ± 0.013 | 3.50 | 5 | 0.951 ± 0.000 | 0.883 ± 0.025 | 1.143 ± 0.000 | 1.791 ± 0.008 | 3.50 | 4 |
| | WiSE-FT | 1.388 ± 0.023 | 0.834 ± 0.012 | **1.114 ± 0.002** | 1.936 ± 0.037 | 4.25 | 3 | **0.947 ± 0.000** | 0.893 ± 0.007 | 1.134 ± 0.011 | 1.800 ± 0.006 | 4.00 | 2 |
| | L2-SP | 1.163 ± 0.026 | 0.813 ± 0.010 | 1.126 ± 0.011 | **1.885 ± 0.011** | 2.50 | 2 | 0.991 ± 0.018 | 0.878 ± 0.012 | 1.128 ± 0.000 | 2.017 ± 0.179 | 4.50 | 7 |
| | FEATURE-MAP | 1.495 ± 0.016 | 0.863 ± 0.005 | 1.118 ± 0.001 | 2.008 ± 0.010 | 5.50 | 7 | 0.966 ± 0.014 | **0.826 ± 0.017** | 1.136 ± 0.003 | 1.792 ± 0.011 | 3.50 | 1 |
| | BSS | 1.188 ± 0.026 | 0.814 ± 0.007 | 1.123 ± 0.005 | **1.881 ± 0.010** | 2.50 | 1 | 0.977 ± 0.021 | 0.885 ± 0.014 | **1.126 ± 0.007** | 1.949 ± 0.127 | 4.25 | 6 |
| SIZE | FULL-FT | 1.692 ± 0.070 | 0.838 ± 0.023 | 0.922 ± 0.013 | **2.364 ± 0.030** | 4.00 | 3 | 1.115 ± 0.019 | 0.848 ± 0.038 | 0.915 ± 0.000 | 2.230 ± 0.009 | 5.25 | 5 |
| | LP | 2.290 ± 0.017 | 1.039 ± 0.005 | 0.908 ± 0.002 | 2.749 ± 0.018 | 6.75 | 8 | **1.073 ± 0.000** | 0.871 ± 0.000 | 0.904 ± 0.000 | 2.435 ± 0.000 | 5.25 | 8 |
| | SURGICAL-FT | 1.928 ± 0.039 | 0.895 ± 0.007 | 0.919 ± 0.007 | 2.397 ± 0.014 | 5.50 | 6 | 1.094 ± 0.000 | 0.807 ± 0.000 | 0.904 ± 0.000 | **2.200 ± 0.000** | 2.75 | 1 |
| | LP-FT | 1.674 ± 0.030 | **0.803 ± 0.006** | 0.954 ± 0.011 | **2.328 ± 0.017** | 3.25 | 5 | 1.081 ± 0.024 | 0.842 ± 0.021 | 0.925 ± 0.000 | 2.280 ± 0.000 | 5.25 | 7 |
| | WiSE-FT | 2.071 ± 0.078 | 0.902 ± 0.016 | 0.912 ± 0.003 | 2.379 ± 0.086 | 5.75 | 7 | 1.116 ± 0.023 | **0.805 ± 0.015** | 0.907 ± 0.001 | 2.228 ± 0.010 | 4.00 | 2 |
| | L2-SP | **1.629 ± 0.084** | 0.821 ± 0.011 | 0.904 ± 0.003 | 2.368 ± 0.013 | 2.50 | 1 | 1.183 ± 0.055 | 0.853 ± 0.031 | 0.903 ± 0.004 | 2.227 ± 0.038 | 5.00 | 6 |
| | FEATURE-MAP | 1.963 ± 0.035 | 0.910 ± 0.009 | **0.895 ± 0.002** | 2.366 ± 0.006 | 4.25 | 4 | 1.193 ± 0.058 | 0.850 ± 0.021 | 0.901 ± 0.025 | **2.203 ± 0.023** | 4.50 | 4 |
| | BSS | 1.630 ± 0.035 | 0.818 ± 0.005 | 0.925 ± 0.019 | 2.370 ± 0.013 | 4.00 | 2 | 1.142 ± 0.049 | 0.834 ± 0.018 | **0.900 ± 0.003** | 2.245 ± 0.027 | 4.00 | 3 |

Table 7: Robust fine-tuning performance on 8 classification datasets (AUC metrics) in the Non-Fewshot setting, evaluated across 3 dataset splits (RANDOM, SCAFFOLD, SIZE) over the Graph-MAE based pre-trained model. AVG, AVG-F, AVG-R denote the average AUC metrics, average AUC without max and min values, and average rank over all the datasets for each evaluated method, respectively. Standard deviations across five replicates are shown in parentheses. We **bold** and underline the best and second-best performances in each scenario.

| SPLIT | METHODS | CLINTOX | BBBP | BACE | HIV | MUV | SIDER | TOX21 | TOXCAST | AVG | AVG-F | AVG-R |
|---|---|---|---|---|---|---|---|---|---|---|---|---|
| RANDOM | FULL-FT | $83.22 \pm 2.07$ | $94.70 \pm 0.32$ | $89.26 \pm 0.40$ | $85.31 \pm 0.29$ | $80.71 \pm 0.58$ | $61.53 \pm 0.48$ | $82.35 \pm 0.15$ | $73.01 \pm 0.16$ | 81.26 | 82.31 | 4.00 |
| | LP | $78.82 \pm 1.55$ | $83.16 \pm 0.58$ | $77.65 \pm 1.27$ | $74.45 \pm 0.31$ | $78.54 \pm 1.16$ | $61.51 \pm 0.35$ | $73.57 \pm 0.16$ | $66.96 \pm 0.16$ | 74.33 | 75.00 | 7.50 |
| | SURGICAL-FT | $83.85 \pm 1.52$ | $92.11 \pm 0.35$ | $86.77 \pm 0.09$ | $84.56 \pm 0.30$ | $\mathbf{82.71 \pm 0.81}$ | $61.79 \pm 0.19$ | $79.90 \pm 0.14$ | $71.51 \pm 0.21$ | 80.40 | 81.55 | 4.50 |
| | LP-FT | $\mathbf{88.09 \pm 1.04}$ | $94.68 \pm 0.19$ | $\underline{89.58 \pm 0.23}$ | $\mathbf{86.06 \pm 0.43}$ | $80.75 \pm 1.53$ | $61.69 \pm 0.26$ | $\mathbf{82.50 \pm 0.21}$ | $\mathbf{73.66 \pm 0.07}$ | 82.13 | 83.44 | 2.25 |
| | WiSE-FT | $80.01 \pm 4.00$ | $93.04 \pm 0.46$ | $\mathbf{90.15 \pm 0.50}$ | $85.42 \pm 0.52$ | $82.07 \pm 2.10$ | $62.18 \pm 0.49$ | $81.55 \pm 0.43$ | $72.48 \pm 0.26$ | 80.86 | 81.95 | 3.38 |
| | L2-SP | $83.39 \pm 1.88$ | $93.89 \pm 0.28$ | $88.70 \pm 0.10$ | $80.22 \pm 0.17$ | $73.35 \pm 1.54$ | $\mathbf{62.36 \pm 0.43}$ | $77.45 \pm 0.47$ | $68.71 \pm 0.31$ | 78.51 | 78.64 | 5.00 |
| | FEATURE-MAP | $73.08 \pm 0.89$ | $85.36 \pm 0.46$ | $75.88 \pm 0.75$ | $77.04 \pm 0.26$ | $79.53 \pm 1.25$ | $62.06 \pm 0.32$ | $75.36 \pm 0.13$ | $65.69 \pm 0.24$ | 74.25 | 74.43 | 6.75 |
| | BSS | $\underline{83.98 \pm 3.00}$ | $\mathbf{94.85 \pm 0.31}$ | $89.31 \pm 0.21$ | $\underline{86.05 \pm 0.40}$ | $80.55 \pm 0.75$ | $61.92 \pm 0.21$ | $\underline{82.48 \pm 0.28}$ | $\underline{73.22 \pm 0.07}$ | 81.54 | 82.60 | 2.62 |
| SCAFFOLD | FULL-FT | $74.74 \pm 0.93$ | $66.35 \pm 0.65$ | $80.33 \pm 0.37$ | $\underline{77.22 \pm 0.38}$ | $77.47 \pm 1.33$ | $60.98 \pm 0.19$ | $\underline{76.18 \pm 0.31}$ | $64.27 \pm 0.36$ | 72.19 | 72.70 | 3.88 |
| | LP | $71.34 \pm 1.48$ | $64.36 \pm 0.24$ | $61.70 \pm 7.34$ | $70.62 \pm 0.64$ | $\underline{79.13 \pm 1.20}$ | $58.23 \pm 1.29$ | $70.89 \pm 0.10$ | $60.03 \pm 0.13$ | 67.04 | 66.49 | 6.75 |
| | SURGICAL-FT | $71.88 \pm 1.07$ | $66.81 \pm 0.29$ | $80.24 \pm 0.90$ | $76.90 \pm 0.30$ | $\mathbf{79.20 \pm 0.50}$ | $\mathbf{64.00 \pm 0.09}$ | $74.18 \pm 0.40$ | $62.60 \pm 0.27$ | 71.98 | 72.16 | 4.12 |
| | LP-FT | $74.88 \pm 2.00$ | $67.39 \pm 0.55$ | $80.67 \pm 0.57$ | $\mathbf{77.97 \pm 0.38}$ | $75.13 \pm 1.06$ | $60.76 \pm 0.45$ | $\underline{76.18 \pm 0.20}$ | $\mathbf{64.29 \pm 0.23}$ | 72.16 | 72.64 | 3.25 |
| | WiSE-FT | $\mathbf{77.30 \pm 5.30}$ | $\mathbf{69.29 \pm 2.34}$ | $\mathbf{82.16 \pm 1.50}$ | $76.75 \pm 0.69$ | $77.76 \pm 1.55$ | $59.76 \pm 0.86$ | $74.99 \pm 0.44$ | $63.61 \pm 0.34$ | 72.70 | 73.28 | 3.25 |
| | L2-SP | $73.40 \pm 0.45$ | $67.39 \pm 0.90$ | $80.36 \pm 0.92$ | $74.63 \pm 0.44$ | $73.20 \pm 0.90$ | $\underline{63.40 \pm 0.29}$ | $73.16 \pm 0.14$ | $61.29 \pm 0.38$ | 70.85 | 70.86 | 5.00 |
| | FEATURE-MAP | $64.74 \pm 0.62$ | $62.46 \pm 0.26$ | $69.22 \pm 2.06$ | $72.34 \pm 0.58$ | $75.63 \pm 0.54$ | $57.13 \pm 1.08$ | $71.25 \pm 0.13$ | $57.78 \pm 0.26$ | 66.32 | 66.30 | 7.38 |
| | BSS | $\underline{75.80 \pm 1.11}$ | $\underline{67.46 \pm 1.35}$ | $\underline{80.82 \pm 0.62}$ | $77.10 \pm 0.77$ | $78.53 \pm 1.03$ | $62.29 \pm 0.51$ | $\mathbf{76.45 \pm 0.24}$ | $64.03 \pm 0.09$ | 72.81 | 73.23 | 2.38 |
| SIZE | FULL-FT | $56.52 \pm 0.81$ | $80.05 \pm 2.01$ | $59.94 \pm 1.83$ | $77.21 \pm 0.94$ | $74.64 \pm 1.72$ | $53.04 \pm 0.74$ | $70.87 \pm 0.24$ | $60.80 \pm 0.50$ | 66.63 | 66.66 | 4.62 |
| | LP | $57.44 \pm 0.94$ | $73.52 \pm 1.68$ | $51.46 \pm 0.97$ | $73.91 \pm 0.89$ | $65.97 \pm 3.36$ | $51.84 \pm 0.31$ | $67.56 \pm 0.10$ | $57.49 \pm 0.11$ | 62.40 | 62.30 | 7.38 |
| | SURGICAL-FT | $57.47 \pm 1.16$ | $\underline{81.96 \pm 0.78}$ | $55.85 \pm 2.81$ | $\mathbf{80.48 \pm 0.18}$ | $\underline{75.86 \pm 2.96}$ | $54.32 \pm 0.43$ | $71.19 \pm 0.30$ | $59.45 \pm 0.18$ | 67.07 | 66.72 | 3.12 |
| | LP-FT | $56.35 \pm 0.62$ | $76.80 \pm 2.24$ | $61.61 \pm 1.01$ | $77.14 \pm 0.69$ | $\mathbf{79.10 \pm 0.89}$ | $52.69 \pm 0.35$ | $\mathbf{71.33 \pm 0.26}$ | $60.98 \pm 0.27$ | 67.00 | 67.37 | 4.00 |
| | WiSE-FT | $\mathbf{59.25 \pm 3.49}$ | $\mathbf{82.99 \pm 1.91}$ | $61.16 \pm 2.31$ | $75.90 \pm 1.94$ | $75.09 \pm 3.95$ | $\mathbf{55.74 \pm 1.28}$ | $70.94 \pm 0.42$ | $\mathbf{61.53 \pm 0.56}$ | 67.83 | 67.31 | 2.50 |
| | L2-SP | $\underline{59.11 \pm 0.88}$ | $80.40 \pm 1.50$ | $61.10 \pm 1.54$ | $76.67 \pm 1.61$ | $65.11 \pm 0.75$ | $53.81 \pm 0.72$ | $68.96 \pm 0.47$ | $57.85 \pm 0.36$ | 65.38 | 64.80 | 4.88 |
| | FEATURE-MAP | $59.02 \pm 0.89$ | $77.60 \pm 0.45$ | $43.17 \pm 0.32$ | $\underline{79.17 \pm 0.23}$ | $73.54 \pm 0.29$ | $52.23 \pm 0.32$ | $68.74 \pm 0.09$ | $53.39 \pm 0.51$ | 63.36 | 64.09 | 5.75 |
| | BSS | $58.58 \pm 1.31$ | $80.86 \pm 1.92$ | $\mathbf{61.96 \pm 2.00}$ | $79.14 \pm 0.79$ | $73.35 \pm 1.27$ | $53.14 \pm 0.63$ | $70.76 \pm 0.37$ | $60.62 \pm 0.35$ | 67.30 | 67.40 | 3.75 |

Table 8: Robust fine-tuning performance on 5 classification datasets (AUC metrics) in the Few-shot setting (covering FEWSHOT-50, FEWSHOT-100, FEWSHOT-500), evaluated across 3 dataset splits (RANDOM, SCAFFOLD, SIZE) over the Graph-MAE based pre-trained model. We **bold** and underline the best and second-best performances in each scenario.

| SPLIT | METHODS | CLINTOX | BBBP | BACE | HIV | SIDER | AVG | AVG-F | AVG-R |
|---|---|---|---|---|---|---|---|---|---|
| | | | | **FEWSHOT-50** | | | | | |
| RANDOM | FULL-FT | 59.67 ± 3.35 | 83.04 ± 0.39 | 74.97 ± 1.30 | **62.63 ± 0.92** | 52.52 ± 0.19 | 66.57 | 65.76 | 4.20 |
| | LP | 57.56 ± 4.09 | 71.69 ± 0.89 | 72.96 ± 0.91 | 48.27 ± 4.06 | 55.09 ± 0.39 | 61.11 | 61.45 | 6.20 |
| | SURGICAL-FT | 59.83 ± 2.64 | 78.37 ± 1.06 | 75.25 ± 0.92 | 55.35 ± 0.81 | 54.97 ± 0.49 | 64.75 | 63.48 | 4.40 |
| | LP-FT | 60.20 ± 2.14 | **84.54 ± 0.41** | **76.82 ± 0.34** | 62.24 ± 0.28 | 54.41 ± 0.32 | 67.64 | 66.42 | 2.60 |
| | WiSE-FT | **63.50 ± 7.72** | 70.77 ± 1.42 | 70.57 ± 1.13 | 58.10 ± 2.35 | 51.23 ± 2.01 | 62.83 | 64.06 | 6.00 |
| | L2-SP | 61.02 ± 2.03 | 83.79 ± 0.60 | 74.24 ± 0.96 | 61.58 ± 0.81 | 55.34 ± 0.44 | 67.19 | 65.61 | 3.20 |
| | FEATURE-MAP | 59.99 ± 3.80 | 73.57 ± 1.12 | 71.18 ± 2.60 | 48.24 ± 4.14 | **55.85 ± 0.10** | 61.77 | 62.34 | 5.20 |
| | BSS | 58.86 ± 3.63 | 83.81 ± 0.57 | 74.38 ± 1.20 | 62.06 ± 0.80 | 54.46 ± 0.56 | 66.71 | 65.10 | 4.20 |
| SCAFFOLD | FULL-FT | 55.61 ± 2.60 | 58.53 ± 0.58 | 58.21 ± 7.54 | 45.89 ± 4.20 | 54.86 ± 0.67 | 54.62 | 56.23 | 5.60 |
| | LP | 62.76 ± 3.66 | 56.21 ± 1.38 | 56.67 ± 6.74 | 52.12 ± 3.82 | 53.39 ± 0.50 | 56.23 | 55.42 | 6.20 |
| | SURGICAL-FT | 63.53 ± 3.11 | 59.33 ± 0.82 | 60.97 ± 3.53 | 52.62 ± 1.46 | **54.94 ± 0.39** | 58.28 | 58.41 | 3.20 |
| | LP-FT | 60.62 ± 2.83 | 58.45 ± 0.72 | 59.51 ± 1.11 | 51.87 ± 3.30 | 54.67 ± 0.64 | 57.02 | 57.54 | 5.20 |
| | WiSE-FT | 55.45 ± 5.80 | 59.33 ± 0.74 | **67.39 ± 2.69** | 58.03 ± 4.66 | 53.77 ± 0.49 | 58.79 | 57.60 | 4.00 |
| | L2-SP | 64.76 ± 2.87 | 59.99 ± 0.63 | 61.49 ± 1.47 | 51.94 ± 3.28 | 54.31 ± 0.86 | 58.50 | 58.60 | 3.60 |
| | FEATURE-MAP | **68.84 ± 1.77** | 56.59 ± 1.37 | 64.71 ± 2.65 | 43.90 ± 0.98 | 50.07 ± 0.75 | 56.82 | 57.12 | 5.20 |
| | BSS | 60.27 ± 3.40 | **60.16 ± 0.57** | 61.83 ± 1.07 | **62.17 ± 1.89** | 54.35 ± 0.96 | 59.76 | 60.75 | 3.00 |
| SIZE | FULL-FT | 53.86 ± 4.15 | 58.43 ± 1.97 | 45.83 ± 8.42 | 51.39 ± 8.97 | 52.27 ± 0.60 | 52.36 | 52.51 | 5.60 |
| | LP | 52.46 ± 3.47 | 47.60 ± 7.34 | 51.80 ± 9.61 | 46.50 ± 11.95 | 51.79 ± 0.75 | 50.03 | 50.40 | 6.60 |
| | SURGICAL-FT | 53.27 ± 3.82 | 48.97 ± 8.11 | **52.03 ± 9.45** | 52.11 ± 9.11 | 51.95 ± 0.51 | 51.95 | 52.47 | 4.40 |
| | LP-FT | 54.43 ± 3.19 | 59.46 ± 1.82 | 40.76 ± 2.04 | 57.05 ± 1.85 | **53.41 ± 0.19** | 53.02 | 54.96 | 3.40 |
| | WiSE-FT | 56.43 ± 2.94 | **60.62 ± 3.42** | 51.59 ± 4.93 | **66.93 ± 5.90** | 50.96 ± 1.29 | 57.31 | 56.21 | 3.00 |
| | L2-SP | 53.09 ± 0.96 | 58.43 ± 4.43 | 45.90 ± 9.25 | 53.69 ± 4.19 | 52.31 ± 0.70 | 52.68 | 53.03 | 5.00 |
| | FEATURE-MAP | 53.75 ± 1.04 | 60.21 ± 7.22 | 46.65 ± 1.64 | 53.42 ± 4.82 | 51.88 ± 0.54 | 53.18 | 53.02 | 4.20 |
| | BSS | **58.80 ± 1.49** | 59.13 ± 4.12 | 46.62 ± 8.69 | 53.94 ± 4.11 | 51.87 ± 0.64 | 54.07 | 54.87 | 3.80 |
| | | | | **FEWSHOT-100** | | | | | |
| RANDOM | FULL-FT | 67.65 ± 1.95 | 82.80 ± 0.74 | 79.73 ± 0.72 | 62.47 ± 0.47 | 55.03 ± 0.56 | 69.54 | 69.95 | 4.20 |
| | LP | 64.03 ± 2.41 | 72.19 ± 1.10 | 75.93 ± 1.12 | 48.46 ± 3.79 | 58.11 ± 0.51 | 63.74 | 64.78 | 6.40 |
| | SURGICAL-FT | 66.99 ± 2.08 | 81.07 ± 0.32 | 79.05 ± 0.49 | 54.93 ± 0.64 | 58.16 ± 0.60 | 68.04 | 68.07 | 5.00 |
| | LP-FT | 66.54 ± 1.29 | **84.02 ± 0.63** | **81.49 ± 0.40** | **62.60 ± 0.83** | 57.29 ± 0.49 | 70.39 | 70.21 | 2.80 |
| | WiSE-FT | **69.92 ± 3.24** | 81.88 ± 3.16 | 71.01 ± 1.00 | 59.41 ± 1.02 | 52.12 ± 1.56 | 66.87 | 66.78 | 5.40 |
| | L2-SP | 68.17 ± 0.71 | 83.52 ± 0.97 | 80.29 ± 0.64 | 61.40 ± 0.73 | **58.85 ± 0.38** | 70.45 | 69.95 | 2.80 |
| | FEATURE-MAP | 63.25 ± 1.14 | 73.95 ± 1.04 | 74.90 ± 2.19 | 48.29 ± 4.11 | 58.80 ± 0.21 | 63.84 | 65.33 | 6.40 |
| | BSS | 68.22 ± 0.52 | 83.55 ± 0.97 | 80.32 ± 0.67 | 62.24 ± 0.61 | 56.13 ± 0.74 | 70.09 | 70.26 | 3.00 |
| SCAFFOLD | FULL-FT | 63.22 ± 5.57 | 60.67 ± 0.99 | 65.72 ± 2.20 | 54.23 ± 2.65 | 54.93 ± 0.84 | 59.75 | 59.61 | 4.80 |
| | LP | 61.64 ± 3.21 | 53.87 ± 0.93 | 60.85 ± 1.01 | 53.99 ± 4.84 | 53.02 ± 0.35 | 56.67 | 56.24 | 7.00 |
| | SURGICAL-FT | 66.38 ± 1.62 | 58.25 ± 0.90 | 62.95 ± 2.47 | **62.20 ± 1.88** | 55.24 ± 0.47 | 61.00 | 61.13 | 4.00 |
| | LP-FT | 65.08 ± 3.59 | 60.15 ± 0.20 | 66.58 ± 0.96 | 57.03 ± 3.48 | 54.12 ± 0.52 | 60.59 | 60.75 | 4.60 |
| | WiSE-FT | 53.83 ± 2.78 | **64.13 ± 1.64** | **72.12 ± 1.43** | 57.64 ± 4.40 | **55.64 ± 2.15** | 60.67 | 59.14 | 2.80 |
| | L2-SP | 66.91 ± 1.79 | 60.77 ± 1.57 | 66.02 ± 1.53 | 54.34 ± 2.25 | 54.72 ± 1.16 | 60.55 | 60.50 | 3.80 |
| | FEATURE-MAP | **68.84 ± 1.56** | 55.98 ± 0.58 | 64.15 ± 2.87 | 50.87 ± 2.38 | 49.55 ± 0.88 | 57.88 | 57.00 | 6.00 |
| | BSS | 67.11 ± 2.10 | 60.54 ± 1.13 | 66.61 ± 1.12 | 60.74 ± 0.93 | 55.06 ± 1.14 | 62.01 | 62.63 | 2.60 |
| SIZE | FULL-FT | 55.01 ± 3.57 | 66.52 ± 1.39 | 51.73 ± 2.47 | 54.13 ± 8.59 | 53.93 ± 0.76 | 56.26 | 54.36 | 3.60 |
| | LP | 52.73 ± 3.21 | 49.27 ± 5.99 | 47.22 ± 6.09 | 46.39 ± 11.18 | 51.72 ± 0.76 | 49.47 | 49.40 | 7.40 |
| | SURGICAL-FT | 53.80 ± 3.52 | 52.34 ± 6.18 | 49.29 ± 5.93 | 51.50 ± 12.55 | 53.47 ± 0.71 | 52.08 | 52.44 | 6.00 |
| | LP-FT | 54.19 ± 2.32 | 67.66 ± 1.06 | 54.39 ± 2.27 | 58.09 ± 1.24 | **55.25 ± 0.33** | 57.92 | 55.91 | 2.40 |
| | WiSE-FT | 54.89 ± 5.22 | 65.76 ± 1.61 | 48.32 ± 2.36 | **67.43 ± 6.52** | 47.06 ± 0.94 | 56.69 | 56.32 | 4.60 |
| | L2-SP | 53.99 ± 1.00 | 66.39 ± 3.08 | 54.50 ± 3.14 | 53.69 ± 7.69 | 54.34 ± 1.20 | 56.75 | 54.45 | 3.40 |
| | FEATURE-MAP | 50.62 ± 1.90 | 58.47 ± 9.57 | 46.18 ± 1.57 | 52.40 ± 5.59 | 51.81 ± 0.64 | 51.90 | 51.61 | 6.80 |
| | BSS | **58.71 ± 1.44** | **67.67 ± 2.91** | **54.89 ± 3.17** | 54.60 ± 7.72 | 54.33 ± 1.18 | 58.04 | 56.07 | 1.80 |
| | | | | **FEWSHOT-500** | | | | | |
| RANDOM | FULL-FT | 78.63 ± 0.77 | 91.08 ± 1.35 | 85.62 ± 0.30 | **70.55 ± 0.32** | 59.68 ± 0.36 | 77.11 | 78.27 | 4.40 |
| | LP | 72.34 ± 2.23 | 79.79 ± 1.23 | 75.57 ± 1.04 | 54.42 ± 2.54 | 61.10 ± 0.33 | 68.64 | 69.67 | 7.20 |
| | SURGICAL-FT | 79.09 ± 0.81 | 85.22 ± 0.36 | 83.77 ± 0.94 | 65.78 ± 0.56 | 61.10 ± 0.47 | 74.99 | 76.21 | 5.00 |
| | LP-FT | **80.52 ± 1.76** | 91.82 ± 0.25 | **86.02 ± 0.20** | 69.28 ± 0.65 | 61.10 ± 0.48 | 77.75 | 78.61 | 2.20 |
| | WiSE-FT | 78.34 ± 3.82 | 91.54 ± 0.76 | 84.49 ± 0.56 | 61.15 ± 1.37 | **63.77 ± 1.03** | 75.86 | 75.53 | 4.20 |
| | L2-SP | 78.56 ± 0.91 | 91.38 ± 0.46 | 85.81 ± 0.40 | 68.73 ± 0.18 | 61.34 ± 0.30 | 77.16 | 77.70 | 3.80 |
| | FEATURE-MAP | 69.96 ± 1.65 | 81.31 ± 0.48 | 71.65 ± 0.61 | 58.54 ± 1.57 | 61.40 ± 0.19 | 68.57 | 67.67 | 6.40 |
| | BSS | 79.17 ± 0.93 | **91.98 ± 0.48** | 85.85 ± 0.41 | 69.74 ± 0.41 | 60.32 ± 0.51 | 77.41 | 78.25 | 2.80 |
| SCAFFOLD | FULL-FT | 68.64 ± 0.79 | 68.65 ± 0.62 | 77.69 ± 0.21 | 66.32 ± 1.81 | 57.55 ± 0.33 | 67.77 | 67.87 | 4.20 |
| | LP | 67.38 ± 2.22 | 60.02 ± 0.77 | 62.66 ± 5.53 | 60.14 ± 4.18 | 58.74 ± 1.34 | 61.79 | 60.94 | 6.40 |
| | SURGICAL-FT | **70.31 ± 2.21** | 65.27 ± 0.39 | 74.86 ± 1.30 | **70.52 ± 1.05** | 61.99 ± 0.40 | 68.59 | 68.70 | 3.00 |
| | LP-FT | 65.58 ± 1.33 | 69.05 ± 0.77 | 78.48 ± 0.58 | 70.22 ± 0.94 | 55.89 ± 0.40 | 67.84 | 68.28 | 4.60 |
| | WiSE-FT | 68.48 ± 3.60 | 65.58 ± 1.56 | **82.78 ± 0.77** | 58.90 ± 2.63 | 57.28 ± 0.75 | 66.60 | 64.32 | 5.00 |
| | L2-SP | 68.86 ± 1.22 | 68.81 ± 0.65 | 78.24 ± 1.13 | 65.12 ± 1.11 | 60.63 ± 0.73 | 68.33 | 67.60 | 3.40 |
| | FEATURE-MAP | 68.16 ± 0.88 | 59.42 ± 0.29 | 68.25 ± 1.93 | 67.01 ± 2.26 | 56.57 ± 0.43 | 63.88 | 64.86 | 6.20 |
| | BSS | 68.59 ± 1.15 | **69.09 ± 0.57** | 78.85 ± 0.93 | 66.05 ± 2.20 | 58.73 ± 0.39 | 68.26 | 67.91 | 2.60 |
| SIZE | FULL-FT | 65.78 ± 1.28 | 83.11 ± 0.77 | 49.15 ± 1.50 | 58.35 ± 9.96 | 52.46 ± 1.33 | 61.77 | 58.86 | 4.00 |
| | LP | 58.59 ± 2.86 | 60.74 ± 5.06 | 47.28 ± 2.25 | 45.96 ± 11.56 | 51.67 ± 0.43 | 52.85 | 52.51 | 7.40 |
| | SURGICAL-FT | 65.88 ± 1.23 | 72.86 ± 1.29 | 47.62 ± 1.58 | 57.44 ± 9.55 | 52.61 ± 0.51 | 59.28 | 58.64 | 4.80 |
| | LP-FT | 66.09 ± 1.44 | 82.96 ± 0.52 | 50.17 ± 0.69 | 63.07 ± 0.97 | 52.25 ± 0.55 | 62.91 | 60.47 | 2.60 |
| | WiSE-FT | 57.72 ± 2.58 | 77.31 ± 1.56 | **60.42 ± 2.45** | **68.17 ± 2.47** | 51.52 ± 0.50 | 63.03 | 62.10 | 4.60 |
| | L2-SP | 65.91 ± 2.13 | 82.22 ± 0.63 | 49.40 ± 0.87 | 60.24 ± 2.10 | **52.79 ± 0.72** | 62.68 | 59.65 | 3.20 |
| | FEATURE-MAP | 60.84 ± 1.37 | 63.60 ± 6.18 | 44.07 ± 0.77 | 49.33 ± 7.05 | 51.80 ± 0.59 | 53.93 | 53.99 | 6.80 |
| | BSS | **66.64 ± 2.47** | 83.60 ± 0.32 | 49.73 ± 0.59 | 62.63 ± 1.27 | 52.24 ± 0.98 | 62.97 | 60.50 | 2.60 |

Table 9: Robust fine-tuning performance on 4 regression datasets (RMSE metrics) in both Fewshot and Non-Fewshot settings (covering NON-FEWSHOT, FEWSHOT-50, FEWSHOT-100, FEWSHOT-500), evaluated across 3 dataset splits (RANDOM, SCAFFOLD, SIZE) over the Graph-MAE based PT model. AVG-R,AVG-R* denote the average rank and the rank based on the average normalized performance over all the datasets for each evavluated method, respectively. Standard deviations across five replicates are shown in parentheses. We **bold** and underline the best and second-best performances in each scenario.

| SPLIT | METHODS | NON-FEWSHOT | | | | | | FEWSHOT-50 | | | | | |
|---|---|---|---|---|---|---|---|---|---|---|---|---|---|
| | | ESOL | LIPO | MALARIA | CEP | AVG-R | AVG-R* | ESOL | LIPO | MALARIA | CEP | AVG-R | AVG-R* |
| RANDOM | FULL-FT | 0.987 ± 0.013 | 0.734 ± 0.007 | 1.109 ± 0.015 | 1.342 ± 0.015 | 3.00 | 3 | 1.432 ± 0.019 | 1.328 ± 0.051 | 1.297 ± 0.015 | 2.927 ± 0.226 | 4.25 | 3 |
| | LP | 1.394 ± 0.012 | 1.156 ± 0.001 | 1.263 ± 0.002 | 3.079 ± 0.105 | 8.00 | 8 | 1.646 ± 0.027 | 1.395 ± 0.076 | 1.334 ± 0.009 | 4.133 ± 0.372 | 7.50 | 8 |
| | SURGICAL-FT | 1.088 ± 0.011 | 0.883 ± 0.007 | 1.120 ± 0.012 | 1.697 ± 0.012 | 6.25 | 6 | 1.497 ± 0.017 | 1.303 ± 0.051 | 1.309 ± 0.017 | 3.300 ± 0.406 | 5.00 | 7 |
| | LP-FT | 0.953 ± 0.009 | 0.743 ± 0.006 | 1.096 ± 0.009 | 1.322 ± 0.025 | 1.75 | 1 | 1.386 ± 0.022 | 1.217 ± 0.021 | 1.399 ± 0.033 | 2.840 ± 0.226 | 3.75 | 2 |
| | WiSE-FT | 1.210 ± 0.032 | 0.846 ± 0.023 | 1.060 ± 0.008 | 1.531 ± 0.030 | 4.50 | 5 | 1.622 ± 0.053 | 1.343 ± 0.010 | 1.248 ± 0.008 | 2.385 ± 0.026 | 3.75 | 5 |
| | L2-SP | 0.995 ± 0.024 | 0.787 ± 0.008 | 1.115 ± 0.006 | 1.363 ± 0.040 | 4.25 | 4 | 1.444 ± 0.027 | 1.354 ± 0.052 | 1.294 ± 0.005 | 2.315 ± 0.106 | 3.75 | 1 |
| | FEATURE-MAP | 1.297 ± 0.007 | 1.080 ± 0.002 | 1.115 ± 0.016 | 1.473 ± 0.018 | 6.25 | 7 | 1.655 ± 0.027 | 1.312 ± 0.020 | 1.278 ± 0.003 | 2.363 ± 0.127 | 3.75 | 6 |
| | BSS | 0.975 ± 0.019 | 0.725 ± 0.011 | 1.100 ± 0.004 | 1.334 ± 0.004 | 2.00 | 2 | 1.439 ± 0.029 | 1.351 ± 0.051 | 1.294 ± 0.005 | 2.682 ± 0.115 | 4.25 | 4 |
| SCAFFOLD | FULL-FT | 1.332 ± 0.015 | 0.808 ± 0.008 | 1.104 ± 0.007 | 1.327 ± 0.017 | 3.50 | 3 | 1.717 ± 0.028 | 1.214 ± 0.051 | 1.169 ± 0.005 | 2.612 ± 0.178 | 5.50 | 5 |
| | LP | 1.703 ± 0.016 | 1.043 ± 0.006 | 1.150 ± 0.003 | 3.102 ± 0.136 | 8.00 | 8 | 2.209 ± 0.039 | 1.183 ± 0.045 | 1.170 ± 0.004 | 4.565 ± 0.048 | 6.75 | 8 |
| | SURGICAL-FT | 1.335 ± 0.005 | 0.884 ± 0.007 | 1.111 ± 0.013 | 1.669 ± 0.022 | 6.00 | 6 | 1.834 ± 0.031 | 1.198 ± 0.049 | 1.166 ± 0.001 | 3.142 ± 0.589 | 5.00 | 6 |
| | LP-FT | 1.312 ± 0.024 | 0.788 ± 0.005 | 1.104 ± 0.006 | 1.318 ± 0.017 | 2.00 | 1 | 1.642 ± 0.026 | 1.147 ± 0.008 | 1.300 ± 0.061 | 2.879 ± 0.264 | 4.00 | 6 |
| | WiSE-FT | 1.617 ± 0.031 | 0.891 ± 0.009 | 1.077 ± 0.004 | 1.498 ± 0.034 | 5.00 | 5 | 2.221 ± 0.047 | 1.175 ± 0.016 | 1.166 ± 0.002 | 2.326 ± 0.031 | 4.00 | 4 |
| | L2-SP | 1.329 ± 0.030 | 0.835 ± 0.011 | 1.108 ± 0.011 | 1.325 ± 0.021 | 4.00 | 4 | 1.718 ± 0.053 | 1.200 ± 0.053 | 1.167 ± 0.002 | 2.366 ± 0.059 | 4.25 | 3 |
| | FEATURE-MAP | 1.551 ± 0.013 | 0.994 ± 0.004 | 1.097 ± 0.008 | 1.415 ± 0.030 | 5.00 | 7 | 2.197 ± 0.075 | 1.148 ± 0.023 | 1.163 ± 0.003 | 2.400 ± 0.175 | 3.00 | 2 |
| | BSS | 1.326 ± 0.029 | 0.803 ± 0.013 | 1.104 ± 0.009 | 1.302 ± 0.012 | 2.50 | 2 | 1.712 ± 0.056 | 1.168 ± 0.050 | 1.168 ± 0.002 | 2.551 ± 0.121 | 3.50 | 1 |
| SIZE | FULL-FT | 1.822 ± 0.099 | 0.814 ± 0.013 | 0.908 ± 0.005 | 1.722 ± 0.016 | 3.25 | 3 | 2.654 ± 0.075 | 1.557 ± 0.093 | 0.943 ± 0.026 | 2.554 ± 0.053 | 4.50 | 3 |
| | LP | 2.309 ± 0.030 | 1.024 ± 0.014 | 0.927 ± 0.010 | 3.814 ± 0.175 | 7.75 | 8 | 2.818 ± 0.087 | 1.676 ± 0.115 | 0.963 ± 0.030 | 5.414 ± 0.036 | 7.25 | 8 |
| | SURGICAL-FT | 1.915 ± 0.036 | 0.886 ± 0.013 | 0.925 ± 0.003 | 2.135 ± 0.038 | 6.00 | 6 | 2.658 ± 0.088 | 1.641 ± 0.114 | 0.929 ± 0.027 | 3.423 ± 0.550 | 6.00 | 6 |
| | LP-FT | 1.754 ± 0.075 | 0.795 ± 0.005 | 0.907 ± 0.020 | 1.710 ± 0.010 | 1.75 | 2 | 2.440 ± 0.056 | 1.422 ± 0.111 | 1.166 ± 0.053 | 2.339 ± 0.049 | 2.75 | 1 |
| | WiSE-FT | 2.323 ± 0.041 | 0.974 ± 0.016 | 0.895 ± 0.011 | 1.982 ± 0.039 | 5.50 | 7 | 3.050 ± 0.087 | 1.513 ± 0.049 | 0.909 ± 0.001 | 3.223 ± 0.224 | 4.25 | 7 |
| | L2-SP | 1.849 ± 0.041 | 0.849 ± 0.025 | 0.911 ± 0.006 | 1.748 ± 0.041 | 4.50 | 4 | 2.606 ± 0.085 | 1.614 ± 0.112 | 0.914 ± 0.016 | 2.466 ± 0.079 | 3.25 | 2 |
| | FEATURE-MAP | 2.136 ± 0.030 | 1.007 ± 0.015 | 0.891 ± 0.012 | 1.947 ± 0.013 | 4.75 | 5 | 2.630 ± 0.036 | 1.497 ± 0.080 | 0.920 ± 0.007 | 2.408 ± 0.057 | 4.25 | 5 |
| | BSS | 1.808 ± 0.039 | 0.818 ± 0.020 | 0.899 ± 0.006 | 1.712 ± 0.021 | 2.50 | 1 | 2.579 ± 0.066 | 1.613 ± 0.110 | 0.926 ± 0.018 | 2.580 ± 0.157 | 3.75 | 4 |

| SPLIT | METHODS | FEWSHOT-100 | | | | | | FEWSHOT-500 | | | | | |
|---|---|---|---|---|---|---|---|---|---|---|---|---|---|
| | | ESOL | LIPO | MALARIA | CEP | AVG-R | AVG-R* | ESOL | LIPO | MALARIA | CEP | AVG-R | AVG-R* |
| RANDOM | FULL-FT | 1.304 ± 0.041 | 1.239 ± 0.032 | 1.289 ± 0.003 | 3.028 ± 0.310 | 3.25 | 3 | 1.042 ± 0.017 | 1.023 ± 0.022 | 1.290 ± 0.004 | 1.958 ± 0.038 | 4.00 | 5 |
| | LP | 1.609 ± 0.032 | 1.285 ± 0.043 | 1.334 ± 0.009 | 4.562 ± 0.047 | 7.50 | 8 | 1.487 ± 0.011 | 1.233 ± 0.019 | 1.331 ± 0.012 | 4.602 ± 0.019 | 8.00 | 8 |
| | SURGICAL-FT | 1.356 ± 0.022 | 1.219 ± 0.016 | 1.298 ± 0.008 | 3.100 ± 0.805 | 4.50 | 4 | 1.164 ± 0.010 | 1.127 ± 0.007 | 1.240 ± 0.011 | 3.577 ± 0.398 | 5.00 | 6 |
| | LP-FT | 1.310 ± 0.021 | 1.226 ± 0.021 | 1.374 ± 0.045 | 3.241 ± 0.438 | 4.75 | 6 | 0.995 ± 0.010 | 0.975 ± 0.007 | 1.310 ± 0.019 | 2.004 ± 0.056 | 3.75 | 4 |
| | WiSE-FT | 1.600 ± 0.051 | 1.324 ± 0.013 | 1.245 ± 0.017 | 2.294 ± 0.024 | 4.75 | 7 | 1.251 ± 0.029 | 0.976 ± 0.010 | 1.231 ± 0.016 | 1.975 ± 0.017 | 3.25 | 2 |
| | L2-SP | 1.323 ± 0.034 | 1.253 ± 0.029 | 1.276 ± 0.014 | 2.271 ± 0.065 | 3.25 | 1 | 1.048 ± 0.014 | 1.036 ± 0.009 | 1.241 ± 0.007 | 1.886 ± 0.032 | 3.25 | 1 |
| | FEATURE-MAP | 1.526 ± 0.030 | 1.243 ± 0.008 | 1.276 ± 0.004 | 2.271 ± 0.116 | 3.75 | 5 | 1.340 ± 0.007 | 1.202 ± 0.004 | 1.241 ± 0.010 | 1.992 ± 0.013 | 5.75 | 7 |
| | BSS | 1.322 ± 0.033 | 1.251 ± 0.028 | 1.293 ± 0.006 | 2.541 ± 0.128 | 4.25 | 2 | 1.031 ± 0.013 | 1.020 ± 0.006 | 1.272 ± 0.007 | 1.896 ± 0.034 | 3.00 | 3 |
| SCAFFOLD | FULL-FT | 1.695 ± 0.045 | 1.168 ± 0.030 | 1.167 ± 0.003 | 3.087 ± 0.765 | 4.50 | 3 | 1.406 ± 0.016 | 0.945 ± 0.021 | 1.199 ± 0.025 | 2.057 ± 0.072 | 4.75 | 3 |
| | LP | 2.045 ± 0.044 | 1.211 ± 0.064 | 1.173 ± 0.004 | 4.579 ± 0.037 | 7.50 | 8 | 1.849 ± 0.028 | 1.102 ± 0.019 | 1.182 ± 0.007 | 4.607 ± 0.020 | 7.00 | 8 |
| | SURGICAL-FT | 1.693 ± 0.019 | 1.146 ± 0.017 | 1.169 ± 0.003 | 3.226 ± 0.563 | 4.50 | 1 | 1.436 ± 0.010 | 1.020 ± 0.006 | 1.156 ± 0.010 | 2.874 ± 0.063 | 5.00 | 5 |
| | LP-FT | 1.626 ± 0.016 | 1.123 ± 0.011 | 1.312 ± 0.023 | 2.782 ± 0.364 | 3.75 | 6 | 1.354 ± 0.011 | 0.940 ± 0.012 | 1.278 ± 0.044 | 2.052 ± 0.053 | 3.75 | 6 |
| | WiSE-FT | 2.069 ± 0.066 | 1.205 ± 0.014 | 1.158 ± 0.008 | 2.244 ± 0.068 | 4.25 | 7 | 1.707 ± 0.029 | 1.028 ± 0.025 | 1.125 ± 0.008 | 1.906 ± 0.020 | 3.50 | 4 |
| | L2-SP | 1.679 ± 0.045 | 1.201 ± 0.048 | 1.168 ± 0.003 | 2.327 ± 0.030 | 3.50 | 3 | 1.413 ± 0.045 | 0.943 ± 0.022 | 1.156 ± 0.012 | 1.931 ± 0.054 | 3.50 | 1 |
| | FEATURE-MAP | 1.964 ± 0.034 | 1.164 ± 0.029 | 1.164 ± 0.001 | 2.341 ± 0.095 | 3.50 | 5 | 1.880 ± 0.035 | 1.081 ± 0.006 | 1.129 ± 0.006 | 1.992 ± 0.008 | 5.25 | 7 |
| | BSS | 1.681 ± 0.043 | 1.191 ± 0.046 | 1.169 ± 0.004 | 2.566 ± 0.149 | 4.50 | 4 | 1.404 ± 0.042 | 0.941 ± 0.019 | 1.199 ± 0.029 | 1.926 ± 0.041 | 3.25 | 2 |
| SIZE | FULL-FT | 2.414 ± 0.081 | 1.283 ± 0.070 | 0.911 ± 0.008 | 2.677 ± 0.139 | 3.00 | 1 | 2.102 ± 0.080 | 0.968 ± 0.032 | 0.955 ± 0.031 | 2.283 ± 0.060 | 3.50 | 4 |
| | LP | 2.859 ± 0.078 | 1.493 ± 0.115 | 0.951 ± 0.030 | 5.420 ± 0.033 | 7.50 | 8 | 2.486 ± 0.040 | 1.140 ± 0.046 | 0.968 ± 0.027 | 5.452 ± 0.008 | 7.50 | 8 |
| | SURGICAL-FT | 2.537 ± 0.059 | 1.301 ± 0.074 | 0.909 ± 0.003 | 3.707 ± 0.589 | 4.75 | 6 | 2.142 ± 0.062 | 0.982 ± 0.014 | 0.949 ± 0.032 | 3.765 ± 0.499 | 4.50 | 7 |
| | LP-FT | 2.217 ± 0.047 | 1.146 ± 0.022 | 1.065 ± 0.020 | 2.562 ± 0.076 | 3.50 | 4 | 2.003 ± 0.037 | 0.889 ± 0.017 | 0.985 ± 0.033 | 2.339 ± 0.049 | 3.75 | 2 |
| | WiSE-FT | 2.507 ± 0.098 | 1.297 ± 0.038 | 0.904 ± 0.002 | 2.823 ± 0.031 | 3.75 | 3 | 2.302 ± 0.057 | 1.040 ± 0.015 | 0.906 ± 0.003 | 2.437 ± 0.032 | 5.00 | 5 |
| | L2-SP | 2.442 ± 0.047 | 1.362 ± 0.082 | 0.916 ± 0.009 | 2.451 ± 0.093 | 4.50 | 5 | 2.030 ± 0.059 | 1.012 ± 0.030 | 0.951 ± 0.030 | 2.208 ± 0.030 | 3.25 | 3 |
| | FEATURE-MAP | 2.716 ± 0.026 | 1.551 ± 0.085 | 0.912 ± 0.003 | 2.424 ± 0.039 | 5.00 | 7 | 2.253 ± 0.017 | 1.174 ± 0.023 | 0.908 ± 0.001 | 2.341 ± 0.027 | 5.25 | 6 |
| | BSS | 2.434 ± 0.046 | 1.358 ± 0.084 | 0.912 ± 0.005 | 2.533 ± 0.103 | 4.00 | 2 | 1.980 ± 0.051 | 0.989 ± 0.025 | 0.956 ± 0.041 | 2.237 ± 0.058 | 3.25 | 1 |

Table 10: DWiSE-FT performance on 4 regression datasets (RMSE metrics) in the few-shot setting with 50,100, 500 samples, evaluated across 3 dataset splits (RANDOM, SCAFFOLD, SIZE) given Mole-BERT model. AVG-R denote the average rank. Standard deviations across five replicates are shown in parentheses. We **bold** and underline the best and second-best performances in each scenario.

| SPLIT | METHODS | FEWSHOT 50 | | | | | FEWSHOT 100 | | | | | FEWSHOT 500 | | | | |
|---|---|---|---|---|---|---|---|---|---|---|---|---|---|---|---|---|
| | | ESOL | LIPO | MALARIA | CEP | AVG | ESOL | LIPO | MALARIA | CEP | AVG | ESOL | LIPO | MALARIA | CEP | AVG |
| RANDOM | WiSE-FT | 1.384 ± 0.047 | 1.212 ± 0.020 | 1.276 ± 0.007 | 2.410 ± 0.051 | 3.75 | 1.189 ± 0.030 | 1.142 ± 0.025 | 1.256 ± 0.006 | 2.211 ± 0.028 | 3.00 | 0.995 ± 0.010 | 0.855 ± 0.011 | 1.193 ± 0.003 | 1.893 ± 0.021 | 3.75 |
| | L2-SP | 1.372 ± 0.029 | 1.196 ± 0.019 | 1.277 ± 0.006 | 2.280 ± 0.031 | 3.00 | 1.161 ± 0.016 | 1.149 ± 0.007 | 1.260 ± 0.004 | 2.131 ± 0.014 | 3.25 | 0.878 ± 0.026 | 0.806 ± 0.007 | 1.192 ± 0.004 | 1.893 ± 0.018 | 1.50 |
| | TOP | 1.329 ± 0.021 | 1.164 ± 0.010 | 1.271 ± 0.007 | 2.275 ± 0.022 | 1.25 | 1.120 ± 0.038 | 1.139 ± 0.017 | 1.256 ± 0.006 | 2.131 ± 0.014 | 1.50 | 0.878 ± 0.026 | 0.806 ± 0.007 | 1.192 ± 0.004 | 1.862 ± 0.010 | 1.00 |
| | DWiSE-FT | 1.378 ± 0.055 | 1.189 ± 0.020 | 1.273 ± 0.009 | 2.222 ± 0.059 | 2.00 | 1.132 ± 0.025 | 1.138 ± 0.028 | 1.256 ± 0.004 | 2.129 ± 0.020 | 1.25 | 0.918 ± 0.012 | 0.818 ± 0.013 | 1.192 ± 0.004 | 1.865 ± 0.030 | 2.25 |
| SCAFFOLD | WiSE-FT | 1.842 ± 0.056 | 1.177 ± 0.009 | 1.162 ± 0.004 | 2.454 ± 0.043 | 3.50 | 1.544 ± 0.063 | 1.151 ± 0.007 | 1.162 ± 0.004 | 2.301 ± 0.042 | 3.50 | 1.388 ± 0.023 | 0.834 ± 0.012 | 1.114 ± 0.002 | 1.936 ± 0.037 | 3.25 |
| | L2-SP | 1.699 ± 0.049 | 1.086 ± 0.009 | 1.162 ± 0.002 | 2.331 ± 0.024 | 2.50 | 1.473 ± 0.009 | 0.961 ± 0.003 | 1.153 ± 0.002 | 2.201 ± 0.038 | 2.50 | 1.163 ± 0.026 | 0.813 ± 0.010 | 1.126 ± 0.011 | 1.885 ± 0.011 | 2.50 |
| | TOP | 1.680 ± 0.042 | 1.036 ± 0.007 | 1.159 ± 0.000 | 2.292 ± 0.026 | 1.25 | 1.436 ± 0.054 | 0.937 ± 0.008 | 1.149 ± 0.003 | 2.187 ± 0.031 | 1.25 | 1.112 ± 0.015 | 0.802 ± 0.003 | 1.114 ± 0.002 | 1.881 ± 0.010 | 1.00 |
| | DWiSE-FT | 1.616 ± 0.047 | 1.110 ± 0.013 | 1.173 ± 0.005 | 2.306 ± 0.030 | 2.50 | 1.485 ± 0.041 | 0.979 ± 0.014 | 1.149 ± 0.040 | 2.149 ± 0.040 | 2.75 | 1.266 ± 0.021 | 0.823 ± 0.010 | 1.121 ± 0.004 | 1.900 ± 0.019 | 3.00 |
| SIZE | WiSE-FT | 2.615 ± 0.072 | 1.391 ± 0.042 | 0.929 ± 0.004 | 2.762 ± 0.053 | 4.00 | 2.216 ± 0.056 | 1.124 ± 0.031 | 0.917 ± 0.004 | 2.543 ± 0.027 | 3.75 | 2.071 ± 0.078 | 0.902 ± 0.016 | 0.912 ± 0.003 | 2.379 ± 0.086 | 3.75 |
| | L2-SP | 2.393 ± 0.068 | 1.306 ± 0.037 | 0.915 ± 0.002 | 2.497 ± 0.019 | 2.50 | 1.731 ± 0.071 | 1.025 ± 0.028 | 0.905 ± 0.002 | 2.424 ± 0.024 | 1.75 | 1.629 ± 0.084 | 0.821 ± 0.011 | 0.904 ± 0.003 | 2.368 ± 0.013 | 2.50 |
| | TOP | 2.369 ± 0.075 | 1.297 ± 0.040 | 0.911 ± 0.002 | 2.497 ± 0.019 | 1.50 | 1.731 ± 0.071 | 1.025 ± 0.028 | 0.898 ± 0.003 | 2.424 ± 0.024 | 1.50 | 1.629 ± 0.084 | 0.803 ± 0.006 | 0.895 ± 0.002 | 2.328 ± 0.017 | 1.50 |
| | DWiSE-FT | 1.488 ± 0.101 | 1.113 ± 0.021 | 0.913 ± 0.007 | 2.539 ± 0.023 | 1.75 | 1.469 ± 0.052 | 1.031 ± 0.022 | 0.920 ± 0.006 | 2.390 ± 0.025 | 2.25 | 1.466 ± 0.040 | 0.816 ± 0.022 | 0.915 ± 0.003 | 2.322 ± 0.031 | 2.00 |

Table 11: Robust fine-tuning performance on 8 classification datasets (AUC metrics) in the Non-Fewshot setting, evaluated across 3 dataset splits (RANDOM, SCAFFOLD, SIZE) under the MoleculeSTM pre-trained model. AVG, AVG-F, AVG-R denote the average AUC metrics, average AUC without max and min values, and average rank over all the datasets for each evaluated method, respectively. Standard deviations across five replicates are shown in parentheses. We **bold** and underline the best and second-best performances in each scenario.

| SPLIT | METHODS | CLINTOX | BBBP | BACE | HIV | MUV | SIDER | TOX21 | TOXCAST | AVG | AVG-F | AVG-R |
|---|---|---|---|---|---|---|---|---|---|---|---|---|
| RANDOM | FULL-FT | 89.90 ± 1.49 | 93.43 ± 0.99 | 89.82 ± 1.08 | 84.72 ± 1.11 | 77.82 ± 3.46 | 62.12 ± 1.15 | 82.49 ± 0.41 | 72.95 ± 0.31 | 81.66 | 82.95 | 3.62 |
|  | LP | 74.32 ± 1.90 | 84.76 ± 0.29 | 74.85 ± 0.27 | 74.15 ± 0.69 | 76.86 ± 1.07 | 59.69 ± 0.24 | 73.72 ± 0.20 | 66.19 ± 0.14 | 73.07 | 73.35 | 7.75 |
|  | SURGICAL-FT | 86.04 ± 0.89 | 93.68 ± 0.51 | 89.99 ± 0.46 | **85.68 ± 0.84** | 79.59 ± 2.47 | **63.64 ± 0.78** | 81.84 ± 0.66 | 71.83 ± 0.55 | 81.54 | 82.50 | 3.38 |
|  | LP-FT | 86.39 ± 1.85 | 93.72 ± 0.93 | 89.82 ± 0.57 | 84.17 ± 1.41 | 76.87 ± 2.38 | 62.19 ± 1.00 | 82.54 ± 0.51 | 72.19 ± 0.52 | 80.99 | 82.00 | 3.75 |
|  | WISE-FT | **90.35 ± 1.26** | 92.93 ± 0.80 | **90.41 ± 0.86** | 84.38 ± 1.05 | 77.23 ± 3.08 | 62.17 ± 1.25 | 82.67 ± 0.32 | 73.08 ± 0.32 | 81.65 | 83.02 | 2.88 |
|  | $L^2$-SP | 89.69 ± 1.39 | 93.77 ± 0.37 | 89.21 ± 0.92 | 81.94 ± 1.20 | 50.21 ± 4.41 | 61.07 ± 1.22 | 82.97 ± 0.39 | 71.02 ± 0.57 | 77.48 | 79.32 | 5.00 |
|  | FEATURE-MAP | 79.93 ± 1.54 | 90.59 ± 0.39 | 83.69 ± 0.24 | 77.66 ± 0.46 | **80.03 ± 1.01** | 59.93 ± 0.14 | 75.32 ± 0.19 | 67.51 ± 0.30 | 76.83 | 77.36 | 6.25 |
|  | BSS | 90.17 ± 2.84 | **94.16 ± 0.55** | 89.74 ± 1.12 | 83.96 ± 1.29 | 76.64 ± 1.29 | 61.87 ± 0.69 | **83.26 ± 0.57** | **74.55 ± 0.31** | 81.79 | 83.05 | 3.38 |
| SCAFFOLD | FULL-FT | 74.94 ± 7.23 | 68.62 ± 0.80 | 75.35 ± 2.06 | 76.03 ± 0.91 | 73.43 ± 2.50 | 57.88 ± 1.18 | 76.67 ± 0.68 | 63.62 ± 0.27 | 70.82 | 72.00 | 4.25 |
|  | LP | 65.07 ± 1.08 | 59.39 ± 0.35 | 69.24 ± 0.16 | 69.97 ± 0.57 | 71.81 ± 2.40 | 59.93 ± 0.37 | 69.87 ± 0.28 | 60.05 ± 0.25 | 65.67 | 65.69 | 7.00 |
|  | SURGICAL-FT | 71.07 ± 4.16 | 67.78 ± 0.60 | 80.16 ± 2.36 | **76.80 ± 1.06** | 75.87 ± 0.82 | 59.24 ± 1.22 | 75.54 ± 0.64 | 63.27 ± 0.70 | 71.22 | 71.72 | 3.75 |
|  | LP-FT | 75.07 ± 2.24 | 67.05 ± 1.42 | 75.33 ± 1.14 | 76.68 ± 0.82 | 71.36 ± 1.39 | 58.51 ± 1.15 | 76.85 ± 0.63 | 62.98 ± 0.51 | 70.48 | 71.41 | 4.62 |
|  | WISE-FT | **77.27 ± 4.28** | 68.72 ± 0.75 | 77.37 ± 1.44 | 75.91 ± 0.74 | 74.38 ± 2.20 | 58.19 ± 1.26 | **76.89 ± 0.69** | **64.05 ± 0.34** | 71.60 | 72.87 | 3.12 |
|  | $L^2$-SP | 74.62 ± 4.99 | 68.30 ± 1.19 | 79.91 ± 2.29 | 73.97 ± 0.78 | 61.62 ± 2.07 | 59.78 ± 0.33 | 75.39 ± 0.51 | 62.34 ± 0.82 | 69.49 | 69.37 | 5.25 |
|  | FEATURE-MAP | 61.06 ± 2.00 | 65.12 ± 1.98 | **82.66 ± 0.62** | 74.54 ± 1.00 | 72.81 ± 1.16 | **60.47 ± 0.45** | 70.39 ± 0.11 | 60.10 ± 0.19 | 68.39 | 67.40 | 5.25 |
|  | BSS | 73.89 ± 6.04 | **70.04 ± 2.00** | 77.94 ± 2.04 | 76.28 ± 1.28 | **76.20 ± 1.33** | 59.99 ± 1.39 | 75.86 ± 1.08 | 63.62 ± 0.50 | 71.73 | 72.65 | 2.75 |
| SIZE | FULL-FT | 61.94 ± 2.67 | 82.80 ± 2.31 | 63.62 ± 1.19 | 77.81 ± 2.99 | 72.05 ± 2.96 | 54.92 ± 0.79 | 71.08 ± 0.77 | 62.47 ± 0.83 | 68.34 | 68.16 | 5.12 |
|  | LP | 55.54 ± 0.65 | 75.89 ± 0.90 | 42.31 ± 0.48 | 67.54 ± 1.27 | 69.87 ± 1.51 | 53.74 ± 0.43 | 68.10 ± 0.39 | 57.50 ± 0.19 | 61.31 | 62.05 | 7.75 |
|  | SURGICAL-FT | 64.54 ± 8.03 | **88.90 ± 0.74** | 61.99 ± 2.13 | 78.10 ± 0.96 | **76.07 ± 0.57** | **57.13 ± 1.87** | 72.24 ± 0.28 | 60.52 ± 0.95 | 69.94 | 68.91 | 2.50 |
|  | LP-FT | 63.79 ± 3.29 | 83.12 ± 5.20 | **65.48 ± 0.70** | 76.47 ± 3.53 | 72.24 ± 2.79 | 56.31 ± 0.72 | 72.98 ± 0.51 | 62.98 ± 0.51 | 68.97 | 68.72 | 3.75 |
|  | WISE-FT | 63.85 ± 3.69 | 81.81 ± 2.80 | 62.71 ± 1.26 | 77.83 ± 2.02 | 73.40 ± 2.08 | 56.63 ± 0.63 | 71.27 ± 0.77 | 62.70 ± 0.87 | 68.78 | 68.63 | 4.00 |
|  | $L^2$-SP | 63.67 ± 1.79 | 88.00 ± 1.00 | 63.98 ± 1.51 | 77.38 ± 1.25 | 58.29 ± 3.74 | 56.23 ± 1.70 | 71.93 ± 0.21 | 59.29 ± 0.72 | 67.35 | 65.76 | 4.50 |
|  | FEATURE-MAP | 64.41 ± 1.38 | 86.82 ± 0.76 | 59.62 ± 1.17 | 70.71 ± 0.99 | 76.01 ± 0.60 | 55.03 ± 0.30 | 67.98 ± 0.41 | 57.91 ± 0.31 | 67.31 | 66.11 | 5.25 |
|  | BSS | **67.80 ± 4.60** | 84.90 ± 2.20 | 62.77 ± 3.69 | **78.13 ± 2.21** | 74.58 ± 1.13 | 54.91 ± 1.34 | 71.40 ± 0.44 | **63.04 ± 0.35** | 69.69 | 69.62 | 3.12 |

Table 12: Robust fine-tuning performance on 5 classification datasets (AUC metrics) in the Few-shot 50 setting, evaluated across 3 dataset splits (RANDOM, SCAFFOLD, SIZE) under the MoleculeSTM pre-trained model. AVG, AVG-F, AVG-R denote the average AUC metrics, average AUC without max and min values, and average rank over all the datasets for each evaluated method, respectively. Standard deviations across five replicates are shown in parentheses. We **bold** and underline the best and second-best performances in each scenario.

| SPLIT | METHODS | CLINTOX | BBBP | BACE | HIV | SIDER | AVG | AVG-F | AVG-R |
|---|---|---|---|---|---|---|---|---|---|
| RANDOM | FULL-FT | 49.60 ± 2.85 | 84.86 ± 1.30 | 74.74 ± 1.44 | 60.58 ± 1.47 | 49.47 ± 0.90 | 63.85 | 61.64 | 4.80 |
|  | LP | 52.66 ± 3.14 | 78.85 ± 1.75 | 58.02 ± 3.19 | 52.39 ± 0.52 | 50.23 ± 0.47 | 58.43 | 54.36 | 6.40 |
|  | SURGICAL-FT | 54.43 ± 4.39 | **86.64 ± 0.96** | 74.92 ± 0.95 | **61.71 ± 0.64** | 51.10 ± 0.82 | 65.76 | 63.69 | 2.00 |
|  | LP-FT | 47.71 ± 2.16 | 84.36 ± 2.65 | 74.92 ± 0.95 | 55.82 ± 1.53 | **51.62 ± 0.37** | 62.89 | 60.79 | 4.60 |
|  | WISE-FT | **55.69 ± 5.37** | 84.62 ± 1.45 | 74.02 ± 1.36 | 60.05 ± 1.26 | 49.41 ± 0.89 | 64.76 | 63.25 | 4.60 |
|  | $L^2$-SP | 50.07 ± 2.37 | 85.69 ± 1.19 | **75.18 ± 1.16** | 58.44 ± 1.98 | 50.58 ± 0.93 | 63.99 | 61.40 | 3.60 |
|  | FEATURE-MAP | 54.09 ± 3.21 | 78.77 ± 4.05 | 67.88 ± 0.54 | 55.43 ± 1.21 | 50.12 ± 0.27 | 61.26 | 59.13 | 6.20 |
|  | BSS | 52.06 ± 3.58 | 85.62 ± 1.18 | 74.31 ± 1.83 | 58.90 ± 0.76 | 51.18 ± 0.69 | 64.41 | 61.76 | 3.80 |
| SCAFFOLD | FULL-FT | 45.62 ± 5.48 | **58.05 ± 2.70** | 62.30 ± 1.27 | 48.87 ± 6.91 | 54.88 ± 0.29 | 53.94 | 53.93 | 2.60 |
|  | LP | 30.76 ± 1.34 | 50.50 ± 1.35 | 56.94 ± 2.34 | 39.19 ± 1.21 | 53.17 ± 0.36 | 46.11 | 47.62 | 7.80 |
|  | SURGICAL-FT | 45.60 ± 9.96 | 56.02 ± 1.54 | 63.07 ± 0.78 | 44.00 ± 3.78 | 55.18 ± 0.47 | 52.77 | 52.27 | 3.80 |
|  | LP-FT | 33.97 ± 3.65 | 55.31 ± 2.06 | 61.87 ± 0.80 | 45.88 ± 1.92 | 55.16 ± 0.46 | 50.44 | 52.12 | 5.20 |
|  | WISE-FT | **47.69 ± 5.22** | 57.80 ± 2.92 | 62.06 ± 1.03 | 47.33 ± 5.84 | 55.16 ± 0.57 | 54.01 | 53.55 | 2.60 |
|  | $L^2$-SP | 45.54 ± 5.40 | 56.06 ± 1.99 | 61.75 ± 1.66 | 45.56 ± 4.10 | **55.29 ± 0.92** | 52.84 | 52.30 | 4.20 |
|  | FEATURE-MAP | 26.69 ± 2.38 | 56.71 ± 1.18 | 61.18 ± 5.30 | 43.71 ± 3.23 | 53.77 ± 0.39 | 48.41 | 51.40 | 6.60 |
|  | BSS | 42.19 ± 1.78 | 57.09 ± 1.32 | **63.74 ± 2.79** | **50.07 ± 8.79** | 54.75 ± 0.37 | 53.57 | 53.97 | 3.20 |
| SIZE | FULL-FT | 58.52 ± 2.98 | 58.80 ± 9.95 | 36.17 ± 6.29 | 52.04 ± 2.74 | 51.97 ± 1.34 | 51.50 | 54.18 | 4.20 |
|  | LP | 57.53 ± 4.82 | 45.54 ± 17.14 | 47.39 ± 1.62 | 48.21 ± 0.61 | 50.89 ± 0.73 | 49.91 | 48.83 | 6.60 |
|  | SURGICAL-FT | 61.32 ± 8.19 | 54.19 ± 11.51 | 44.96 ± 7.70 | 51.79 ± 2.35 | 51.41 ± 0.98 | 52.73 | 52.46 | 4.80 |
|  | LP-FT | 54.70 ± 9.04 | 55.56 ± 3.73 | 43.08 ± 1.91 | 47.90 ± 2.39 | 51.88 ± 0.55 | 50.62 | 51.49 | 5.80 |
|  | WISE-FT | 61.60 ± 5.18 | 56.83 ± 9.47 | 42.48 ± 6.40 | 50.61 ± 2.71 | **52.28 ± 1.23** | 52.76 | 53.24 | 3.80 |
|  | $L^2$-SP | 60.54 ± 2.21 | **62.77 ± 6.52** | 47.51 ± 8.30 | **52.06 ± 2.80** | 51.52 ± 1.67 | 54.88 | 54.71 | 2.60 |
|  | FEATURE-MAP | 59.85 ± 1.06 | 50.21 ± 1.87 | 47.65 ± 3.15 | 44.09 ± 1.27 | 51.48 ± 0.50 | 50.66 | 49.78 | 5.40 |
|  | BSS | **62.26 ± 1.89** | 60.79 ± 7.04 | 49.70 ± 2.37 | 51.85 ± 3.42 | 51.19 ± 1.56 | 55.16 | 54.61 | 2.80 |

Table 13: Robust fine-tuning performance on 5 classification datasets (AUC metrics) in the Few-shot 100 setting, evaluated across 3 dataset splits (RANDOM, SCAFFOLD, SIZE) under the MoleculeSTM pre-trained model. AVG, AVG-F, AVG-R denote the average AUC metrics, average AUC without max and min values, and average rank over all the datasets for each evaluated method, respectively. Standard deviations across five replicates are shown in parentheses. We **bold** and underline the best and second-best performances in each scenario.

| SPLIT | METHODS | CLINTOX | BBBP | BACE | HIV | SIDER | AVG | AVG-F | AVG-R |
|---|---|---|---|---|---|---|---|---|---|
| RANDOM | FULL-FT | 73.60 ± 7.53 | 82.09 ± 2.90 | 80.72 ± 1.22 | 61.92 ± 2.62 | 51.58 ± 0.43 | 69.98 | 72.08 | 5.00 |
| | LP | 69.43 ± 1.40 | 73.63 ± 0.97 | 60.60 ± 3.89 | 54.74 ± 0.90 | 53.47 ± 0.21 | 62.37 | 61.59 | 6.60 |
| | SURGICAL-FT | 71.20 ± 2.70 | 83.50 ± 0.95 | 80.44 ± 0.62 | 62.65 ± 1.44 | 53.43 ± 0.90 | 70.24 | 71.43 | 4.20 |
| | LP-FT | 68.16 ± 1.86 | **84.26 ± 1.37** | 79.93 ± 2.67 | 60.14 ± 3.04 | 52.18 ± 0.81 | 68.93 | 69.41 | 5.20 |
| | WISE-FT | 72.72 ± 8.35 | 83.52 ± 3.24 | **88.26 ± 1.45** | 62.19 ± 2.74 | 51.66 ± 0.43 | 71.67 | 72.81 | 3.80 |
| | $L^2$-SP | 73.05 ± 2.80 | 82.49 ± 1.95 | 81.60 ± 1.23 | 63.21 ± 2.21 | 53.92 ± 0.82 | 70.85 | 72.62 | 3.00 |
| | FEATURE-MAP | 68.01 ± 2.06 | 78.35 ± 0.58 | 69.27 ± 0.87 | 58.07 ± 1.89 | **54.33 ± 0.73** | 65.61 | 65.12 | 6.00 |
| | BSS | **76.21 ± 6.50** | 83.52 ± 1.90 | 81.69 ± 0.40 | **63.54 ± 2.05** | 53.26 ± 0.84 | 71.64 | 73.81 | 2.20 |
| SCAFFOLD | FULL-FT | 54.76 ± 2.86 | 56.25 ± 1.78 | 64.85 ± 1.26 | 56.18 ± 6.68 | 55.07 ± 1.47 | 57.42 | 55.83 | 4.20 |
| | LP | 49.89 ± 3.86 | 48.69 ± 1.72 | 60.40 ± 2.76 | 40.97 ± 1.51 | 52.98 ± 0.26 | 50.59 | 50.52 | 7.40 |
| | SURGICAL-FT | 56.64 ± 4.28 | 54.30 ± 2.39 | 66.81 ± 0.67 | 53.60 ± 2.54 | 55.29 ± 0.58 | 57.33 | 55.41 | 4.20 |
| | LP-FT | 49.82 ± 6.97 | 52.74 ± 3.13 | 64.81 ± 3.24 | 57.02 ± 4.98 | **57.58 ± 0.29** | 56.39 | 55.78 | 4.40 |
| | WISE-FT | **58.53 ± 5.22** | 56.16 ± 1.85 | 64.17 ± 1.08 | 53.49 ± 6.18 | 55.11 ± 1.23 | 57.49 | 56.60 | 4.40 |
| | $L^2$-SP | 57.60 ± 4.63 | 57.53 ± 1.08 | 64.50 ± 1.83 | **59.39 ± 3.16** | 57.05 ± 1.02 | 59.21 | 58.17 | 2.60 |
| | FEATURE-MAP | 44.86 ± 3.28 | 55.25 ± 0.79 | 57.69 ± 5.35 | 45.60 ± 4.50 | 54.00 ± 0.88 | 51.48 | 51.62 | 7.00 |
| | BSS | 58.38 ± 5.39 | **58.27 ± 0.49** | **70.00 ± 2.70** | 58.52 ± 2.49 | 56.50 ± 1.02 | 60.33 | 58.39 | 1.80 |
| SIZE | FULL-FT | 70.85 ± 5.54 | 75.13 ± 3.96 | 54.43 ± 3.01 | 60.05 ± 6.91 | 52.07 ± 1.73 | 62.51 | 61.78 | 5.20 |
| | LP | 58.36 ± 3.23 | 56.25 ± 8.75 | 43.06 ± 1.32 | 45.90 ± 2.48 | 52.35 ± 0.37 | 51.18 | 51.50 | 7.60 |
| | SURGICAL-FT | 67.51 ± 7.23 | 81.75 ± 2.07 | 60.97 ± 1.53 | 62.45 ± 1.60 | 54.19 ± 0.38 | 65.37 | 63.64 | 3.00 |
| | LP-FT | 67.07 ± 2.45 | **82.12 ± 3.68** | 57.30 ± 2.65 | **65.84 ± 5.10** | 53.10 ± 0.96 | 65.09 | 63.40 | 3.20 |
| | WISE-FT | 70.06 ± 5.49 | 73.88 ± 4.80 | 52.09 ± 3.06 | 56.91 ± 5.90 | 54.21 ± 0.75 | 61.43 | 60.39 | 4.80 |
| | $L^2$-SP | 65.62 ± 4.40 | 79.46 ± 0.79 | 55.84 ± 4.07 | 63.81 ± 7.20 | 53.82 ± 1.27 | 63.71 | 61.76 | 4.40 |
| | FEATURE-MAP | 65.63 ± 1.73 | 70.03 ± 3.19 | **63.06 ± 1.89** | 45.09 ± 2.28 | **55.32 ± 0.92** | 59.83 | 61.34 | 4.60 |
| | BSS | **70.90 ± 2.39** | 77.56 ± 2.51 | 59.84 ± 4.41 | 65.31 ± 6.67 | 52.59 ± 1.16 | 65.24 | 65.35 | 3.20 |

Table 14: Robust fine-tuning performance on 5 classification datasets (AUC metrics) in the Few-shot 500 setting, evaluated across 3 dataset splits (RANDOM, SCAFFOLD, SIZE) under the MoleculeSTM pre-trained model. AVG, AVG-F, AVG-R denote the average AUC metrics, average AUC without max and min values, and average rank over all the datasets for each evaluated method, respectively. Standard deviations across five replicates are shown in parentheses. We **bold** and underline the best and second-best performances in each scenario.

| SPLIT | METHODS | CLINTOX | BBBP | BACE | HIV | SIDER | AVG | AVG-F | AVG-R |
|---|---|---|---|---|---|---|---|---|---|
| RANDOM | FULL-FT | **85.93 ± 2.06** | 91.93 ± 0.96 | 83.67 ± 0.92 | 69.71 ± 1.63 | 58.42 ± 2.20 | 77.93 | 79.77 | 3.20 |
| | LP | 76.92 ± 0.43 | 85.18 ± 0.26 | 70.83 ± 0.51 | 64.43 ± 0.53 | 56.80 ± 0.21 | 70.83 | 70.73 | 8.00 |
| | SURGICAL-FT | 83.62 ± 1.90 | 91.68 ± 0.46 | **86.18 ± 0.83** | 68.37 ± 0.74 | **60.29 ± 0.87** | 78.03 | 79.39 | 3.40 |
| | LP-FT | 81.89 ± 2.72 | 90.93 ± 2.04 | 83.92 ± 0.84 | 68.20 ± 1.53 | 58.56 ± 0.71 | 76.70 | 78.00 | 5.80 |
| | WISE-FT | 85.10 ± 2.16 | 91.53 ± 1.15 | 84.19 ± 0.86 | 69.60 ± 1.37 | 58.25 ± 2.04 | 77.73 | 79.63 | 4.20 |
| | $L^2$-SP | 84.17 ± 3.97 | **92.19 ± 1.11** | 84.82 ± 0.95 | **70.06 ± 0.93** | 59.31 ± 0.96 | 78.11 | 79.68 | 2.00 |
| | FEATURE-MAP | 83.37 ± 1.03 | 88.80 ± 0.29 | 79.88 ± 0.14 | 69.38 ± 0.54 | 57.64 ± 0.65 | 75.81 | 77.54 | 6.40 |
| | BSS | 85.84 ± 1.94 | 91.81 ± 0.80 | 84.68 ± 0.83 | 69.38 ± 1.98 | 58.85 ± 1.05 | 78.11 | 79.97 | 3.00 |
| SCAFFOLD | FULL-FT | 63.02 ± 3.19 | 64.84 ± 1.51 | 71.94 ± 2.43 | 68.53 ± 2.78 | 56.27 ± 0.94 | 64.92 | 65.46 | 5.60 |
| | LP | 56.80 ± 1.80 | 58.21 ± 0.93 | 67.33 ± 0.37 | 53.12 ± 1.19 | 56.58 ± 0.58 | 58.41 | 57.20 | 7.20 |
| | SURGICAL-FT | **69.47 ± 3.18** | 65.26 ± 0.62 | **76.72 ± 1.60** | 69.94 ± 2.17 | 55.72 ± 0.55 | 67.42 | 68.22 | 3.00 |
| | LP-FT | 65.09 ± 3.54 | 64.23 ± 1.67 | 69.36 ± 2.11 | 69.41 ± 1.48 | 57.33 ± 0.44 | 65.08 | 66.23 | 4.60 |
| | WISE-FT | 64.89 ± 4.07 | 64.85 ± 1.47 | 71.94 ± 2.08 | 69.00 ± 2.32 | 56.23 ± 0.76 | 65.38 | 66.25 | 5.00 |
| | $L^2$-SP | 69.03 ± 2.49 | 66.06 ± 1.43 | 74.07 ± 1.26 | 67.67 ± 2.21 | 56.42 ± 0.97 | 66.65 | 67.59 | 3.80 |
| | FEATURE-MAP | 60.04 ± 3.11 | 63.87 ± 0.70 | 75.42 ± 0.70 | 60.08 ± 2.03 | **58.45 ± 0.38** | 63.57 | 61.33 | 4.80 |
| | BSS | 68.30 ± 2.86 | **67.26 ± 0.98** | 74.83 ± 2.15 | **69.99 ± 1.80** | 57.43 ± 0.73 | 67.56 | 68.52 | 2.00 |
| SIZE | FULL-FT | 60.10 ± 5.25 | 76.35 ± 2.26 | 50.25 ± 3.29 | **5623 ± 5.29** | 54.40 ± 1.70 | 1172.82 | 63.62 | 4.80 |
| | LP | 59.95 ± 0.51 | 63.98 ± 1.71 | 40.46 ± 4.26 | 58.26 ± 7.53 | 51.43 ± 0.20 | 54.82 | 56.55 | 7.60 |
| | SURGICAL-FT | 61.92 ± 5.41 | **86.62 ± 1.84** | 51.72 ± 2.80 | 58.76 ± 3.21 | 56.61 ± 1.07 | 63.13 | 59.10 | 3.20 |
| | LP-FT | 55.39 ± 4.42 | 78.83 ± 7.22 | **53.66 ± 3.35** | 62.85 ± 4.81 | 55.21 ± 1.62 | 61.19 | 57.82 | 4.80 |
| | WISE-FT | 62.14 ± 1.97 | 75.21 ± 2.23 | 48.40 ± 2.94 | 53.63 ± 3.76 | 56.19 ± 1.22 | 59.11 | 57.32 | 5.80 |
| | $L^2$-SP | **64.97 ± 0.50** | 83.22 ± 1.87 | 51.14 ± 4.26 | 69.62 ± 3.36 | **56.72 ± 1.04** | 65.13 | 63.77 | 2.00 |
| | FEATURE-MAP | 63.06 ± 1.12 | 80.15 ± 1.70 | 43.45 ± 0.50 | 66.24 ± 0.37 | 53.29 ± 0.71 | 61.24 | 60.86 | 4.80 |
| | BSS | 62.87 ± 5.70 | 80.69 ± 2.55 | 51.61 ± 4.52 | 67.37 ± 4.52 | 56.48 ± 2.00 | 63.80 | 62.24 | 3.00 |

Table 15: Robust fine-tuning performance on 4 regression datasets (RMSE metrics) in the Non-Fewshot settings, evaluated across 3 dataset splits (RANDOM, SCAFFOLD, SIZE) over the Graph-MAE based PT model. AVG-R,AVG-R$^*$ denote the average rank and the rank based on the average normalized performance over all the datasets for each evavluated method, respectively. Standard deviations across five replicates are shown in parentheses. We **bold** and underline the best and second-best performances in each scenario.

| SPLIT | METHODS | ESOL | LIPO | MALARIA | CEP | AVG-R | AVG-R$^*$ |
|---|---|---|---|---|---|---|---|
| | FULL-FT | $0.901 \pm 0.063$ | $0.660 \pm 0.013$ | $1.067 \pm 0.009$ | $1.401 \pm 0.035$ | 3.00 | 2 |
| | LP | $1.374 \pm 0.011$ | $1.067 \pm 0.015$ | $1.207 \pm 0.004$ | $1.999 \pm 0.003$ | 8.00 | 8 |
| | SURGICAL-FT | $1.056 \pm 0.028$ | $0.724 \pm 0.011$ | $1.074 \pm 0.010$ | $1.547 \pm 0.011$ | 6.00 | 6 |
| RANDOM | LP-FT | $0.922 \pm 0.023$ | $0.654 \pm 0.023$ | $1.076 \pm 0.014$ | $1.365 \pm 0.029$ | 3.25 | 3 |
| | WISE-FT | $0.934 \pm 0.061$ | $0.662 \pm 0.016$ | **$1.064 \pm 0.007$** | $1.460 \pm 0.042$ | 3.75 | 5 |
| | $L^2$-SP | **$0.884 \pm 0.025$** | $0.666 \pm 0.014$ | $1.087 \pm 0.011$ | $1.385 \pm 0.031$ | 3.75 | 4 |
| | FEATURE-MAP | $1.018 \pm 0.024$ | $0.789 \pm 0.018$ | $1.106 \pm 0.005$ | $1.536 \pm 0.008$ | 6.50 | 7 |
| | BSS | $0.887 \pm 0.030$ | **$0.641 \pm 0.014$** | $1.070 \pm 0.016$ | **$1.351 \pm 0.016$** | 1.75 | 1 |
| | FULL-FT | $1.360 \pm 0.049$ | $0.752 \pm 0.018$ | $1.105 \pm 0.018$ | $1.395 \pm 0.041$ | 4.50 | 5 |
| | LP | $1.608 \pm 0.030$ | $0.983 \pm 0.006$ | $1.133 \pm 0.002$ | $2.009 \pm 0.004$ | 8.00 | 8 |
| | SURGICAL-FT | **$1.297 \pm 0.044$** | $0.765 \pm 0.013$ | $1.105 \pm 0.013$ | $1.518 \pm 0.010$ | 4.50 | 6 |
| SCAFFOLD | LP-FT | $1.331 \pm 0.033$ | $0.743 \pm 0.017$ | $1.107 \pm 0.011$ | $1.356 \pm 0.030$ | 4.00 | 4 |
| | WISE-FT | $1.347 \pm 0.036$ | **$0.740 \pm 0.018$** | $1.090 \pm 0.015$ | $1.505 \pm 0.045$ | 3.00 | 2 |
| | $L^2$-SP | $1.300 \pm 0.017$ | $0.756 \pm 0.017$ | $1.106 \pm 0.005$ | $1.347 \pm 0.020$ | 3.75 | 3 |
| | FEATURE-MAP | $1.383 \pm 0.008$ | $0.824 \pm 0.009$ | $1.098 \pm 0.004$ | $1.518 \pm 0.003$ | 6.00 | 7 |
| | BSS | $1.300 \pm 0.024$ | $0.746 \pm 0.010$ | $1.097 \pm 0.013$ | **$1.319 \pm 0.023$** | 2.25 | 1 |
| | FULL-FT | $1.490 \pm 0.153$ | $0.711 \pm 0.017$ | **$0.883 \pm 0.008$** | $1.834 \pm 0.038$ | 3.25 | 2 |
| | LP | $2.172 \pm 0.065$ | $0.935 \pm 0.004$ | $0.912 \pm 0.004$ | $2.402 \pm 0.018$ | 8.00 | 8 |
| | SURGICAL-FT | $1.499 \pm 0.093$ | $0.769 \pm 0.013$ | $0.889 \pm 0.014$ | $1.998 \pm 0.020$ | 5.25 | 6 |
| SIZE | LP-FT | $1.401 \pm 0.053$ | $0.703 \pm 0.012$ | $0.897 \pm 0.009$ | $1.763 \pm 0.037$ | 3.25 | 3 |
| | WISE-FT | $1.583 \pm 0.118$ | $0.727 \pm 0.018$ | $0.889 \pm 0.008$ | $1.902 \pm 0.053$ | 5.25 | 5 |
| | $L^2$-SP | **$1.390 \pm 0.115$** | $0.725 \pm 0.019$ | $0.896 \pm 0.007$ | $1.786 \pm 0.022$ | 3.25 | 4 |
| | FEATURE-MAP | $1.458 \pm 0.045$ | $0.849 \pm 0.012$ | $0.896 \pm 0.011$ | $2.007 \pm 0.018$ | 6.00 | 7 |
| | BSS | $1.408 \pm 0.100$ | **$0.700 \pm 0.020$** | $0.887 \pm 0.011$ | **$1.725 \pm 0.026$** | 1.75 | 1 |

Table 16: Robust fine-tuning performance on 4 regression datasets (RMSE metrics) in the Few-shot 50 setting, evaluated across 3 dataset splits (RANDOM, SCAFFOLD, SIZE) over the Graph-MAE based PT model. AVG-R,AVG-R$^*$ denote the average rank and the rank based on the average normalized performance over all the datasets for each evavluated method, respectively. Standard deviations across five replicates are shown in parentheses. We **bold** and underline the best and second-best performances in each scenario.

| SPLIT | METHODS | ESOL | LIPO | MALARIA | CEP | AVG-R | AVG-R$^*$ |
|---|---|---|---|---|---|---|---|
| | FULL-FT | $2.128 \pm 0.072$ | $1.247 \pm 0.031$ | $1.310 \pm 0.025$ | $3.433 \pm 0.226$ | 5.00 | 6 |
| | LP | $2.971 \pm 0.017$ | $1.638 \pm 0.014$ | $1.309 \pm 0.012$ | $3.519 \pm 0.052$ | 6.75 | 8 |
| | SURGICAL-FT | $2.315 \pm 0.081$ | $1.327 \pm 0.017$ | $1.317 \pm 0.024$ | $3.272 \pm 0.199$ | 6.50 | 7 |
| RANDOM | LP-FT | $1.600 \pm 0.129$ | $1.181 \pm 0.030$ | $1.356 \pm 0.011$ | $2.358 \pm 0.037$ | 4.25 | 4 |
| | WISE-FT | $2.135 \pm 0.072$ | $1.261 \pm 0.035$ | $1.298 \pm 0.023$ | $3.576 \pm 0.235$ | 5.50 | 5 |
| | $L^2$-SP | $1.472 \pm 0.036$ | **$1.165 \pm 0.037$** | **$1.297 \pm 0.006$** | $2.304 \pm 0.055$ | 1.50 | 1 |
| | FEATURE-MAP | $1.632 \pm 0.028$ | $1.257 \pm 0.025$ | $1.301 \pm 0.009$ | $2.398 \pm 0.037$ | 4.00 | 3 |
| | BSS | **$1.450 \pm 0.057$** | $1.171 \pm 0.021$ | $1.314 \pm 0.018$ | **$2.244 \pm 0.036$** | 2.50 | 2 |
| | FULL-FT | $2.790 \pm 0.116$ | $1.434 \pm 0.072$ | $1.195 \pm 0.025$ | $3.395 \pm 0.191$ | 5.75 | 6 |
| | LP | $3.538 \pm 0.075$ | $1.755 \pm 0.021$ | $1.206 \pm 0.012$ | $3.870 \pm 0.038$ | 7.75 | 8 |
| | SURGICAL-FT | $3.018 \pm 0.118$ | $1.491 \pm 0.085$ | $1.191 \pm 0.004$ | $3.304 \pm 0.347$ | 5.75 | 7 |
| SCAFFOLD | LP-FT | $1.636 \pm 0.021$ | $1.181 \pm 0.029$ | $1.263 \pm 0.009$ | $2.294 \pm 0.024$ | 4.00 | 4 |
| | WISE-FT | $2.762 \pm 0.091$ | $1.405 \pm 0.067$ | **$1.181 \pm 0.008$** | $3.496 \pm 0.199$ | 4.50 | 5 |
| | $L^2$-SP | $1.654 \pm 0.086$ | $1.178 \pm 0.022$ | $1.185 \pm 0.008$ | **$2.255 \pm 0.026$** | 2.25 | 2 |
| | FEATURE-MAP | $1.783 \pm 0.034$ | $1.252 \pm 0.012$ | $1.195 \pm 0.008$ | $2.401 \pm 0.028$ | 4.50 | 3 |
| | BSS | **$1.632 \pm 0.048$** | **$1.173 \pm 0.022$** | $1.182 \pm 0.016$ | $2.287 \pm 0.028$ | 1.50 | 1 |
| | FULL-FT | $3.457 \pm 0.086$ | $1.407 \pm 0.088$ | $1.064 \pm 0.067$ | $3.311 \pm 0.158$ | 6.25 | 7 |
| | LP | $3.758 \pm 0.010$ | $1.773 \pm 0.025$ | $0.990 \pm 0.056$ | $4.114 \pm 0.042$ | 6.75 | 8 |
| | SURGICAL-FT | $3.429 \pm 0.139$ | $1.543 \pm 0.083$ | $0.990 \pm 0.054$ | $3.195 \pm 0.306$ | 5.25 | 6 |
| SIZE | LP-FT | **$2.035 \pm 0.080$** | $1.208 \pm 0.078$ | $1.102 \pm 0.018$ | $2.500 \pm 0.045$ | 4.00 | 4 |
| | WISE-FT | $3.527 \pm 0.112$ | $1.392 \pm 0.062$ | **$0.983 \pm 0.053$** | $3.386 \pm 0.142$ | 5.00 | 5 |
| | $L^2$-SP | $2.111 \pm 0.091$ | $1.159 \pm 0.037$ | $0.988 \pm 0.032$ | $2.421 \pm 0.045$ | 2.00 | 1 |
| | FEATURE-MAP | $2.331 \pm 0.050$ | $1.225 \pm 0.049$ | $1.000 \pm 0.034$ | $2.439 \pm 0.024$ | 4.00 | 3 |
| | BSS | $2.197 \pm 0.084$ | **$1.106 \pm 0.027$** | $1.019 \pm 0.033$ | **$2.419 \pm 0.045$** | 2.75 | 2 |

Table 17: Robust fine-tuning performance on 4 regression datasets (RMSE metrics) in the Few-shot 100 setting, evaluated across 3 dataset splits (RANDOM, SCAFFOLD, SIZE) over the Graph-MAE based PT model. AVG-R, AVG-R$^*$ denote the average rank and the rank based on the average normalized performance over all the datasets for each evavluated method, respectively. Standard deviations across five replicates are shown in parentheses. We **bold** and underline the best and second-best performances in each scenario.

| SPLIT | METHODS | ESOL | LIPO | MALARIA | CEP | AVG-R | AVG-R$^*$ |
|---|---|---|---|---|---|---|---|
| RANDOM | FULL-FT | $1.842 \pm 0.208$ | $1.205 \pm 0.059$ | $1.289 \pm 0.032$ | $2.784 \pm 0.110$ | 5.75 | 6 |
| | LP | $2.391 \pm 0.044$ | $1.623 \pm 0.011$ | $1.279 \pm 0.007$ | $3.176 \pm 0.093$ | 7.00 | 8 |
| | SURGICAL-FT | $1.650 \pm 0.063$ | $1.301 \pm 0.037$ | $1.277 \pm 0.012$ | $2.777 \pm 0.181$ | 5.00 | 4 |
| | LP-FT | $\underline{1.540 \pm 0.123}$ | $1.234 \pm 0.030$ | $1.350 \pm 0.016$ | $2.203 \pm 0.030$ | 4.50 | 7 |
| | WISE-FT | $1.790 \pm 0.147$ | $1.207 \pm 0.058$ | $1.282 \pm 0.017$ | $2.842 \pm 0.123$ | 5.50 | 5 |
| | $L^2$-SP | $\mathbf{1.486 \pm 0.105}$ | $\mathbf{1.190 \pm 0.038}$ | $\mathbf{1.267 \pm 0.007}$ | $2.207 \pm 0.046$ | 1.75 | 1 |
| | FEATURE-MAP | $1.557 \pm 0.034$ | $1.252 \pm 0.007$ | $\underline{1.269 \pm 0.002}$ | $\mathbf{2.130 \pm 0.020}$ | 3.25 | 2 |
| | BSS | $1.543 \pm 0.044$ | $\mathbf{1.190 \pm 0.031}$ | $\underline{1.285 \pm 0.011}$ | $\underline{2.170 \pm 0.028}$ | 3.25 | 3 |
| SCAFFOLD | FULL-FT | $2.036 \pm 0.119$ | $1.108 \pm 0.017$ | $1.205 \pm 0.050$ | $2.942 \pm 0.208$ | 5.75 | 6 |
| | LP | $2.906 \pm 0.093$ | $1.389 \pm 0.008$ | $1.180 \pm 0.017$ | $3.635 \pm 0.051$ | 6.75 | 8 |
| | SURGICAL-FT | $1.956 \pm 0.170$ | $1.190 \pm 0.027$ | $1.183 \pm 0.016$ | $2.848 \pm 0.120$ | 5.50 | 5 |
| | LP-FT | $1.775 \pm 0.178$ | $1.103 \pm 0.024$ | $1.288 \pm 0.012$ | $2.310 \pm 0.034$ | 4.75 | 7 |
| | WISE-FT | $2.052 \pm 0.082$ | $1.112 \pm 0.023$ | $1.188 \pm 0.027$ | $3.049 \pm 0.246$ | 6.25 | 4 |
| | $L^2$-SP | $\mathbf{1.559 \pm 0.047}$ | $\mathbf{1.069 \pm 0.044}$ | $\underline{1.166 \pm 0.004}$ | $2.227 \pm 0.036$ | 1.75 | 1 |
| | FEATURE-MAP | $\underline{1.576 \pm 0.028}$ | $1.123 \pm 0.009$ | $1.181 \pm 0.005$ | $\underline{2.216 \pm 0.014}$ | 3.50 | 3 |
| | BSS | $1.680 \pm 0.098$ | $\underline{1.081 \pm 0.019}$ | $\mathbf{1.163 \pm 0.004}$ | $\mathbf{2.212 \pm 0.018}$ | 1.75 | 2 |
| SIZE | FULL-FT | $2.527 \pm 0.152$ | $1.113 \pm 0.054$ | $1.022 \pm 0.046$ | $2.587 \pm 0.100$ | 6.25 | 7 |
| | LP | $3.020 \pm 0.061$ | $1.492 \pm 0.039$ | $0.951 \pm 0.011$ | $3.408 \pm 0.041$ | 6.75 | 8 |
| | SURGICAL-FT | $2.435 \pm 0.119$ | $1.119 \pm 0.037$ | $0.970 \pm 0.020$ | $2.607 \pm 0.040$ | 6.25 | 6 |
| | LP-FT | $1.937 \pm 0.120$ | $\mathbf{1.050 \pm 0.052}$ | $1.045 \pm 0.012$ | $2.506 \pm 0.042$ | 4.25 | 5 |
| | WISE-FT | $2.580 \pm 0.096$ | $1.086 \pm 0.051$ | $0.962 \pm 0.043$ | $2.556 \pm 0.089$ | 5.00 | 4 |
| | $L^2$-SP | $\underline{1.860 \pm 0.183}$ | $\underline{1.063 \pm 0.006}$ | $\mathbf{0.931 \pm 0.007}$ | $\underline{2.436 \pm 0.043}$ | 1.75 | 1 |
| | FEATURE-MAP | $1.921 \pm 0.086$ | $1.098 \pm 0.036$ | $\underline{0.936 \pm 0.009}$ | $\mathbf{2.374 \pm 0.011}$ | 2.75 | 2 |
| | BSS | $\mathbf{1.854 \pm 0.109}$ | $1.075 \pm 0.032$ | $0.962 \pm 0.017$ | $2.444 \pm 0.014$ | 3.00 | 3 |

Table 18: Robust fine-tuning performance on 4 regression datasets (RMSE metrics) in the Few-shot 500 setting, evaluated across 3 dataset splits (RANDOM, SCAFFOLD, SIZE) over the Graph-MAE based PT model. AVG-R, AVG-R$^*$ denote the average rank and the rank based on the average normalized performance over all the datasets for each evavluated method, respectively. Standard deviations across five replicates are shown in parentheses. We **bold** and underline the best and second-best performances in each scenario.

| SPLIT | METHODS | ESOL | LIPO | MALARIA | CEP | AVG-R | AVG-R$^*$ |
|---|---|---|---|---|---|---|---|
| RANDOM | FULL-FT | $1.093 \pm 0.085$ | $0.834 \pm 0.014$ | $1.245 \pm 0.018$ | $1.874 \pm 0.042$ | 5.00 | 6 |
| | LP | $1.542 \pm 0.011$ | $1.136 \pm 0.006$ | $1.253 \pm 0.003$ | $2.435 \pm 0.019$ | 8.00 | 8 |
| | SURGICAL-FT | $1.177 \pm 0.043$ | $0.888 \pm 0.010$ | $1.233 \pm 0.009$ | $1.948 \pm 0.005$ | 6.00 | 7 |
| | LP-FT | $1.001 \pm 0.020$ | $0.838 \pm 0.020$ | $1.244 \pm 0.011$ | $1.850 \pm 0.019$ | 4.00 | 5 |
| | WISE-FT | $1.076 \pm 0.074$ | $\underline{0.833 \pm 0.007}$ | $1.236 \pm 0.012$ | $1.898 \pm 0.051$ | 4.25 | 4 |
| | $L^2$-SP | $\underline{0.992 \pm 0.034}$ | $0.838 \pm 0.009$ | $\underline{1.225 \pm 0.005}$ | $\underline{1.839 \pm 0.024}$ | 2.75 | 1 |
| | FEATURE-MAP | $1.070 \pm 0.020$ | $0.948 \pm 0.010$ | $\mathbf{1.216 \pm 0.002}$ | $1.904 \pm 0.003$ | 4.50 | 3 |
| | BSS | $\mathbf{0.990 \pm 0.046}$ | $\mathbf{0.829 \pm 0.018}$ | $1.231 \pm 0.009$ | $\mathbf{1.835 \pm 0.023}$ | 1.50 | 2 |
| SCAFFOLD | FULL-FT | $1.434 \pm 0.044$ | $0.885 \pm 0.028$ | $1.186 \pm 0.017$ | $1.910 \pm 0.022$ | 5.00 | 6 |
| | LP | $2.047 \pm 0.020$ | $1.026 \pm 0.003$ | $1.168 \pm 0.005$ | $2.572 \pm 0.018$ | 7.25 | 8 |
| | SURGICAL-FT | $\mathbf{1.323 \pm 0.053}$ | $0.940 \pm 0.016$ | $1.159 \pm 0.014$ | $1.920 \pm 0.010$ | 4.50 | 5 |
| | LP-FT | $1.394 \pm 0.025$ | $0.888 \pm 0.017$ | $1.204 \pm 0.015$ | $1.876 \pm 0.024$ | 5.00 | 7 |
| | WISE-FT | $1.423 \pm 0.032$ | $0.885 \pm 0.023$ | $1.170 \pm 0.014$ | $1.926 \pm 0.035$ | 5.50 | 4 |
| | $L^2$-SP | $1.375 \pm 0.030$ | $\mathbf{0.879 \pm 0.008}$ | $\underline{1.139 \pm 0.001}$ | $\underline{1.870 \pm 0.032}$ | 1.75 | 1 |
| | FEATURE-MAP | $1.453 \pm 0.028$ | $0.903 \pm 0.004$ | $1.154 \pm 0.003$ | $1.913 \pm 0.016$ | 5.25 | 3 |
| | BSS | $\underline{1.367 \pm 0.043}$ | $\underline{0.881 \pm 0.024}$ | $1.150 \pm 0.020$ | $\mathbf{1.866 \pm 0.018}$ | 1.75 | 2 |
| SIZE | FULL-FT | $1.797 \pm 0.088$ | $0.793 \pm 0.019$ | $0.997 \pm 0.019$ | $2.353 \pm 0.033$ | 5.50 | 7 |
| | LP | $2.581 \pm 0.049$ | $1.030 \pm 0.004$ | $0.943 \pm 0.005$ | $2.990 \pm 0.030$ | 6.75 | 8 |
| | SURGICAL-FT | $\mathbf{1.540 \pm 0.078}$ | $0.846 \pm 0.011$ | $0.944 \pm 0.010$ | $2.403 \pm 0.038$ | 4.50 | 4 |
| | LP-FT | $1.717 \pm 0.077$ | $0.809 \pm 0.004$ | $0.956 \pm 0.014$ | $\underline{2.287 \pm 0.043}$ | 4.50 | 5 |
| | WISE-FT | $1.874 \pm 0.084$ | $0.805 \pm 0.012$ | $0.955 \pm 0.019$ | $2.363 \pm 0.035$ | 5.50 | 6 |
| | $L^2$-SP | $1.592 \pm 0.089$ | $\underline{0.788 \pm 0.014}$ | $\underline{0.930 \pm 0.008}$ | $2.297 \pm 0.014$ | 2.75 | 1 |
| | FEATURE-MAP | $\underline{1.580 \pm 0.070}$ | $0.873 \pm 0.016$ | $\mathbf{0.921 \pm 0.002}$ | $\mathbf{2.286 \pm 0.036}$ | 2.75 | 2 |
| | BSS | $1.617 \pm 0.117$ | $\mathbf{0.783 \pm 0.018}$ | $0.957 \pm 0.007$ | $2.295 \pm 0.038$ | 3.75 | 3 |

