# OpenReview forum: "RoFt-Mol: Benchmarking Robust Fine-tuning with Molecular Graph Foundation Models"
_ICLR.cc/2025/Conference — Submitted to ICLR 2025_

### Official Review · Reviewer_iuUw · 2024-10-16

**Soundness:** 3
**Presentation:** 2
**Contribution:** 3
**Rating:** 5
**Confidence:** 4

**Summary:**

This paper addresses the challenges and importance of fine-tuning pre-trained molecular graph models. The work explores 8 fine-tuning methods categorized into weight-based, representation-based, and partial FT, benchmarked across 12 datasets with 36 experimental settings. These settings aim to simulate real-world FT scenarios, including OOD generalization and Fewshot settings. The paper highlights the strengths of different FT strategies depending on whether tasks are classification or regression-based. A new method, DWiSE-FT, is proposed, combining aspects of weight-based methods to enhance fine-tuning efficiency.

**Strengths:**

1. The paper benchmarks 8 FT methods across 12 datasets and various experimental settings, providing a thorough evaluation of the fine-tuning methods for molecular graph models.

2. This paper proposes DWiSE-FT based on their findings, and DWiSE-FT performs well on regression tasks.

3. The benchmark is similar to the practical scenarios of molecular representation learning, by including scaffold and size splitting.

**Weaknesses:**

1. The pre-trained molecular models evaluated in this paper, such as Mole-BERT and Graph MAE are small in scale compared to other foundation models. The authors should evaluate more powerful pre-trained models such as [1] and [2].

2. Though DWiSE-FT shows improved performance, it is based on existing FT strategies rather than novel FT methods.

**Questions:**

1.  Recently, many multi-modal molecular graph models have been proposed. These models can gain additional information from texts and thus are more expressive. Can you conduct experiments on these more expressive molecular graph models such as [1] and [2]?

2. How do scaffold and size splits, which simulate OOD challenges, impact the robustness of fine-tuning methods in molecular property prediction tasks?

[1] MolCA: Molecular Graph-Language Modeling with Cross-Modal Projector and Uni-Modal Adapter
[2] Multi-modal Molecule Structure-text Model for Text-based Retrieval and Editing

---

> ### Author Response · Authors · 2024-11-22
>
> We thank reviewer iuUw for appreciating that our benchmark aligns with practical scenarios and recognizing our contribution in method introduction. We will respond to the questions as follows.
>
> > W1 / Q1: More powerful pre-trained model
>
> Regarding the Mole-BERT and GraphMAE, we directly use the checkpoints from existing works as improving pre-training methods is beyond the scope of this work. And selecting more complex backbones presents uncertainty about whether they can be effectively pre-trained using current pre-training methods.
>
> Thank you for suggesting the inclusion of additional pre-trained models. We have added MoleculeSTM [1] as a pre-trained model for our downstream fine-tuning evaluations and have discussed related multi-modal molecular graph models to our revised draft. We are currently benchmarking the results and aim to incorporate the available findings into the new paper draft by the end of the rebuttal period. However, due to time constraints and computational costs, some benchmarking results on the large MUV and HIV datasets may not be ready by then; we plan to include any remaining findings in a subsequent version of the paper.
>
> [1]. Liu, Shengchao, et al. "Multi-modal molecule structure–text model for text-based retrieval and editing." Nature Machine Intelligence 5.12 (2023): 1447-1457.
>
> > Q2: How do scaffold and size split impact the robustness of fine-tuning methods in molecular property prediction tasks?
>
> In general, scaffold and size splits introduce larger distribution shifts compared to random splits, making them more receptive to advanced fine-tuning strategies beyond basic vanilla fine-tuning. Between scaffold and size splits, scaffold splitting calls for more specialized fine-tuning techniques, resulting in significant performance gains over baseline methods, particularly in severe few-shot scenarios like few-shot 50. As discussed in Finding 5, size splits are notably vulnerable under linear probing with fixed representations and a tunable prediction head. The prediction head tends to overfit to the specific mapping from representations to output labels for molecules within a certain size range, impairing its ability to generalize to out-of-distribution molecules of differing sizes.

---

> > ### Comment · Reviewer_iuUw · 2024-11-25
> > **Response by Reviewer**
> >
> > Thanks for your time and effort. However, I am still concerned about the innovation of this work. So I will keep the score.

---

### Official Review · Reviewer_CyuA · 2024-11-01

**Soundness:** 3
**Presentation:** 3
**Contribution:** 3
**Rating:** 6
**Confidence:** 3

**Summary:**

This paper presents a benchmarking study on robust fine-tuning (FT) with molecular graph foundation models. The authors evaluate different fine-tuning strategies under supervised and self-supervised pre-training (PT) paradigms, using various molecular datasets. They introduce a refined method called DWiSE-FT, which consolidates the strengths of existing FT methods and demonstrates improved efficiency while maintaining promising results for regression tasks.

**Strengths:**

1. Comprehensive Evaluation:The paper conducts an extensive evaluation of different fine-tuning strategies across multiple datasets, providing a comprehensive understanding of their performance.
2. The introduction of DWiSE-FT represents a novel approach to fine-tuning, combining the strengths of existing methods.
3. The paper provides valuable insights into the design of FT methodologies and practical guidance for molecular representation learning. The findings have implications for both researchers and practitioners working in the field of molecular graph representation learning.

**Weaknesses:**

1. While the paper compares different fine-tuning strategies, it does not provide a detailed comparison of DWiSE-FT with other state-of-the-art methods in the field. This paper presents a benchmarking study, the comparisons and findings are important and novel. I want to know the importance of this proposed method DWiSE-FT , can it be presented in a research paper?
2. The abbreviation PT in Line 15 has no explanations.
3. In sec 2.2, the authors present different fine-tuning methods, the proposed Adaptive post-hoc ensemble method is similar with WiSE-FT. More information about the computational efficiency and scalability of DWiSE-FT would be beneficial.
4. Explore the performance of DWiSE-FT in combination with other pre-training paradigms.
5. The findings in this paper seems common, which is similar in other public datasets, could you provide more insights on MOLECULAR research?

**Questions:**

1. In Line111-115, why choose GraphMAE and Mole-BERT as the PT model?

---

> ### Author Response · Authors · 2024-11-22
>
> We thank reviewer CyuA for acknowledging the importance of our findings and the introduction of DWiSE-FT. We will address the listed questions below.
>
> > W1: The paper does not provide a detailed comparison of DWiSE-FT with other state-of-the-art methods in the field.
>
> Thank you for your interest in our proposed DWiSE-FT method and the feedback. We chose to compare our DWiSE-FT method with WiSE-FT and $L^2$-SP because they are the top-performing algorithms among all SOTA robust fine-tuning methods included in our benchmark, as shown in Figures 2(a) and 2(c). By doing so, we effectively compare our method against all included baselines, since these two represent the best performances. Additionally, we report the "best" scores for each experimental setting in Table 3, which encapsulate the optimal possible results considering all fine-tuning methods together. Therefore, we believe our comparisons can show the position and importance of DWiSE-FT. if the reviewer knows some potentially SOTA methods to compare, we would like to incorporate. If the rebuttal time is limited, we would like to put in the final version.
>
> > W2: The abbreviation PT in Line 15 has no explanations
>
> Thanks for pointing out the confusion. To avoid confusion, we omit using the “PT” abbreviation in the revised draft.
>
> > W3: Computational efficiency and scalability of DWiSE-FT
>
> DWiSE-FT is significantly more efficient in optimization compared to other fine-tuning methods involving re-training. DWiSE-FT is essentially a post-hoc weight interpolation without any training needed given the interpolation coefficients, but we further introduce an automatic hyperparameter search to find the optimal interpolation coefficients. Even with the hyperparameter search, it only has $l$ number of parameters to optimize where $l$ is the number of layers in the encoder. In contrast, other fine-tuning methods initially require fine-tuning on model weights that scale with $l*d$ with $d$ being the model hidden dimension. Additionally, the tuning on hyperparameters will further lead to multiple rounds of model re-training.
>
> > W4: The performance of DWiSE-FT in combination with other pre-training paradigms
>
> Thank you for highlighting this intriguing idea. Indeed, we have conducted some preliminary experiments combining DWiSE-FT with other pre-training approaches, such as graphMAE and supervised pre-training, and observed promising improvements over WiSE-FT and $L^2$-SP. However, due to time constraints, we did not include a complete evaluation in the current paper. We will include these comprehensive evaluations in a future version.

---

> ### Author Response · Authors · 2024-11-22
>
> > W5: The findings in this paper seem common, which is similar in other public datasets, could you provide more insights on MOLECULAR research?
>
> - **Evaluation on Regression Tasks**: Previous research on robust fine-tuning has largely focused on classification tasks [1, 2], especially in computer vision and NLP. In NLP, tasks like token generation are akin to classification because they involve selecting the correct token from a discrete set, rather than focusing on numerical precision. However, in molecular research, regression tasks, particularly for molecular property predictions, are prevalent and critical. Understanding robust fine-tuning for regression datasets is essential due to their numerical sensitivity. As noted in the second takeaway of our revised introduction, we found that regression tasks generally have a lower risk of overfitting compared to classification, owing to the need for precise numerical labels and detailed molecular modeling. These insights are vital for choosing suitable fine-tuning methods based on task type.
> - **Consideration of Diverse Pre-training Models**: Previous research predominantly examines self-supervised pre-trained models like CLIP [1, 2]. However, there's increasing interest in supervised pre-trained molecular models [3, 4]. By evaluating both self-supervised and supervised models, our study reflects the diverse molecular models in practice and provides insights into how pre-training affects downstream fine-tuning choices. In our revised draft introduction, we compare these models and outline fine-tuning methods suited to each in the respective takeaways.
> - **Analysis unique for Molecular Tasks**: We provide analysis specifically within the context of molecular tasks. First, in Q1 and Finding 1, we highlight the distinct needs of regression versus classification tasks: classification tasks often require coarser features, like functional groups, while regression tasks demand finer details, such as charge distribution and hydrogen bond patterns. Second, we identify the misalignment between general-purpose self-supervised pre-training in molecular settings (which focuses on substructure discovery [5]) and the task-specific requirements of downstream applications. This misalignment explains why linear probing underperforms in molecular tasks and underscores the importance of weight-based methods, which effectively integrate broad pre-trained patterns with task-specific fine-tuning to enhance performance.
>
> These molecular-specific insights have also directly inspired the development of DWiSE-FT, which leverages weight-based methods to address the unique needs of both regression and classification tasks effectively. For more detailed discussions on these findings and further insights relevant to molecular research, please refer to sections 4.1 and 4.2 of our revised paper.
>
> [1]. Kumar, Ananya, et al. "Fine-Tuning can Distort Pretrained Features and Underperform Out-of-Distribution." International Conference on Learning Representations.
>
> [2]. Wortsman, Mitchell, et al. "Robust fine-tuning of zero-shot models." Proceedings of the IEEE/CVF conference on computer vision and pattern recognition. 2022.
>
> [3]. Shoghi, Nima, et al. "From Molecules to Materials: Pre-training Large Generalizable Models for Atomic Property Prediction." The Twelfth International Conference on Learning Representations.
>
> [4]. Beaini, Dominique, et al. "Towards Foundational Models for Molecular Learning on Large-Scale Multi-Task Datasets." The Twelfth International Conference on Learning Representations.
>
> [5]. Wang, Hanchen, et al. "Evaluating self-supervised learning for molecular graph embeddings." Advances in Neural Information Processing Systems 36 (2024).
>
> > Q: Why choose GraphMAE and Mole-BERT as pre-training model
>
> We selected these two pre-training models because they represent leading strategies in both reconstruction-based and contrastive-based pre-training approaches. Additionally, they are recognized as well-known and relatively new state-of-the-art models in the field.

---

> > ### Comment · Reviewer_CyuA · 2024-11-26
> > **Response by Reviewer**
> >
> > Thanks for your time and effort. This rebuttal is too late. The authors did not answer my questions completely, especially the W1 (I want to know the importance of this proposed method DWiSE-FT, can it be presented in a research paper), I guess a pure benchmark paper is better. The Analysis unique for Molecular Tasks is good. I will keep my score.

---

> > > ### Author Response · Authors · 2024-12-01
> > >
> > > Thank you so much for your response. We see DWiSE-FT as a discovery and key finding from this benchmarking effort, and therefore, we do not intend to write a separate paper focused solely on this method. However, we greatly appreciate your suggestions and insights!

---

### Official Review · Reviewer_Cxf8 · 2024-11-02

**Soundness:** 2
**Presentation:** 3
**Contribution:** 2
**Rating:** 5
**Confidence:** 4

**Summary:**

In this work, the authors proposed a novel benchmark to understand the impact of different fine-tuning methods for pre-trained molecular foundation models in downstream tasks. Specifically, 8 fine-tuning methods grouped in 3 categories are benchmarked over 12 datasets and 36 experimental settings. Insights are then derived from the experimental results to understand which methods are best suited for which needs. In addition, a refined method named DWiSE-FT is proposed to enable more efficient fine-tuning with competitive performance.

**Strengths:**

Recently, various foundation models for molecular graphs have been proposed, however, to apply them in downstream tasks, a well-designed fine-tuning procedure is required. This work benchmarks a wide range of fine-tuning methods to provide insight into this aspect. Here are the strengths of the work:

- **Important direction.** Understanding how to select the best fine-tuning strategy for a downstream task is highly important for molecular foundation models.

- **Extensive experiments.** In this work, 8 FT methods are examined and compared across 12 datasets and 36 experimental settings, providing an empirical comparison for a variety of settings.

- **novel method DWiSE-FT.** The authors proposed DWiSE-FT which is a great candidate for regression datasets.

**Weaknesses:**

Here are the weaknesses of this work.

- **lack of new datasets.** The main contribution of this work is proposing a novel benchmark for fine-tuning. However, there are no novel datasets to provide more benchmark environments outside of existing datasets.

- **over-simplified backbone.** The authors mention that "high model expressiveness" is needed to capture the semantics of molecular datasets. However, for experiments, only a 5-layer GIN architecture is used as the backbone, which is known to be limited by the 1-WL test in expressiveness. A more powerful architecture can be used such as GraphGPS[1] or Graphormer [2]. Experiments with a more powerful backbone PT architecture can also provide more insight into the impact of the choice of PT architecture.

- **limited pre-training for Graphium.** In the Graphium paper, three categories of datasets are provided: Toymix, Largemix, and UltraLarge. It would be more interesting to observe results for PT models trained on Largemix or both Toymix and Largemix.


[1] Rampášek L, Galkin M, Dwivedi VP, Luu AT, Wolf G, Beaini D. Recipe for a general, powerful, scalable graph transformer. Advances in Neural Information Processing Systems. 2022 Dec 6;35:14501-15.

[2] Ying C, Cai T, Luo S, Zheng S, Ke G, He D, Shen Y, Liu TY. Do transformers really perform badly for graph representation?. Advances in neural information processing systems. 2021 Dec 6;34:28877-88.

**Questions:**

-  The tables in the paper are too small and very hard to read or draw conclusions from. For example, Table 3 is very small and not readable at all.  The authors should update the format of the Table or move less relevant content to the appendix.

- I can't entire agree with the claim on line 47 "pre-trained on insufficient amount of PT data
(1M-100M samples) and vocabularies". The authors made several comparisons between the molecular foundation model and foundation models from NLP / vision throughout the paper. However, the reality is that molecular data is significantly harder to curate and needs to be explicitly constructed for effective learning. Thus, I don't believe that the same scale of available data for NLP or vision would ever be available for molecular learning. For example, 100 M molecular graphs are already a large number and the focus should instead be in more to design effective PT and FT strategies. If possible, the authors should revise this claim.

---

> ### Author Response · Authors · 2024-11-22
>
> We appreciate reviewer Cxf8 for acknowledging the significance of the direction highlighted in our benchmark, as well as recognizing our efforts in benchmarking and proposing the new method. We will address the weaknesses and questions raised below.
> > W1: Lack of new datasets
>
> We would like to kindly state that "Datasets and Benchmarks" is an established focus area at ICLR. Although specific details about this direction are not outlined on the ICLR webpage, we reference the NeurIPS Datasets and Benchmarks Track, which defines its scope as encompassing "benchmarks on new or existing datasets, as well as benchmarking tools." This indicates that contributing a new dataset is not necessary; impactful work can also concentrate on developing benchmarks or tools that are applied to existing datasets.
>
> > W2: Over-simplified backbones
>
> As improving pre-training methods is beyond the scope of this work, we utilize checkpoints from existing studies. Selecting more complex backbones presents uncertainty about whether they can be effectively pre-trained using current pre-training methods.
>
> > W3: Limited pre-training for Graphium
>
> We adopted the pre-trained model on ToyMix for Graphium, as larger pre-training datasets do not currently have available checkpoints. We attempted to reproduce the pre-training on LargeMix using the Graphium library, but encountered computational constraints when processing the datasets. Additionally, using ToyMix allows for a more fair comparison with other self-supervised pre-trained models regarding the scale of the pre-training model and data. In the future, we will attempt again to obtain the LargeMix model and evaluate whether the downstream fine-tuning performances align with the trends observed using the ToyMix model.
>
> > Q: Table format and claim revision
>
> Thanks for pointing out the inappropriate wordings and points to improve in the paper presentation, we have updated those in our new draft.

---

> > ### Comment · Reviewer_Cxf8 · 2024-11-26
> > **Reviewer Reply to Authors**
> >
> > Thank you for your rebuttal and answering my questions.
> > Here are my replies to the author comments:
> > - "lack of new datasets" I understand that new datasets are not required, however I believe the contribution of the paper would be made stronger and more compelling if new datasets are included.
> > - "Over-simplified backbones" I believe at least some studies on the effect of the backbone is in scope of this work especially considering all the downstream performance and conclusion would depend on the choice of the backbone. A simple alternative backbone and comparing the results and see the differences can enrich the paper.
> > Thanks for the responses, however my concerns remain not addressed thus I will keep my score.

---

> > > ### Author Response · Authors · 2024-12-01
> > >
> > > Thank you very much for your constructive feedback and objective evaluation. We agree with your observation that, although new datasets are not strictly required given the "Call for Papers" policy, including them would certainly enhance the quality of this work. Regarding the backbone, we plan to incorporate the one trained over Largemix in the next version of this paper.

---

### Official Review · Reviewer_u2Et · 2024-11-03

**Soundness:** 2
**Presentation:** 3
**Contribution:** 2
**Rating:** 5
**Confidence:** 4

**Summary:**

The paper explores optimal Fine Tuning (FT) strategies for molecular representation, addressing challenges such as label scarcity and distribution shifts. It introduces an enhanced FT method, DWiSE-FT, tailored for diverse pre-trained molecular graph models. This method promises efficiency and automation in specific FT scenarios, consistently achieving top-ranking results.

**Strengths:**

- I have found the paper well written and self-contained. I think a non-expert could find most of the information in the paper, and I appreciate this aspect.

 - Figure 1, which demonstrate the problem domain and architecture, are interesting and easy to read. I commend the authors for their explicit effort in making these illustrations clear and informative.

 - The insights are didactical and well communicated. The conclusion given by the experiments looks interesting and valuable for future practitioners, while I think a synthesis would be beneficial for the reader.

**Weaknesses:**

Despite these merits, I have the following concerns about the paper.

1- While there is a careful analysis of the different design decisions/performance tradeoffs, I feel that there is only a limited understanding about what are the properties of the Architecture that lead to these decisions/performance differences.


2-  The study's scope, while broad, does not extend to a multi-task, multi-modality approach that could significantly enhance its applicability and impact. It does not explore foundation models across diverse scientific domains such as RNA,  and proteins nor does it address varied scientific tasks likechemical reactions. Expanding the research to cover these aspects would substantially enrich the study's utility and relevance.

3- While the study incorporates several representative FineTuning (FT) methods from various categories, it does not investigate additional FT methods from other categories that could offer significant insights. I have mentioned some noteworthy models here that deserve further exploration:
“DPA-2: A Large Atomic Model as a Multi-Task Learner [Zhang 2023]”
“Scalable Training of Trustworthy and Energy-Efficient Predictive Graph Foundation Models for Atomistic Materials Modeling: A Case Study with HydraGNN [Pasini 2024]”
“MiniMol: A Parameter-Efficient Foundation Model for Molecular Learning [Klaser 2024]”

**Questions:**

is there any sensitivity analysis on key hyperparameters of DWiSE-FT? I would also suggest comparing the hyperparameter sensitivity of DWiSE-FT to that of baseline methods.

---

> ### Author Response · Authors · 2024-11-22
>
> We thank reviewer u2Et for appreciating our paper presentation and insights provided, we will respond to the concerns accordingly below.
>
> > W1: Limited understanding about the properties of architectures that leads to the performance difference
>
> Thanks for raising the clarification question. We suppose the properties of architectures you are referring to are the mechanisms of the fine-tuning methods. To better analyze and discuss the fine-tuning performances of different methods, we categorize these 8 different fine-tuning methods to 3 groups based on their fine-tuning mechanisms. With the first takeaway in our revised draft, we point out the key insights that explain the performance of different categories of methods. For instance,
> - As discussed in finding 1, since weight-based methods have the property to ensemble weights from pre-trained and fine-tuned models, they are able to extract molecular patterns learned from general-purpose self-supervised pre-training [1]  and combine them together with task-specific knowledge from fine-tuning.
> - As discussed in finding 4, since regularization-based methods have the mechanism in enforcing close proximity of fine-tuned representation to pre-trained ones, the fine-tuned model successfully preserves domain-relevant representations learned from supervised pre-training given similar pre-training and fine-tuning tasks.
> More findings that explain the fine-tuning mechanisms can be found in section 4.
>
> [1]. Wang, Hanchen, et al. "Evaluating self-supervised learning for molecular graph embeddings." Advances in Neural Information Processing Systems 36 (2024).
>
> > W2: The study’s scope does not extend to multi-task, multi-modality foundation models nor foundation models across diverse scientific domains.
>
> Thank you for suggesting potential directions to extend our benchmark. Looking ahead, one future direction is to investigate robust fine-tuning with larger multi-task and multi-modality models. Currently, our benchmark focuses on robust fine-tuning methods for molecular graph foundation models involving small molecules. We have included diverse fine-tuning methods and representative pre-trained models to support this focus. Furthermore, our range of scope is aligned with previous robust fine-tuning studies, which commonly focus on a single pre-trained model, like CLIP, and primarily target a single application domain, such as image or object classification [2, 3]. As these prior works demonstrate, it is not essential for robust fine-tuning studies to encompass a wide array of pre-trained models across diverse applications.
>
> [2]. Kumar, Ananya, et al. "Fine-Tuning can Distort Pretrained Features and Underperform Out-of-Distribution." International Conference on Learning Representations.
>
> [3]. Wortsman, Mitchell, et al. "Robust fine-tuning of zero-shot models." Proceedings of the IEEE/CVF conference on computer vision and pattern recognition. 2022.
>
> > W3: Additional fine-tuning methods from other categories
>
> Thank you for providing the relevant works. However, the studies you mentioned primarily focus on pre-training foundation models that span multi-task applications from molecules to materials, and they do not introduce novel downstream fine-tuning techniques. Since our aim is to benchmark fine-tuning methods, we believe that these suggested works fall outside the direct scope of our current focus and are not directly comparable.
>
> > Q: Sensitivity analysis on DWISE-FT hyperparameters
>
> This is an insightful question that highlights a key aspect of fine-tuning methods. Our proposed DWiSE-FT includes hyperparameters being the initial values for the mixing coefficients in each encoder layer. Since DWiSE-FT automatically optimizes these coefficients using the validation loss, the initializations are less critical compared to the manually set hyperparameters in other methods that cannot be adjusted. In our experiments, we tested various starting points for the mixing coefficients within the range of [0,1], and found that the variance in final results was small. This is because the coefficients converge to their optimal values, regardless of the initial settings, given adequate training time. We have also clarified this in the revised draft, specifically in Appendix E, where we discuss the hyperparameter tuning.

---

### Author Response · Authors · 2024-11-22

Dear Reviewers,

We sincerely appreciate your time and effort in providing us with valuable feedback to enhance our paper. We are grateful for your recognition of the importance of our benchmark, which offers valuable insights to the field, and for your appreciation of our comprehensive evaluation and the novel method we proposed, DWiSE-FT.

Additionally, we thank you for raising insightful questions and suggestions. We will address each of these points in our detailed responses to individual reviewers. In the new paper draft, we have polished the writing and adjusted the presentation for clarity, while maintaining all the original conclusions and results. We are also incorporating MoleculeSTM [1], as suggested by reviewer iuUw as an additional pre-trained model for our downstream fine-tuning evaluations. We are currently benchmarking the results and will incorporate the available findings in the updated paper draft by the end of the rebuttal period. However, due to time constraints and computational costs, some results for the large MUV and HIV datasets may not be finalized in time for the rebuttal period, and we will include them in a future version.

[1] Liu, Shengchao, et al. "Multi-modal molecule structure–text model for text-based retrieval and editing." Nature Machine Intelligence 5.12 (2023): 1447-1457.

---

### Meta-Review · Area_Chair_rmvi · 2024-12-18

**Metareview:**

### Summary
The paper explores fine-tuning (FT) strategies for pre-trained molecular graph models, focusing on improving downstream performance under label scarcity and distribution shifts. The study benchmarks 8 FT methods across 12 datasets and 36 experimental settings, highlighting insights for regression and classification tasks. A new method, DWiSE-FT, is proposed to enhance fine-tuning efficiency, achieving competitive results, particularly for regression tasks.

### Strengths
- The study systematically evaluates 8 FT methods across multiple datasets and experimental settings, offering valuable insights into FT strategies for molecular graph models.
- The benchmark closely simulates real-world scenarios such as OOD generalization and few-shot learning, making the findings useful for practitioners.
- The proposed DWiSE-FT method consolidates existing FT strategies, providing improved efficiency and competitive performance, especially for regression tasks.

### Weaknesses
- Limited Scope of Pre-training and Architectures: The study uses relatively small-scale pre-trained models (e.g., Mole-BERT, Graph MAE) and a limited backbone architecture (5-layer GIN), which lacks expressiveness. The evaluation does not explore more powerful models like GraphGPS, Graphormer, or larger-scale pre-training paradigms.
- Lack of Novelty: While DWiSE-FT performs well, it is primarily a refinement of existing FT strategies rather than a fundamentally novel method. Its computational efficiency and scalability compared to other FT methods are not clearly demonstrated.
- The study focuses solely on molecular graphs and does not extend to multi-task, multi-modal scenarios (e.g., RNA, protein structures, or chemical reactions). The findings lack deeper insights specific to molecular research, with conclusions appearing generalizable to other public datasets.


While the paper provides a valuable benchmark of FT strategies for molecular graph models and introduces an improved method (DWiSE-FT), it suffers from limited novelty, narrow evaluation scope, and lack of scalability tests. The reliance on small-scale pre-trained models and backbones weakens the paper's impact. Extending the study to more powerful architectures, diverse pre-training paradigms, and broader tasks would significantly enhance its contributions.

**Additional Comments On Reviewer Discussion:**

The major concerns raised by the reviewers are summarized in the above weaknesses. The authors added MoleculeSTM as a pre-trained model to address the concern regarding the size of the pretrained model. However, the authors could not convince the reviewers regarding the novelty of this work, which adopts existing FT strategies rather than introducing novel methods.

---

### Decision · Program_Chairs · 2025-01-22

Reject